# The immersion freezing behavior of ash particles from wood and brown coal burning

Sarah Grawe[1], Stefanie Augustin-Bauditz[1], Susan Hartmann[1], Lisa Hellner[1], Jan B. C. Pettersson[2], Andrea Prager[3], Frank Stratmann[1], and Heike Wex[1]

[1]Leibniz Institute for Tropospheric Research, Leipzig, Germany.
[2]Department of Chemistry and Molecular Biology, University of Gothenburg, Gothenburg, Sweden.
[3]Leibniz Institute of Surface Modification, Leipzig, Germany.

*Correspondence to:* Sarah Grawe (grawe@tropos.de)

**Abstract.**

It is generally known that ash particles from coal combustion can trigger ice nucleation when they interact with water vapor and/or supercooled droplets. However, data on the ice nucleation of ash particles from different sources, including both anthropogenic and natural combustion processes, is still scarce. As fossil energy sources still fuel the largest proportion of electric power production worldwide, and biomass burning contributes significantly to the global aerosol loading, further data is needed to better assess the ice nucleating efficiency of ash particles. In the framework of this study, we found that ash particles from brown coal (i.e., lignite) burning are up to two orders of magnitude more ice active in the immersion mode below -32 °C than those from wood burning. Fly ash from a coal-fired power-plant was shown to be the most efficient at nucleating ice. Furthermore, the influence of various particle generation methods on the freezing behavior was studied. For instance, particles were generated either by dispersion of dry sample material, or by atomization of ash-water suspensions, and then led into the Leipzig Aerosol Cloud Interaction Simulator (LACIS) where the immersion freezing behavior was examined. Whereas the immersion freezing behavior of ashes from wood burning was not affected by the particle generation method, it depended on the type of particle generation for ash from brown coal. It was also found that the common practice of treating prepared suspensions in an ultrasonic bath to avoid aggregation of particles led to an enhanced ice nucleation activity. The findings of this study suggest (a) that ash from brown coal burning may influence immersion freezing in clouds close to the source and (b) that the freezing behavior of ash particles may be altered by a change in sample preparation and/or particle generation.

## 1 Introduction

Gaining a comprehensive knowledge of the formation and behavior of ice particles in clouds is of the utmost importance to achieve a better representation of ice related processes in weather and climate models (Koop and Zobrist, 2009). Ice particles in clouds can be formed either by primary mechanisms, i.e., homogeneous and heterogeneous ice nucleation (Pruppacher and Klett, 1997), or by secondary mechanisms (Heymsfield and Willis, 2014, and references therein). In the atmosphere, homogeneous ice nucleation, i.e., the freezing of pure water or solution droplets, takes place at temperatures below -38 °C (Rosenfeld and Woodley, 2000; Koop et al., 2000; Murray et al., 2010a). At water saturation, this temperature limit of droplet

freezing is raised in the presence of so-called ice nucleating particles (INPs) acting as catalysts for ice formation which is then referred to as immersion freezing.

Combustion aerosol, accounting for a large fraction of the global aerosol loading, has been frequently investigated concerning its freezing behavior. It was discovered that soot (DeMott, 1990; Diehl and Mitra, 1998), metal oxides from furnaces and smelters (Szyrmer and Zawadzki, 1997), lead containing particles (Schäfer, 1975; Cziczo et al., 2009), and aerosol from biomass burning (Petters et al., 2009; Prenni et al., 2012; McCluskey et al., 2014) are able to act as INPs. To date, only a few studies on the freezing behavior of ash particles have been conducted (Schnell et al., 1976; Parungo et al., 1978a; Pueschel et al., 1979; Havlíček et al., 1989, 1993; Umo et al., 2015). Ash is defined as the solid material, which remains after the combustion of organic substances (e.g., fossil fuels, biofuels, and plant parts). It mainly consists of noncombustible components in the fuel such as mineral inclusions and of so-called heteroatmos, i.e., atoms other than carbon and hydrogen (Flagan and Seinfeld, 1988). This is what separates ash from carbonaceous particles, e.g., soot. During the combustion process, a fraction of fine ash particles is directly emitted into the atmosphere together with flue gases whereas coarse ash particles mainly remain in the fireplace, boiler, or on the ground after a wildfire and may be lofted by the action of wind (Andreae et al., 2004). The former is termed "fly ash", the latter is referred to as "bottom ash".

The globally increasing electric power demand is expected to continue to be covered by fossil fuels, making power generation by coal combustion grow faster than all renewable energy sources put together (International Energy Agency, 2012). Besides ash being the primary coal combustion by-product (Kalyoncu and Olson, 2001), ash particles are also formed during biomass burning including wildfires. The impact of ash particles as potential INPs must be put into perspective by comparing ash emission rates to those of other INP containing aerosols, e.g., mineral dust which is present in the atmosphere in abundance. An estimate for fly ash from coal combustion yields global annual emissions of 30 Mt in the year 2000 (Smil, 2008), whereas global annual dust emissions are estimated to be as high as 700 Mt to 3000 Mt per year (Textor et al., 2006). As there are no further values concerning the amount and distribution of different types of ash in the atmosphere, it is a difficult task to assess their impact on heterogeneous ice nucleation on a larger scale. DeMott et al. (2003), who examined atmospheric particles regarding their ability to function as INPs at cirrus temperatures over a period of 20 days, give a value of approximately 7 % of ice crystal residues, which were formed under conditions favorable to heterogeneous nucleation, to be fly ash particles. The horizontal dispersion of these particles has been shown by Zhang et al. (2011) who found fly ash in surface snow crystal residues at a remote central Asian glacier. Backward air mass trajectories indicated that the particles originated from strongly populated areas to the west of the sampling site and were transported over thousands of kilometers through the high-level westerly jet stream. It must be mentioned that the chemical composition of mineral dust and ash is very similar, i.e., both include several common mineral components such as Si, Na, Ca, Fe and oxides (Cziczo et al., 2004). Because of this difficulty, DeMott et al. (2003) and Zhang et al. (2011) used the spherical shape of fly ash particles, originating from the combustion process in the furnace, as a criterion to distinguish from other particle types. A discussion about the atmospheric relevance of coal fly ash particles in terms of their influence on heterogeneous ice nucleation can be found in Sect. 4.

Already in the 1960s, first presumptions arose that aerosol particles in the plumes of coal-fired power plants might be efficient at nucleating ice. Reasons for this were observations of ice fog (Benson, 1965) and so-called "industrial snow" (Agee, 1971;

Parungo et al., 1978b) in close proximity to the stacks. Laboratory studies showed that coal fly ash particles are able to serve as INPs in the deposition (Parungo et al., 1978a; Havlíček et al., 1993) and immersion modes (Havlíček et al., 1989, 1993). Apart from untreated fly ash samples, Havlíček et al. (1993) also investigated the freezing behavior of the insoluble fraction alone and found up to 3 orders of magnitude less INPs in samples freed from water soluble components in comparison to the

untreated samples at -15 °C. Additionally, Havlíček et al. (1993) found that water soluble components, which were dominantly composed of anhydrous $CaSO_4$, were responsible for differences in the ice nucleation ability of fly ash samples from different power plants. In comparison to coal fly ash, bottom ashes from coal and biomass burning have been rarely investigated. Recently, Umo et al. (2015) conducted first experiments on the immersion freezing behavior of bottom ash particles from coal and wood burning and compared the results to the ice nucleation ability of coal fly ash. These measurements were done with

the help of a cold stage setup (Murray et al., 2010a; Whale et al., 2015). For Umo et al. (2015), droplets from 0.1 wt% ash-water suspensions were firstly pipetted ($\mu$L droplets) and secondly nebulized (nL droplets) onto a glass slide which was placed on a cooled plate. The samples were investigated in temperature ranges from -12 °C to -36 °C (bottom ashes) and from -16 °C to -31 °C (coal fly ash), respectively. It was shown that bottom ashes nucleate ice in the immersion mode. Additionally, it was found that the fly ash particles are more efficient at nucleating ice than the bottom ash particles in a temperature range from -17

°C to -27 °C. At -20 °C there is a difference of 2 orders of magnitude in $n_s$ when comparing coal fly ash to the most efficient of the bottom ashes. The bottom ashes differed from each other by two orders of magnitude in $n_s$, with wood bottom ash being the most efficient and coal bottom ash being the least efficient at nucleating ice. Umo et al. (2015) suggest that the different fuels and combustion temperatures causing changes in composition and morphology are the reason for the difference between bottom ash and fly ash.

As information concerning the ice nucleation efficiency of ash particles is still sparse, further investigations are needed to work out differences and similarities between the freezing behavior of ashes of varying origin and composition. In the present study, the immersion freezing behavior of five different ash samples, similar but not identical to those investigated by Umo et al. (2015), was quantified at the Leipzig Aerosol Cloud Interaction Simulator (LACIS, Hartmann et al., 2011). With our experimental setup, it was possible to study the influence of particle generation on the measured ice fractions as particles were

produced both by dispersion of dry sample material and atomization of ash-water suspensions. Suspensions were prepared according to the method described in Umo et al. (2015), which includes treatment in an ultrasonic bath and subsequent stirring. As similar procedures are often used in the sample preparation for ice nucleation experiments, the effect of ultrasonic treatment of the sample on the immersion freezing behavior was investigated as well.

## 2 Methods and materials

### 2.1 Experimental setup

#### 2.1.1 Particle generation and size selection

Airborne ash particles were generated in two different ways: a) dispersion of dry sample material and b) atomization of ash-water suspensions. Airborne particles from dry ash were generated using an aerosol generator consisting of a tilted glass bottle which is connected to an electric imbalance motor (Rösch, 2015). Dry ash particles being situated at the bottom of the bottle become airborne along with particle free pressurized air streaming into the bottle through a tube. Coarse material, which does not leave the bottle through the outlet at the top and deposits on its walls, is continually transported downwards. This is due to vibrations caused by the motor and the mounting of the bottle at a certain angle. The efficiency of the aerosol generator can be enhanced by mixing millimeter-sized glass beads into the samples which was done for the fine sample material in this study. We do not expect the beads to have any influence on the surface properties of the particles, as only a small number of 20 beads was used which did not appear milky after several hours of particle generation. In addition to this herein called dry particle generation, particles were generated from an ash-water suspension using a custom-built atomizer (similar to TSI Model 3076) and a diffusion dryer unit. In the following, this procedure will be referred to as wet particle generation.

Before the aerosol was brought to a bipolar charge equilibrium inside a neutralizer, particles passed a cyclone (cut-off diameter 450 nm) which was installed to minimize the amount of multiply charged particles. After each set of measurements with LACIS, lasting usually 20 min, the cyclone was cleaned. For all experiments presented here, 300 nm particles were selected by means of a differential mobility analyzer (DMA, Vienna type, medium, Knudson and Whitby, 1975). 300 nm particles were chosen because, at this size, there were sufficiently high and stable number concentrations for each of the five ash samples. Furthermore, industrial particle removal techniques tend to be less efficient for submicron particles (Flagan and Seinfeld, 1988), which is supported by size distribution measurements downstream of a coal-fired power-plant featuring a bimodal distribution with a second mode diameter at 300 nm (Parungo et al., 1978a). Selecting other particle sizes could be expected to lead to differences in freezing behavior due to a size dependent particle composition and/or morphology (Wheeler et al., 2014). However, in case of bottom ash, measurements were performed with 500 nm brown coal bottom ash particles which showed no difference in $n_s$ within the range of our measurement uncertainty compared to the measurements with 300 nm particles as will be demonstrated later (see Fig. 5). For fly ash we would not expect to see a size dependence of $n_s$, at least not in the size range we are able to select, because there is evidence that the chemical composition of fly ash particles with aerodynamic diameters between 0.2 $\mu$m and 4.8 $\mu$m is remarkably consistent (Kaufherr and Lichtman, 1984).

As for insoluble substances the ice nucleation efficiency depends on the surface area of the INP (Archuleta et al., 2005; Welti et al., 2009; Pinti et al., 2012; Hartmann et al., 2016), multiply charged particles in the investigated aerosol are more ice active than singly charged particles due to the presence of a larger surface area. To account for the multiply charged particles, measurements were performed with an Ultra High Sensitivity Aerosol Spectrometer (UHSAS, DMT, Boulder, CO, USA). An example of the UHSAS measurements with dry generated brown coal fly ash particles after size selection can be seen in Fig. 1.

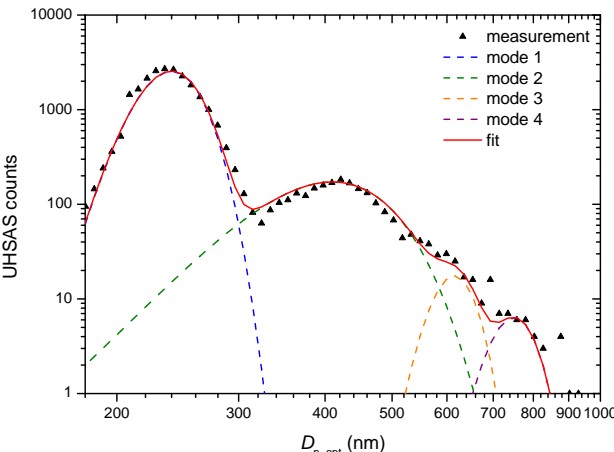

**Figure 1.** UHSAS measurements of dry dispersed brown coal fly ash particles after size selection of 300 nm particles with a DMA. Shown is a mean over the first 30 min of dry particle generation. The red line represents the least squares fit of a four-modal normal distribution to the measurement. The integrals over each of the four modes represent the amount of singly, doubly, triply, and quadruply charged particles. Here, the multiple charge fractions were determined to 79.8 % singly, 18.9 % doubly, 0.9 % triply, and 0.4 % quadruply charged particles.

Note that the UHSAS detects the optical particle diameter which is smaller than the electrical mobility diameter in the shown example. To determine the multiple charge fractions, a four-modal normal distribution was fit to the measured data and the respective integrals were determined. In this case, we found 79.8 % singly, 18.9 % doubly, 0.9 % triply, and 0.4 % quadruply charged particles, i.e., the triply and quadruply charged particles contribute only 6 % to the total surface area. Because of the low fraction of particles with three or more negative charges, which was in the range of 1 % for all of the dry dispersed ash samples, only the doubly charged particles (fractions given in Table 1) were accounted for the correction of the immersion freezing experimental results. A detailed explanation of the multiple charge correction procedure is given by Hartmann et al. (2016).

A cloud condensation nuclei counter (CCNc, CCN-100, DMT, Boulder, CO, USA, Roberts and Nenes, 2005) was operated at certain times to investigate the hygroscopicity of size selected particles. Care had been taken to set up similarly long sampling lines from the DMA to LACIS and the UHSAS and CCNc, respectively, in order to avoid a difference in particle losses.

### 2.1.2 LACIS

The immersion freezing behavior of the previously generated and size selected ash particles was investigated with LACIS. In comparison to cold stage methods, where a set of suspension droplets is brought onto a cooled surface, LACIS offers the opportunity to examine airborne droplets. Furthermore, as water is brought into the system via the gas phase, impurities which are known to cause the freezing of pure water droplets above the homogeneous freezing limit on cold stages (Budke

**Table 1.** Fractions of doubly charged particles generated from dry sample material as determined from UHSAS measurements. The selected particle size was 300 nm. The fraction of particles with three or more negative charges was determined to be in the range of 1 % for all samples and hence neglected.

| Sample | Doubly charged fraction |
| --- | --- |
| Spruce bottom ash | 0.12 |
| Birch bottom ash | 0.24 |
| Beech bottom ash | 0.06 |
| Brown coal bottom ash | 0.07 |
| Brown coal fly ash | 0.19 |

and Koop, 2015; Whale et al., 2015), can be ruled out for our experiments. LACIS consists of seven connected 1 m long tube sections with an inner diameter of 15 mm. The 2 mm wide particle beam, being surrounded by humidified, particle free sheath air, is situated along the center line of the tube. As each of the seven sections can be temperature controlled individually with the help of thermostats, particles pass along defined temperature and saturation profiles. For the measurements presented here, LACIS was operated in a way that each particle was activated to a droplet in the second to last section. Further cooling caused a certain fraction of droplets, hereafter referred to as ice fraction $f_{ice}$, to freeze within a nucleation time of 1.6 s. The discrimination between supercooled droplets and ice particles was realized with the help of the Thermo-stabilized Optical Particle Spectrometer for the detection of Ice (TOPS-Ice, Clauß et al., 2013). The approach to determining the phase state of the hydrometeors is based on the fact that the former polarization of light is maintained for scattering at spherical hydrometeors (supercooled water droplets) while non-spherical hydrometeors (ice particles) cause depolarization.

For each measurement at a given temperature, at least 2000 particles were detected. Due to this large number of counted particles and the small temperature uncertainty of $\pm$ 0.3 K, LACIS measurements are very reproducible.

## 2.2 Sample preparation and characterization

Five different kinds of ashes were investigated concerning their immersion freezing behavior (see Table 2). While LACIS measurements were performed with all five samples from dry generation, only selected samples were also generated from a suspension. This is due to the fact that we chose not to investigate those samples further which already featured an ice nucleation efficiency close to the TOPS-Ice detection limit for dry particle generation. Bottom ashes from spruce, birch, and beech burning were examined to study the effect of wood type on the freezing behavior of the respective ash particles. Wood logs including bark were burned without the addition of leave material or small branches. The wood was stored for drying prior to the combustion process. It has to be noted that the coal bottom and fly ashes stem from lignite burning but will be referred to as "brown coal ashes", corresponding to the generic term. The brown coal bottom and fly ashes were not produced from brown coal with identical compositions. All bottom ash samples were taken from domestic heaters after the combustion of the

**Table 2.** Sample overview with temperature range in which ice nucleation (IN) was observed.

| Ash type | Combustion material | Dry generation IN observed T range | Wet generation +US IN observed T range | Wet generation -US IN observed T range |
|---|---|---|---|---|
| Bottom ash | Wood (spruce) | $\leq$ -35 °C | $\leq$ -34 °C | - |
|  | Wood (birch) | $\leq$ -35 °C | - | - |
|  | Wood (beech) | $\leq$ -35 °C | - | - |
|  | Brown coal | $\leq$ -32 °C | $\leq$ -35 °C | - |
| Fly ash | Brown coal | $\leq$ -27 °C | $\leq$ -24 °C | $\leq$ -34 °C |

pure substances. The fly ash sample was extracted from the electrostatic precipitators of the Lippendorf power station, which is situated 15 km south of Leipzig, Germany, and has a power output of 1840 MW.

Dry samples were placed in the aerosol generator without further preparations. The ash-water suspensions were prepared as described in Umo et al. (2015) with bottom ashes from brown coal and spruce burning, as well as fly ash from brown coal burning. For this, a suspension of 0.05 wt% ash in Milli-Q® water was placed in an ultrasonic bath (RK100H Sonorex Super, BANDELIN electronic GmbH & Co. KG, Berlin, Germany) for 10 minutes. According to Umo et al. (2015), this step is necessary to break down ash aggregates. It has already been shown that the size distribution of soil particles in a suspension can be modified by ultrasonic dispersion (Oorts et al., 2005). To see whether a treatment with the ultrasonic bath influences the ice nucleation efficiency of the suspension particles as well, a fly ash suspension sample was prepared without ultrasonic treatment. Afterwards, all samples were stirred for $\approx$ 24 hours. Additionally to this procedure, part of the brown coal fly ash suspension sample was filtered using syringe filters (200 nm pore size, Millex™, Merck KGaA, Darmstadt, Germany) to remove the majority of fly ash particles from the suspension and leave water soluble material.

Four of the dry samples were investigated by means of inductively coupled plasma-sector field mass spectrometry (ICP-SFMS, Zheng and Yamada, 2006) at ALS Scandinavia AB (Luleå, Sweden). Beech bottom ash was provided late in the course of the experiments and hence not analyzed for its chemical composition. Fig. 2 and Table 3 show the results of the chemical composition analysis, i.e., the mass fractions of certain oxides and elements. The former were estimated by recalculating the measured concentrations of major ions into their most common oxide forms. Any missing percentage is due to other than the major elements and the fact that other counterions were involved. E.g., apart from $K_2O$, K may also occur in the form of KCl or $K_2CO_3$.

Fly ash from brown coal burning contains 15 % more $SiO_2$ than any other sample which qualitatively corresponds to findings by Umo et al. (2015). Also, bottom ash from brown coal burning includes 4 % more $SiO_2$ than spruce bottom ash and 12 % more than birch bottom ash. On the other hand, the wood bottom ashes contain more $K_2O$ than those from coal burning. This is an important point as K in biomass burning ash, in contrast to K in coal ash, is largely water soluble (Steenari et al., 1999b) and might influence its ice nucleation efficiency. As for single elements, it is most striking that the brown coal ashes include

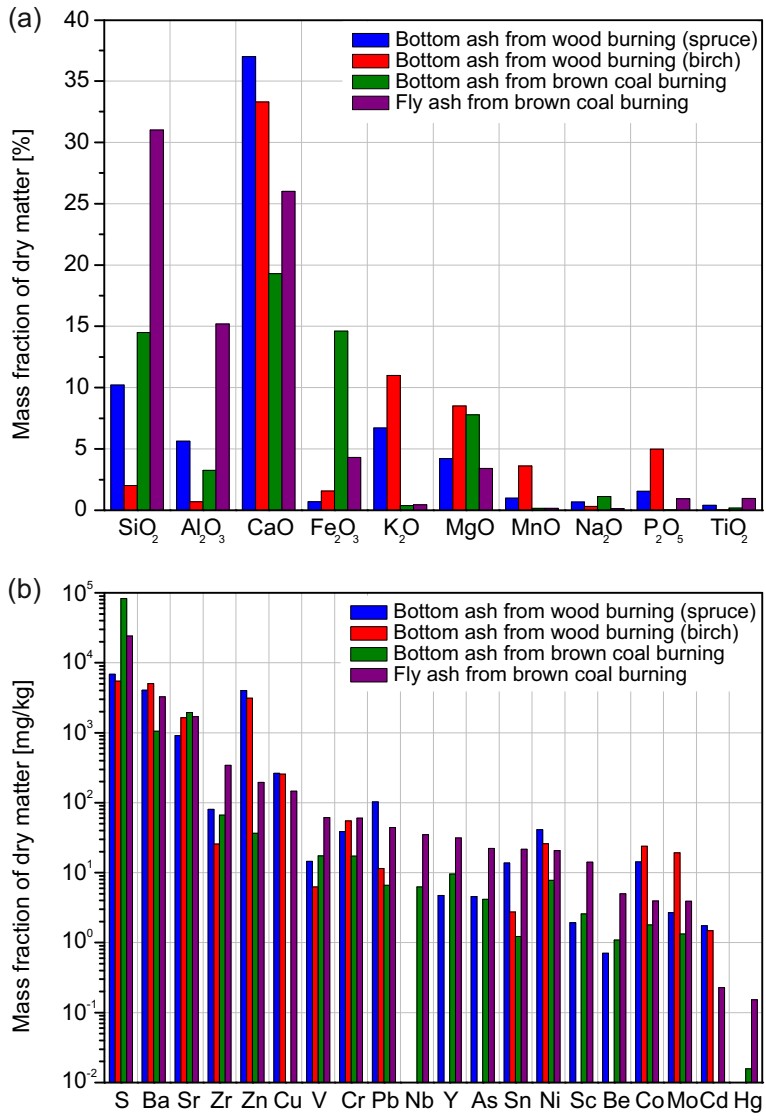

**Figure 2.** Mass fractions of (a) certain oxides and (b) additional elements in the dry matter of bottom ashes from spruce, birch and brown coal burning, as well as fly ash from brown coal burning. The analysis was performed by means of inductively coupled plasma-sector field mass spectrometry (ICP-SFMS) at ALS Scandinavia AB (Luleå, Sweden). Table 3 shows the values given in the bar chart and the associated uncertainties.

Hg which cannot be found in the wood ash samples. Suggestions on how these differences might affect the immersion freezing behavior of the different ash types can be found in Sect. 3.

To investigate particle shape and surface properties, scanning electron microscope (SEM) images were taken. Bottom and fly ash particles from dry and wet generation with electrical mobility diameters of $D_{p,el}$ = 300 nm were generated as described

**Table 3.** Mass fractions in % of certain oxides and elements in the dry matter of bottom ashes from spruce, birch, and brown coal burning, as well as fly ash from brown coal burning. The numbers given here correspond to the bars depicted in Fig. 2. The loss on ignition (LOI) test was performed at 1000 °C to assess the completeness of the combustion processes.

| | Spruce bottom ash | Uncertainty (±) | Birch bottom ash | Uncertainty (±) | Brown coal bottom ash | Uncertainty (±) | Brown coal fly ash | Uncertainty (±) |
|---|---|---|---|---|---|---|---|---|
| Oxide | Mass fraction of dry matter [%] | | | | | | | |
| $SiO_2$ | 10.2 | 1.5 | 2.03 | 0.3 | 14.5 | 2.1 | 31 | 4.6 |
| $Al_2O_3$ | 5.64 | 0.95 | 0.69 | 0.117 | 3.25 | 0.55 | 15.2 | 2.6 |
| CaO | 37 | 6.2 | 33.3 | 5.6 | 19.3 | 3.3 | 26 | 4.4 |
| $Fe_2O_3$ | 0.698 | 0.133 | 1.57 | 0.29 | 14.6 | 2.9 | 4.3 | 0.81 |
| $K_2O$ | 6.7 | 1.35 | 11 | 2.2 | 0.392 | 0.078 | 0.451 | 0.09 |
| MgO | 4.22 | 0.88 | 8.5 | 1.76 | 7.79 | 1.67 | 3.4 | 0.71 |
| MnO | 0.988 | 0.134 | 3.61 | 0.48 | 0.164 | 0.022 | 0.148 | 0.02 |
| $Na_2O$ | 0.67 | 0.125 | 0.321 | 0.06 | 1.12 | 0.21 | 0.144 | 0.027 |
| $P_2O_5$ | 1.55 | 0.25 | 4.99 | 0.77 | 0.0211 | 0.0046 | 0.934 | 0.145 |
| $TiO_2$ | 0.412 | 0.075 | 0.0247 | 0.0045 | 0.178 | 0.037 | 0.969 | 0.175 |
| $\sum$ | 68.1 | - | 66 | - | 61.3 | - | 82.5 | - |
| LOI (1000 °C) | 22.9 | 5 | 26.7 | 5 | 10.5 | 5 | -0.8 | 5 |
| Element | Mass fraction of dry matter [mg/kg] | | | | | | | |
| As | 4.57 | 8.04 | <3 | - | 4.17 | 7.97 | 22.2 | 8.9 |
| Ba | 4050 | 667 | 5060 | 827 | 1050 | 171 | 3260 | 557 |
| Be | 0.708 | 0.271 | <0.5 | - | 1.09 | 0.34 | 4.98 | 0.87 |
| Cd | 1.75 | 0.34 | 1.48 | 0.31 | <0.1 | - | 0.228 | 0.135 |
| Co | 14.2 | 3.2 | 23.9 | 5.4 | 1.79 | 0.56 | 3.93 | 0.98 |
| Cr | 38.5 | 7.2 | 54.7 | 10 | 17.3 | 3.3 | 60.1 | 11 |
| Cu | 263 | 57 | 256 | 56 | <10 | - | 146 | 32 |
| Hg | <0.01 | - | <0.01 | - | 0.0157 | 0.0047 | 0.152 | 0.032 |
| Mo | 2.69 | - | 19.2 | 3.5 | 1.33 | - | 3.92 | - |
| Nb | <5 | - | <5 | | 6.26 | 0.93 | 34.8 | 4.9 |
| Ni | 41.2 | 9.9 | 25.9 | 6.2 | 7.8 | 2.04 | 20.6 | 5 |
| Pb | 103 | 21 | 11.5 | 2.8 | 6.61 | 2.01 | 44.4 | 9.1 |
| S | 6830 | - | 5460 | - | 83000 | - | 24200 | - |
| Sc | 1.93 | 0.63 | <0.9 | - | 2.58 | 0.66 | 14.1 | 2.8 |
| Sn | 13.8 | - | 2.75 | - | 1.23 | - | 21.6 | - |
| Sr | 913 | 137 | 1640 | 250 | 1940 | 313 | 1700 | 280 |
| V | 14.5 | 2.2 | 6.27 | 1.18 | 17.4 | 2.3 | 61.1 | 7.9 |
| W | <50 | - | <50 | - | <50 | - | <50 | - |
| Y | 4.74 | 0.72 | <2 | - | 9.6 | 1.41 | 31.3 | 4.6 |
| Zn | 4010 | 782 | 3120 | 613 | 36.4 | 7.9 | 195 | 38 |
| Zr | 80.1 | 20.2 | 25.6 | 6.4 | 66.7 | 18 | 341 | 45 |

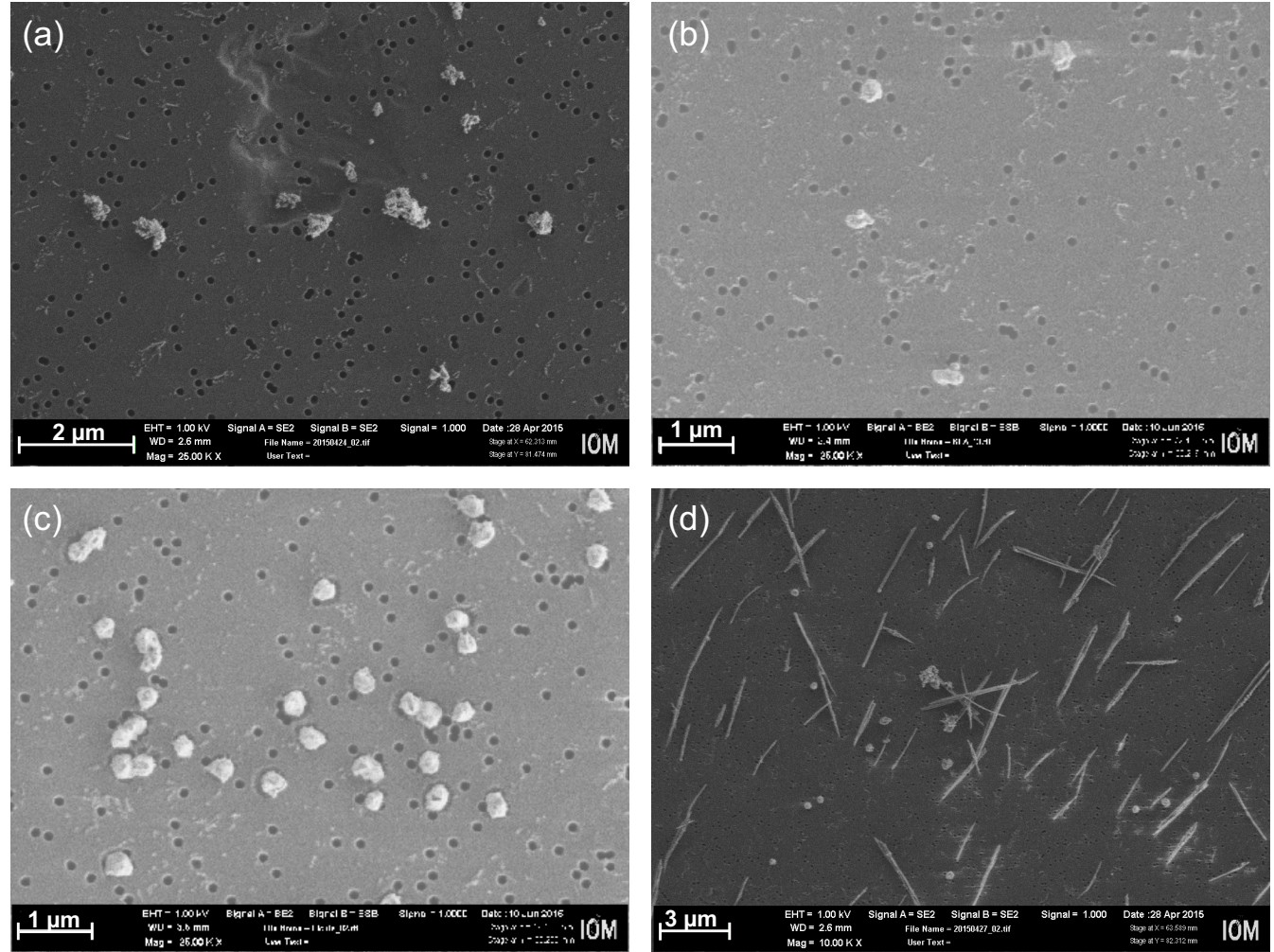

**Figure 3.** Scanning Electron Microscope (SEM) images of (a) bottom ash particles from brown coal burning (dry generation), (b) fly ash particles from brown coal burning (dry generation), (c) bottom ash particles from spruce burning (wet generation), and (d) fly ash particles from brown coal burning (wet generation). In each case particles with $D_{p,el}$ = 300 nm were selected. Note the different magnification.

in Sect. 2.1.1, collected on Nuclepore$^{TM}$ track-etched membrane polycarbonate filters (Sigma-Aldrich, St. Louis, MO, USA), and pictured by the SEM. Fig. 3 shows images of (a) brown coal bottom ash (dry generation), (b) brown coal fly ash (dry generation), (c) spruce bottom ash (wet generation), and (d) brown coal fly ash (wet generation). The high amount of particles larger than 300 nm in Fig. 3 (a) is due to the fact that the cyclone was not emptied sufficiently often during the several hours of collecting particles from dry generation leading to a disproportionate accumulation of multiply charged particles which were not present in the LACIS experiments (experiments are terminated after 30 min at most because of wall glaciation effects). We did not observe this accumulation for wet particle generation. For further examination of this issue, we performed an

additional experiment in which we used an aerodynamic particle sizer (APS, TSI model 3321) in addition to the UHSAS to detect the particle size distribution after size selection. It was observed that after roughly 30 min of sampling from dry particle generation, the fraction of supermicron particles increased strongly. Cleaning of the cyclone removed the occurrence of these supermicron particles. While the cyclone was only cleaned every few hours during the sampling of the filter shown in Fig.

3 (a), it was cleaned every 30 min for the filters sampled later on (see Fig. 3 (b)). On these, only singly and doubly charged particles occurred where the fraction of doubly charged particles derived from counting was in agreement with the respective UHSAS measurements (see Table 1).

The direct comparison of the brown coal ashes from dry generation shows differences in particle shape: The bottom ash particles appear more irregular and fractal-like than the fly ash particles. This difference might be related to the carbonaceous

content of the two samples as particles with high carbon content tend to develop irregular structures due to aggregation (Hiranuma et al., 2008). From the loss on ignition analysis (LOI, see Table 3), we know that the brown coal bottom ash still contains 11 % of unburnt fuel, i.e., carbonaceous particles which form during incomplete combustion (Kucbel et al., 2016). The brown coal combustion in the power plant, however, was very efficient, as no organic fraction, i.e., no carbon could be found in this sample. This observation is supported by specific surface area measurements of coal fly ash and bottom ashes

from coal and biomass burning, where the lowest value was detected for fly ash (Umo et al., 2015).

Apart from a small number of 300 nm sized fly ash particles, the SEM image of brown coal fly ash particles from wet generation (see Fig. 3 (d)) shows a majority of particles that appear to be needle shaped crystals. These crystals appear to be several microns long. It is reasonable to assume that the crystals consist of water soluble components which exist in the suspension separated from the fly ash particles. Inside the atomizer, droplets are formed which can consist of either only

soluble material, or additional insoluble particles. The subsequent drying process supposedly leads to a crystallization of the soluble components. This crystal formation does not take place in the experiments by Umo et al. (2015) as droplets are directly produced from the suspension. In this case, the components which are present in the crystals are dissolved in the droplets. There are several possible implications from the presence of crystals in the immersion freezing experiments with fly ash suspension particles which will be discussed in Sect. 3. Possibly, $CaCO_3$ is the dominant phase of the water soluble fraction. During the

combustion process, $CaO$ is produced (present in the initial sample, see Fig. 2 and Table 3) which may react with $H_2O$ to form $Ca(OH)_2$. $CaCO_3$ may form from $Ca(OH)_2$ upon reaction with $CO_2$ from the air (Steenari et al., 1999b). That we do not see any needles on the SEM image of spruce ash suspension particles (Fig. 3 (c)), even though this sample contains 11 % more $CaO$ than the fly ash sample, is possibly due to the variety of different crystal shapes which $CaCO_3$ is known to occur in and which include needles, hexagonal plates, and others (Kim et al., 2009). Particularly hexagonal plates might not be as easily

distinguishable from the insoluble particles as the prominent needles and might be seen, at least to some extent, in Fig. 3 (c), where the resolution of the pictures unfortunately does not allow a better analysis.

The presence of Ca in the needles was confirmed by investigating filter samples by means of Scanning Electron Microscopy coupled with Energy Dispersive X-ray (SEM/EDX) spectroscopy. However, there is no definite proof that $CaCO_3$ is the dominant phase of the water soluble fraction of fly ash particles from wet generation. It has to be mentioned that Havlíček et al.

(1993) found a majority of $CaSO_4$ in the soluble fractions of most of their investigated fly ash samples. This is unlikely for our fly ash sample as the amount of S was below the detection limit of the EDX on all of the examined filter sections.

In addition to the above discussed samples, also a filtered fly ash suspension sample was used to generate size segregated particles. These were then also collected and examined under the SEM (not shown here). Almost none of the approximately spherical insoluble fly ash particles were observed under the SEM, from which it can be interpreted that the filtering process removed the insoluble material almost entirely. We are referring to the remaining compounds as "water soluble" even though we are aware that insoluble material smaller than the filter pore size of 200 nm might still be present. This is justified, because the selected 300 nm particles predominantly include soluble substances. CCNc measurements with particles from the filtered fly ash suspension indicate a low hygroscopicity ($\kappa = 0.06 \pm 0.01$). However, this does not necessarily mean that the components in the generated particles are not soluble. Sullivan et al. (2009) give a value of $\kappa = 0.011$ for $CaCO_3$, which is weakly soluble (Plummer and Busenberg, 1982). The generated particles could hence be composed of a mixture of $CaCO_3$ together with other compounds.

## 3    Results and discussion

The immersion freezing behavior of the five ash samples was investigated in a temperature range from -24 °C to -40 °C. Fig. 4 shows the obtained $f_{ice}$ values for particles from dry (see full circles in Fig. 4) and wet generation (see open circles in Fig. 4 (b), (c), (d)). Four data sets were obtained for fly ash from brown coal burning (see Fig. 4 (d)). These include $f_{ice}$ values of dry particles, suspended particles with ultrasonic treatment (+US), suspended particles without ultrasonic treatment (-US), and the water soluble material remaining in the filtered ash-water suspension (+US). The data for ash particles from dry generation was multiple charge corrected according to the method presented in Hartmann et al. (2016). A multiple charge correction was not possible for ash from wet particle generation because of the two particle populations (particles from soluble material and insoluble ash particles) causing overlapping signals in the UHSAS measurements. In Fig. 4, vertical error bars represent the standard deviation of three or more measurements. In case no error bar was added, the shown data point is either a mean of two measurements or originates from a single measurement.

Additionally, model calculations of three of the measured data sets (fly and bottom ash from brown coal burning, dry particle generation; fly ash from brown coal burning, wet particle generation +US) are included in Fig. 4, together with previous measurements of mineral dust (Augustin-Bauditz et al., 2014) for comparison. The calculations are based on the Soccer Ball Model (SBM) as described in Niedermeier et al. (2015). In the SBM, the ice nucleation activity of a sample is described using a contact angle distribution with $\mu$ and $\sigma$ being its mean and standard deviation. The average number of INPs per droplet $\lambda$ was determined according to Hartmann et al. (2013) after a similar approach by Vali (1971):

$$\lambda = -\ln(1 - f_{ice}^*). \tag{1}$$

with $f_{ice}^*$ being the ice fraction in the temperature range in which $f_{ice}$ saturates. The values for $\lambda$, as well as $\mu$ and $\sigma$ can be found in Table 4. All SBM fit curves are shown as thick lines in the measured temperature range and as thin lines in

**Table 4.** Parameters for the model calculations based on the SBM. $\lambda$ is the ice fraction in the saturation range, $\mu$ the mean contact angle, and $\sigma$ the standard deviation of the contact angle distribution. The clay mineral baseline corresponds to different kinds of mineral dust which featured a similar immersion freezing behavior after coating with sulfuric acid.

|  | $\lambda$ | $\mu$ (rad) | $\sigma$ (rad) |
|---|---|---|---|
| **Augustin-Bauditz et al. (2014)** | | | |
| Clay Mineral Baseline | 0.40 | 1.82 | 0.12 |
| Feldspar | 1.84 | 1.30 | 0.10 |
| **This work** | | | |
| Brown Coal Bottom Ash dry | 0.87 | 1.60 | 0.08 |
| Brown Coal Fly Ash dry | 1.91 | 1.40 | 0.07 |
| Brown Coal Fly Ash wet (+US) · 4.54 | 0.24 | 1.13 | 0.10 |

the extrapolation range. The similar values for $\sigma$ indicate that there is a comparable variability of the contact angles for all investigated ash samples and the mineral dusts.

### 3.1 Dry particle generation

Figure 4 (a) shows a significant difference between wood ashes and brown coal ashes. In case of wood ash particles, $f_{ice}$ does
not exceed 10 % between -35 °C and -37 °C. It is striking that the ice nucleation efficiency of all three examined wood ashes is very similar, which leads us to the conclusion that the influence of the burned wood type on the immersion freezing behavior of the bottom ash particles is small for the investigated samples. In this context, the amount of K in the wood ashes could play a role. Although no chemical composition analysis was performed in case of beech ash, it is known that wood ash in general contains K (Steenari et al., 1999a), which has also been found in the here examined spruce and birch ash samples (see Fig. 2 and
Table 3). Additionally, it has been shown that different wood ash samples contain similar amounts of K (Steenari et al., 1999a), justifying the assumption that the beech ash sample also contains K in amounts similar to the other two wood ashes. On the one hand, the fact that K in wood ashes is largely soluble, because it is present in the form of soluble salts and oxides (Steenari et al., 1999b), might be the reason for the low ice nucleation efficiency of the wood ashes. On the other hand, the supposedly similar amount of soluble K could be the reason for the comparable ice nucleation efficiency of the three investigated wood
ashes.

In comparison to the wood ashes, bottom and fly ashes from brown coal burning are more effective INPs in the immersion mode. For within these experiments, we can compare the onset freezing temperatures of the different samples. From these, it is apparent that the coal fly ash particles are more efficient than the coal bottom ash particles, as ice fractions above the optic's detection limit could be found at -27 °C, whereas we were not able to detect ice nucleation activity for coal bottom ash particles
until temperatures as cold as -32 °C. Again, the ice nucleation efficiency of the coal ashes could be related to the K content,

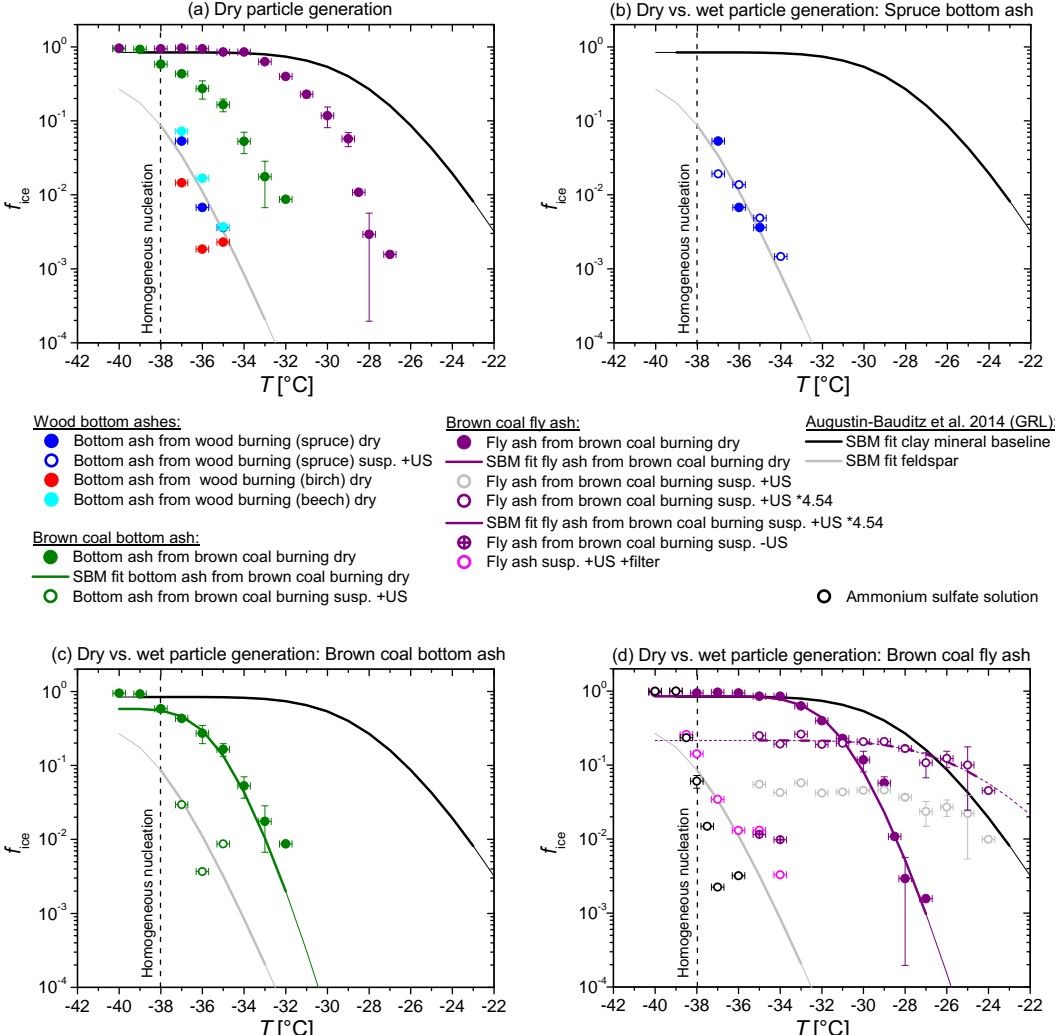

**Figure 4.** Ice fraction $f_{ice}$ as a function of temperature $T$ for ash particles from dry (full circles) and wet generation (open circles, (b), (c), (d)) with $D_{p,el}$ = 300 nm. Error bars represent the LACIS temperature error of 0.3 K and the $f_{ice}$ standard deviation of three or more measurements, respectively. A scaling factor of 4.54 was applied to the fly ash suspension measurements to account for the occurrence of crystals during wet particle generation. The solid and dashed lines represent the SBM fits to selected samples from this work and to data published in Augustin-Bauditz et al. (2014), all for $D_{p,el}$ = 300 nm. Parameters for the SBM fits are given in Table 4. The feldspar sample from Augustin-Bauditz et al. (2014) is composed of 76 % microcline (K-feldspar) and 24 % albite (Na-feldspar). The clay mineral baseline corresponds to different kinds of mineral dust which featured a similar immersion freezing behavior after coating with sulfuric acid. The homogeneous region is determined by investigating the immersion freezing behavior of highly diluted ammonium sulfate droplets (black open circles, (d)).

because, in contrast to the soluble K in wood ashes, coal ashes contain K in clay minerals with low solubility (Steenari et al., 1999b).

To date, there is no experimental evidence on the ice-nucleation-determining characteristics in ashes, presumably because it has rarely been studied and because it is a very complex mixture (Ramsden and Shibaoka, 1982; Umbría et al., 2004; Zhang et al., 2011). The chemical composition may influence the immersion freezing behavior and hence could be causing differences between the different kinds of ashes. Apart from the differences in the K content, we found that the $SiO_2$ concentration correlates with the ice nucleation efficiency of the investigated samples. As $SiO_2$ has been shown to be ice active (Pruppacher and Sänger, 1955; Isono et al., 1959; Eastwood et al., 2008; Zimmermann et al., 2008; Atkinson et al., 2013), it might be an important component influencing the immersion freezing behavior of the brown coal ashes in comparison to one another and to the wood bottom ashes. However, even though this holds true for the ashes presented here, a larger number of samples would have to be investigated to make a conclusive statement. Another relevant component could be Hg, as Mason and van den Heuvel (1959) found that $HgI_2$ has the potential to act as INP. This previous observation may be relevant to our study as Hg was detected in the brown coal ashes, but not in the wood bottom ashes.

Although studies dealing with the nature of active sites commonly investigated mineral dust particles, the findings might be transferable to other material systems such as ash. As for $SiO_2$ (Zolles et al., 2015), the brown coal ash particles might be more efficient at nucleating ice because of surface defects such as lattice dislocations caused by impurities or crystallographic dislocations. The high ice nucleation efficiency of the fly ash particles could also be related to a large fraction of amorphous material (Umo et al., 2015, found more than 80 % in a coal fly ash sample). As amorphous particles in fly ash are mainly composed of aluminosilicate glass (Ramsden and Shibaoka, 1982; Querol et al., 1996), the high content of $SiO_2$ and $Al_2O_3$ in our fly ash sample in comparison to the bottom ashes, could be an indication for a high amorphous fraction. It has been shown that amorphous organic aerosol particles are able to nucleate ice (Murray et al., 2010b; Wilson et al., 2012; Ignatius et al., 2016), but it remains to be examined whether amorphous components in fly ash are ice active as well.

To assess the ice nucleating ability of ash particles in comparison to mineral dust, Table 4 and Fig. 4 additionally show the parameters and fit curve to measurements with an untreated feldspar sample (76 % microcline (K-feldspar), and 24 % albite (Na-feldspar), Augustin-Bauditz et al., 2014), which was the most efficient dust sample ever examined at LACIS so far. Also shown is the curve for different kinds of mineral dust particles (same feldspar sample, Arizona Test Dust, NX-illite, Fluka kaolinite) coated with sulfuric acid (Augustin-Bauditz et al., 2014). The coating caused all different dusts to show a similar immersion freezing behavior even though differences were observed without coating, presumably due to different amounts of K-feldspar contained in the samples. Weathering feldspars turns them into clay minerals and it was argued in Augustin-Bauditz et al. (2014) that the coating had a comparable effect, i.e., consuming all feldspars in the different samples and leaving clay minerals only. Hence the line on which the data from all the different coated mineral dusts fell was termed the "clay mineral baseline". Comparing the dry wood bottom ash particles to the clay mineral baseline shows similarities with a tendency of wood bottom ashes being slightly less ice active than clay minerals. On the other hand, the brown coal ashes are more efficient at nucleating ice than the clay minerals, yet not as efficient as the feldspar sample. This can also be seen when comparing the mean of the contact angle distribution ($\mu$ in Table 4) determined for brown coal bottom and fly ash (1.60 and 1.40, respectively)

with those for the clay mineral baseline and the feldspar sample (1.82 and 1.30, respectively), where higher values correspond to ice nucleation activity at lower temperatures.

## 3.2 Wet particle generation

When comparing the results from dry and wet particle generation, it must be noted that a multiple charge correction was not possible for the latter due to overlapping signals in the UHSAS measurements caused by insoluble ash particles and soluble material. However, we estimated the multiple charge fractions in the suspension measurements by weighting the bipolar charge distribution (Wiedensohler, 1988) with measured size distributions. We found that the highest amount of multiply charged particles was probably present in the experiments with the fly ash suspension with ultrasonic treatment (80.5 % singly, 16.8 % doubly, and 2.7 % triply charged particles). Would we perform the multiple charge correction using these fractions, our measured data would be reduced by a maximum factor of 2 only. There is a caveat to this estimate, as crystals will probably have been present during the size distribution measurements as well and we cannot be sure how the size distributions would look like for the insoluble particles only.

In Fig. 4 (b), it can be seen that in the case of spruce bottom ash, the ultrasonic treatment and stirring process did not affect the ice nucleation ability considerably. The ice nucleation efficiency of the brown coal bottom ash suspension sample (+US, Fig. 4 (c)), however, is reduced by 20 % at -35 °C due to the change in sample preparation and particle generation.

For the filtered fly ash suspension sample, the $f_{ice}$ values (magenta open circles in Fig. 4 (d)) are comparable to what was found for similarly sized droplets containing one 300 nm ammonium sulfate particle each (black open circles), i.e., droplets grown on these particles showed solely homogeneous freezing behavior. This supports the hypothesis that the needle shaped crystals occurring during wet particle generation are likely composed of water soluble material. As a result, a solid substrate for heterogeneous ice nucleation is missing and droplets containing the material from a single crystal each, freeze due to homogeneous nucleation only. The immersion freezing behavior of the sample hence originates solely from the insoluble fly ash particles and our measurements need to be corrected with respect to the occurrence of the crystals. As mentioned in Sec. 2.2, the crystalline soluble particles were also observed in the form of hexagonal plates. However, this difference in shape does not influence the size distribution of the cloud droplets in LACIS as high supersaturations (> 10 %) ensure that every particle is activated. Hence, only the fraction of crystalline particles in comparison to the insoluble particles was determined. For this, SEM images of particles from the non-filtered fly ash suspension (+US) were analyzed. By counting ≈ 900 particles, it was determined that ≈ 78 % ± 3 % of all particles are crystals (uncertainty calculated from the 95 % confidence limit). This value may be smaller in the flow tube as the fragile crystals might break upon impact on the filter leading to a multiplication. Assuming that only 22 % of the droplets contained an insoluble fly ash particle during the experiments with the suspension sample, the original data (gray open circles) was corrected by a factor of 1/0.22 = 4.54 which is also shown (purple open circles) for a direct comparability to the ice nucleation ability of dry particles from the same sample. This correction should be seen as an upper limit for $f_{ice}$ of the fly ash suspension particles, the lower limit would correspond to the original data (gray open circles).

In the plateau region below -31 °C, a maximum of 25 % of all insoluble fly ash particles from suspension is ice active, which is a clear lowering of the ice nucleation activity by a factor of almost 4 in the plateau region compared to dry particle generation, i.e., suspending the particles in water reduced their ice nucleation efficiency in the temperature range below -31 °C. Note that this lowering might be larger depending on the multiple charge fractions in case of wet particle generation. However, fly ash suspension particles are more efficient at nucleating ice in the temperature range from -24 °C to -31 °C compared to fly ash particles from dry generation. These differences are, as for brown coal bottom ash but apparently not for spruce bottom ash, related to a change of physical and/or chemical particle properties due to the change in particle generation. A change in particle composition has been observed before for mineral dust particles which featured different hygroscopicities when being generated firstly via dry dispersion and secondly via atomization (Herich et al., 2009; Sullivan et al., 2010). It was assumed that soluble material present in a fraction of the dry particles was redistributed across all particles contained in the droplets as a coating (Herich et al., 2009). Sullivan et al. (2010) state that changes in surface structure and chemistry from dry to wet particle generation might not only affect the hygroscopicity but also the ice nucleation behavior of the particles.

Furthermore, it is interesting to see that the $f_{\text{ice}}$ values of the fly ash suspension which was not put in the ultrasonic bath are clearly lower than those of the fly ash suspension with ultrasonic treatment. Here, it is valid to compare to the uncorrected curve of the sample with ultrasonic treatment (gray open circles in Fig. 4 (d)), assuming a similar, or smaller crystal fraction in the dispersed sample without ultrasonic treatment. The observed increase in ice nucleation efficiency for the sample with ultrasonic treatment might be related to the fragmentation of large particles in the ultrasonic bath and the redistribution to smaller grain sizes associated therewith (Oorts et al., 2005). The freshly exposed surface could feature an increased number of defects serving as active sites, which would then be comparable to the increase in ice nucleation efficiency of mineral dust particles after milling (Hiranuma et al., 2014; Zolles et al., 2015). It must be mentioned, that the enhancing effect of ultrasonic treatment on the immersion freezing behavior has been observed before during experiments with soil dust at LACIS (Hellner, 2015). As there are many publications describing the use of an ultrasonic bath during sample preparation in order to avoid aggregation and for resuspension purposes (e.g., Zobrist et al., 2008; Stetzer et al., 2008; Eastwood et al., 2009; Chernoff and Bertram, 2010; O'Sullivan et al., 2015; Umo et al., 2015) we recommend cautiousness, since this practice alters particle properties in a way that may lead to larger ice fractions in the immersion mode.

### 3.3 Comparison to previous results

In the next step, the ice nucleation active surface site density $n_{\text{s}}$ was determined from the measured $f_{\text{ice}}$ according to the singular approach (DeMott, 1995; Connolly et al., 2009; Niemand et al., 2012):

$$n_{\text{s}} = -\frac{\ln(1 - f_{\text{ice}})}{A_{\text{p}}}. \tag{2}$$

For this, the particles were assumed to be spherical with a surface area of $A_{\text{p}} = 4\pi(D_{\text{p}}/2)^2$, the diameter $D_{\text{p}}$ corresponding to the electrical mobility diameter of the selected particles (300 nm). In case of fly ash from wet particle generation, the corrected $f_{\text{ice}}$, i.e., multiplied by factor 4.54, was used for the calculation of $n_{\text{s}}$. In taking the particle surface area into account

and assuming a time-independent behavior, a comparability to results obtained from experimental setups other than LACIS is made possible.

Fig. 5 shows $n_s$ for ash particles from dry and wet generation in comparison to the results published by Umo et al. (2015). Identical colors indicate similar fuels and combustion conditions. Umo et al. (2015) used a cold stage setup, where each droplet contains numerous ash particles, i.e., a large available INP surface area, leading to an increase in freezing probability in comparison to LACIS measurements. As a result, Umo et al. (2015) investigated the immersion freezing behavior of ash particles in a high temperature range ($\approx$ -12 °C to -35 °C) and there is a limited overlap in comparison to our measurements. Apart from the contrasting measurement principle, another methodological difference is the determination of the particle surface area. Umo et al. (2015) used a Brunauer-Emmet-Teller (BET, Brunauer et al., 1938) nitrogen gas adsorption method, which yields specific surface area values. A second ordinate was introduced in Fig. 5 to differentiate between our $n_s$ values assuming spherical particles (circles, left ordinate) and the BET specific surface area determined $n_s$ values reported by Umo et al. (2015) (lines, right ordinate).

All wood bottom ashes, either from dry or wet particle generation, fit the data by Umo et al. (2015) (wood: commercially available solid wood fuel, domestic: unspecified soft and hard woods) within one order of magnitude. However, for brown coal bottom ash, only the suspension particles yield $n_s$ values close to what Umo et al. (2015) found for a comparable sample, while the respective dry generated particles show $n_s$ values that are two orders of magnitude larger at the same temperature. The $n_s$ values determined from LACIS measurements with 500 nm brown coal bottom ash particles (triangles) are similar to what was found for 300 nm particles, indicating that particle properties causing ice nucleation scale with the surface area. In case of brown coal fly ash, we observe a plateau between -24 °C and -32 °C, which can also be seen in the measurements by Umo et al. (2015). However, our results are three orders of magnitude higher in $n_s$. This difference might decrease slightly, bearing in mind that $n_s$ for the fly ash suspension particles could be lower depending on the actual fraction of needle shaped crystals in the flow tube (with the deviation still being at least two orders of magnitude in $n_s$).

To make sure that this disagreement is not caused by the difference in particle surface area determination, we also calculated BET-based $n_s$ values for our measurements. For this, the particle surface area was determined according to $A_p = \rho \cdot V_p \cdot SSA$, with $\rho$ the ash density taken from the literature (wood bottom ash: 827 kg m$^{-3}$, Naik et al., 2001; brown coal bottom ash: 1415 kg m$^{-3}$, U.S. Department of Transportation, 1998; brown coal fly ash: 2456 kg m$^{-3}$, Shoumkova et al., 2005), $V_p$ the particle volume assuming a spherical shape and a diameter of 300 nm, and $SSA$ the specific surface area reported by Umo et al. (2015) from BET measurements (wood bottom ash: 6.98 m$^2$ g$^{-1}$; coal bottom ash: 8.86 m$^2$ g$^{-1}$; coal fly ash: 2.54 m$^2$ g$^{-1}$). It must be noted that the BET specific surface area is a sample specific quantity. However, as similar materials were combusted at similar conditions, we may assume that the BET values reported by Umo et al. (2015) are comparable to those of our samples. With this, $n_{s,\,BET}$ values derived from our data would even increase, in maximum by a factor of 3.5 in comparison to the $n_s$ values shown as circles in Fig. 5, meaning that effects other than the difference in surface area determination must be responsible for the discrepancy between our data set and that by Umo et al. (2015).

To estimate a possible influence of particle non-sphericity on $n_s$, one could assume a dynamic shape factor of $\chi$ = 1.25, which was observed for atmospheric dust particles (Kaaden et al., 2009) but which is likely much larger than $\chi$ for the here

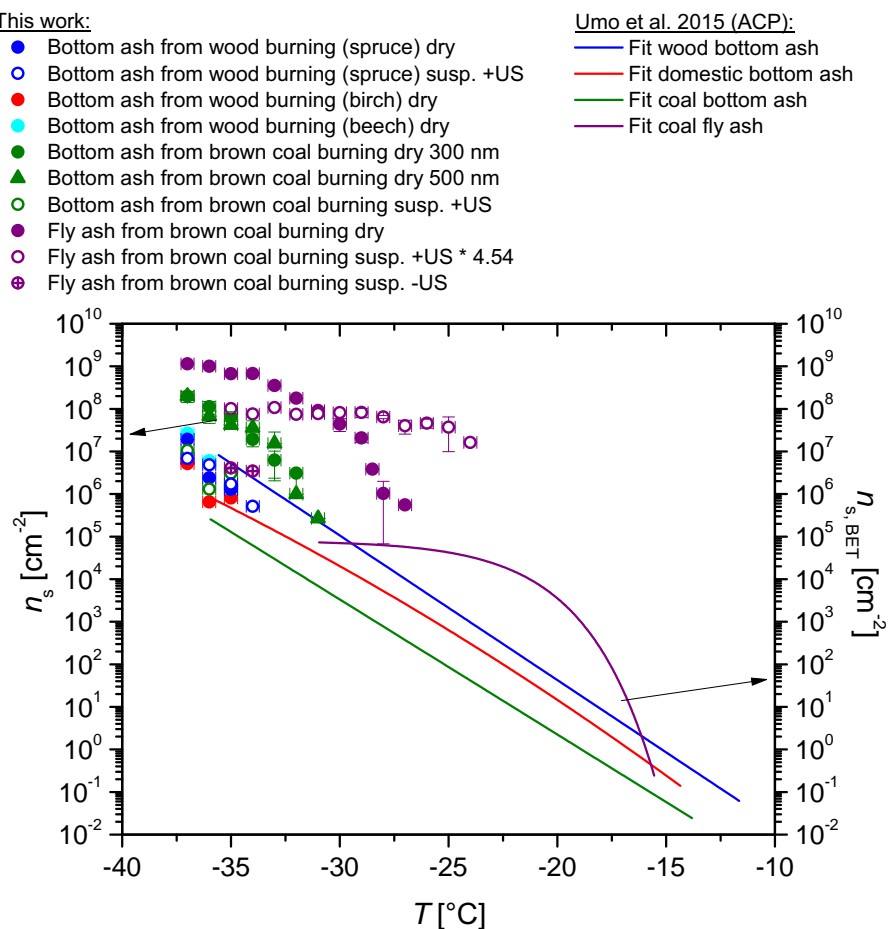

**Figure 5.** Comparison of ice nucleation surface site density $n_s$ as a function of temperature $T$ for ash particles from dry (full circles: 300 nm particles, triangles: 500 nm particles) and wet generation (open circles: 300 nm particles) with results by Umo et al. (2015, lines). For the LACIS measurements, the particle surface area was determined assuming spherical particles (left ordinate), whereas the $n_s$ values reported by Umo et al. (2015) are based on the BET specific surface area of the particles (right ordinate).

examined ash particles. This would only account for a lowering in $n_s$ by a factor of 1.4. Even a much larger $\chi$ of 1.57 as given by Hinds (1999) for sand would only increase $n_s$ by a factor of 1.9. So overall, omitting the possible effects of non-sphericity, as was done in this work, only accounts for comparably small changes in $n_s$ and cannot explain the difference between our fly ash sample and the one examined by Umo et al. (2015). Although a time-dependence of the nucleation process has been observed before (Ervens and Feingold, 2012; Welti et al., 2012; Wex et al., 2014), this effect would also be too small to describe the here found discrepancies. It is more likely, that the change in particle generation, e.g., the extended time that the ash particles spend in suspension, leads to a change of the particle properties causing ice nucleation. During this prolonged time in comparison to the few seconds from droplet activation to ice nucleation, other compounds and/or a higher amount of previously partly

dissolved compounds could enter the solution leading to a decrease in ice nucleation efficiency. E.g., Peckhaus et al. (2016) observed the dissolution of different elements from feldspar samples continuing for weeks. Furthermore, there are recent studies investigating the effect of different experimental methods on the ice nucleation behavior where most cold stage methods yielded lower $n_s$ values than dry dispersion methods (Hiranuma et al., 2015; Emersic et al., 2015). Hiranuma et al. (2015) argue that a high degree of agglomeration in the dry-dispersed particle measurements leads to a larger surface area being exposed to liquid water and consequently larger $n_s$ values in comparison to the rather de-agglomerated suspensions. On the other hand, Emersic et al. (2015) present the hypothesis that particles may coalesce in suspension leading to enhanced sedimentation and therefore a reduction of the surface area available for ice nucleation. Even though these studies investigated mineral dust particles, the findings could equally be applied to our ash samples. Differences in ice nucleation efficiency between our samples and those investigated by Umo et al. (2015) could also be related to differences in composition due to size selection. In our case, the immersion freezing behavior of 300 nm particles was investigated, whereas the suspensions examined by Umo et al. (2015) contained much larger particles (average volume-equivalent diameters of 10 $\mu$m and 8 $\mu$m for coal fly ash and bottom ashes, respectively). This might be relevant as there are studies indicating that the trace elemental composition in fly ash is inversely proportional to the particle size in the supermicron range and not strongly size dependent for submicron particles (Davison et al., 1974; Smith et al., 1979).

## 4   Atmospheric implications

There is evidence indicating that fly ash particles can be transported over several hundreds of kilometers to remote locations (Hicks and Isaksson, 2006; Li and Shao, 2009; Zhang et al., 2011; Rose et al., 2012), because, as stated earlier, industrial particle removal techniques are less efficient for small (i.e., submicron) particles. In case of wood bottom ash, also millimeter sized particles may stay airborne for hours because of their low density and convoluted shape (Andreae et al., 2004). In addition to the emission during combustion, ash may also be lofted into the atmosphere from dry ash disposal sites, i.e., during unloading off a truck and driving across the ash surface (Mueller et al., 2013).

Reliable information on the global emission of ash particles, especially biomass burning ash, is scarce which is why it is difficult to evaluate their impact on heterogeneous ice nucleation on a large scale. However, we would like to present a rough estimate on INP concentrations from fly ash emissions close to the source based on previous findings:

Emission rates of coal fly ash are strongly dependent on the efficiency of the particle removal technique in the power-plant. The horizontal and vertical plume dispersion depends on the meteorological conditions (temperature of surrounding air, stratification, wind direction and speed), as well as the exit velocity from the stacks and the stack height (Seinfeld and Pandis, 2006; Hinneburg et al., 2009). For simplicity and due to the lack of in-stack size distribution measurements downstream of the electrostatic precipitator, we assume a monodisperse population of 300 nm sized fly ash particles. In case of the Lippendorf power-plant, particulate emissions and flue gasses are released over two natural-draught cooling towers, which are 174.5 m high. On average, 32,000 t of brown coal are burned daily which, according to an estimate by Querol et al. (1996), results in the production of 6,400 t of fly ash. As the electrostatic precipitator has an efficiency of 99.98 %, 0.02 % of this mass are emitted.

This corresponds to 1.3 t of fly ash and eventually to an amount of $N_p = 4 \cdot 10^{19}$ spherical fly ash particles with a diameter of 300 nm, assuming a density of 2456 kg m$^{-3}$ (Shoumkova et al., 2005). To estimate the particle number concentration $C$ at the exit, it is necessary to assess the volume flow $Q$ from both towers and therefore the exit velocity $v_e$. This was done by evaluating video recordings of the plume rise close to the exit. On average, $v_e = 6$ m s$^{-1}$ was observed which is consistent with the mean exit velocity from natural-draught cooling towers given by Fisher (1997). $C$ can be estimated according to:

$$C \approx \frac{N_p}{Qt} \quad \text{with} \quad Q \approx \frac{N_t}{4}\pi d_t^2 v_e. \tag{3}$$

Here, the time $t$ is 24 h, as we consider the daily coal consumption, $N_t = 2$ is the number of cooling towers, and $d_t = 75$ m the cooling tower diameter. This estimation yields a particle number concentration of $\approx 8000$ cm$^{-3}$ at the cooling tower exit. From airborne size distribution measurements performed by Parungo et al. (1978a), we know the total number concentration in a stable plume 35 km and 80 km downstream of a coal-fired power-plant, which is 3510 cm$^{-3}$ and 2710 cm$^{-3}$, respectively. At the same time, the total background concentration was determined to be 1100 cm$^{-3}$. This would correspond to an atmospheric dilution factor of $\approx 3$ to $5$ for this specific set of meteorological conditions. With $n_s = 5.55 \cdot 10^5$ cm$^{-2}$ at -27 °C (fly ash from dry particle generation, taken from Fig. 5), we can calculate $f_{ice}$ for the differently sized ash particles in the plume and from that, together with the information on the size distribution taken from Parungo et al. (1978a), estimate the atmospheric INP concentrations related to the ash particles in the plume. This approximation was done assuming that $n_s$ is equal for different particle sizes and that the fly ash particles are the only component of the plume aerosol nucleating ice. Also, only particles in the submicron size range were considered. From our estimate, we found that 1.8 cm$^{-3}$ are active INPs at a temperature of -27 °C at a distance of 35 km downstream of the power-plant cooling towers, and still 1.3 cm$^{-3}$ 80 km downstream. These values are well above typical INP concentrations of $10^{-4}$ to $10^{-2}$ cm$^{-3}$ reported in the literature (Pruppacher and Klett, 1997; Rogers et al., 1998; DeMott et al., 2010; Petters and Wright, 2015). In conclusion, it can be said that the impact of fly ash particles on immersion freezing in mixed-phase clouds can be significant in close proximity to the power-plant at sufficiently low temperatures. However, to make a conclusive statement, one would have to model the horizontal and vertical dispersion of fly ash in the plume depending on the meteorological parameters and power-plant characteristics.

## 5   Summary and conclusions

In the framework of this study, the immersion freezing behavior of ash particles has been investigated in a temperature range from -24 °C to -40 °C. Airborne aerosol particles of 300 nm in size were generated from five different combustion ash samples and analyzed at the Leipzig Aerosol Cloud Interaction Simulator (LACIS). The samples included bottom ashes from spruce, birch, and beech burning, bottom ash from brown coal burning, and fly ash from brown coal burning.

It was found that there are differences in the immersion freezing behavior of dry dispersed bottom ashes from wood burning and the two brown coal ash samples, the latter showing a significantly higher ice nucleating efficiency. Bottom and fly ash

from brown coal burning initiated freezing at temperatures as high as -33 °C and -29 °C, respectively, whereas the examined wood ashes were observed to nucleate ice only below -35 °C. It was shown that the brown coal ashes are more efficient at nucleating ice than sulfuric acid coated mineral dusts (Augustin-Bauditz et al., 2014), yet not as efficient as the feldspar sample used by Augustin-Bauditz et al. (2014). The investigation of the chemical composition gave no definite indication

of what exactly causes the differences between wood and coal ashes, however, a number of different factors was discussed. Investigating further particle properties which have been shown to influence freezing behavior such as lattice structure, surface chemical configuration, number and type of surface defects, and ability of the surface to participate in electrostatic interactions (Shen et al., 1977; Yakobi-Hancock et al., 2013; Zolles et al., 2015; Kulkarni et al., 2015) might be the key to understanding the differences between different kinds of ashes.

Furthermore, measurements were conducted with ash particles from suspensions which were prepared by putting them in an ultrasonic bath followed by a 24 h stirring process. This was done for bottom ashes from spruce and brown coal burning, as well as fly ash from brown coal burning. LACIS measurements showed barely any change in the ice nucleation efficiency of bottom ash from spruce burning. On the other hand, a difference due to the change in particle generation was observed for the brown coal ashes. SEM images of the fly ash suspension particles were taken on which we observed a majority of needle

shaped crystals and some insoluble fly ash particles. The crystals may have formed from soluble components, likely $CaCO_3$, in the drying process. After a potential ice nucleation activity of the crystals was excluded, a correction of the determined $n_s$ values was performed with respect to the fraction of insoluble fly ash particles. As the results do not match the $f_{ice}$ and $n_s$ spectra of fly ash particles from dry generation, it can be concluded that the difference in immersion freezing behavior is caused by a change in particle properties from dry to wet particle generation which has been observed before for mineral

dusts (Herich et al., 2009; Sullivan et al., 2010). Eventually, another fly ash suspension sample without ultrasonic treatment was prepared for which we only observed a very low ice nucleation activity below -34 °C. This could imply that previous studies using ultrasonification might have overestimated the ice nucleation ability of certain substances. Hence, we advise to be cautious when using an ultrasonic bath for sample preparation prior to freezing experiments.

Regarding previous investigations on the immersion freezing of ash particles, we found that the wood bottom ashes, either

from wet or dry particle generation, show similar $f_{ice}$ and $n_s$ spectra in comparison to the samples examined by Umo et al. (2015). However, brown coal bottom ash only features a comparable immersion freezing behavior to the sample investigated by Umo et al. (2015) when particles are being generated from a suspension. It is striking that, although the shape of the $n_s$ spectrum of our brown coal fly ash suspension particles is similar to the one observed by Umo et al. (2015) in the overlap area, $n_s$ is higher by up to three orders of magnitude. This discrepancy cannot be explained by the difference in particle surface area

determination and therefore is most likely related to the physical and/or chemical properties of the two fly ash samples and the difference in methodology (flow tube vs. cold stage).

To summarize, we found the ice nucleation activity of brown coal bottom and fly ash to be similar to mineral dusts. However, bulk ash is a very heterogeneous material containing several different particle types (Ramsden and Shibaoka, 1982; Umbría et al., 2004; Zhang et al., 2011) which complicates the interpretation of experimental results. More work has to be done in the

field of sample characterization to identify features that cause differences in the immersion freezing behavior of different kinds

of ashes. Complementary to experimental data on the freezing behavior of ash particles, more information is needed about the atmospheric abundance of these particles. As reliable estimates of global ash emissions are missing, this knowledge can only be acquired by developing feasible methods to clearly distinguish between mineral dusts and ash particles in the framework of in situ measurements. A long-term objective is the implementation of parameterizations of ash particles as INPs into weather

5   and climate models.

*Acknowledgements.*  This research was conducted in the framework of the DFG funded Ice Nuclei research UnIT (INUIT, FOR 1525), WE 4722/1-2. We would like to thank the staff of the Lippendorf power station for providing the fly ash sample, as well as Paul Herenz and Markus Hartmann who performed and evaluated the CCNc measurements.

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
