# Peer review of "The immersion freezing behavior of ash particles from wood and brown coal burning"

_Atmospheric Chemistry and Physics, 2016_

## Referee Comment (RC1) · Anonymous Referee #3 · 22 Apr 2016

Review of "The immersion freezing behavior of ash particles from wood and brown coal burning" by Grawe et al.

The authors present the immersion freezing ability and efficiency of combustion ash particles from the laminar flow tube study. The authors provide the immersion freezing dataset in two metrics, $f_{ice}$ and $n_s$, in the temperature ($T$) range of -38 °C < $T$ < -24 °C. More specifically, this study suggests the followings: (1) the brown coal burning ash particles, which is the proxy of anthropogenic combustion ashes, are more ice nucleation (IN) active/efficient as compared to the wood burning products (the natural proxy), (2) the fly ash particles are more IN active/efficient as compared to the bottom ash particles, (3) two aerosolization processes, namely dry and wet dispersion, result in different $f_{ice}(T)$ and $n_s(T)$ spectra, (4) the ultrasonic bath application prior to particle generation increases the IN activity/efficiency of a ash sample, presumably due to the presence of fewer agglomerates in the sonicated sample than the non-treated one.

**General comments**

The topic itself is an important addition to ACP, and the authors' new IN results potentially complement the results from previous study (*Umo et al.*, 2015, ACP), in which the droplet-freezing assay was used to investigate in the immersion freezing ability and efficiency of combustion ashes in the $T$ range of -11 to -36 °C. In general, the authors conducted a careful study, with their dedicated effort to examine a variety of sample preparation methods, e.g. atomization vs. dry dispersion, ultrasonic bath application. Unfortunately, such care was not taken in the preparation of the manuscript, with the manuscript containing a number of ambiguous statements, non-intuitive figures and over-interpreted results (e.g., Sect. 4, P5 L19-30). I have numerous critical revisions as listed below. Major revisions/suggestions are listed first, followed by the section-based specific and technical revisions. I would urge the authors of the manuscript to thoroughly proof read their manuscript as this list is too long.

While authors may be able to address these issues, I do believe that the revision of the manuscript could be time consuming and result in a significantly different paper. For these reasons, I encourage the authors to resubmit it with a completely different format.

**Specific comments**

I suggest the following major revisions.

***Discuss the representativeness of 300 nm diameter ash particles*** - Justification of selecting 300 nm diameter (P3 L26-28) is missing. The authors state that the ash samples are a composite material (e.g., P15 L22), but do not discuss why 300 nm diameter is representative for their IN analyses. Different size of particles may possess different composition and/or IN behavior (e.g., Wheeler et al., 2014, J. Phys. Chem. A). I strongly suggest the authors to conduct additional IN measurements and surface characterization with different sizes. Having another set of measurements for e.g. 600 nm (or even with polydispersity aerosol population) would add clarity.

***Provide the results (no hype) to support your data interpretations and conclusions*** - One of the major findings out of this study is that the immersion behavior of brown coal ash particles changes depending on the sample preparation methods. The authors need to **investigate and discuss** in-depth physical reasons of why the observed difference appears. It seems not meaningful to attribute the reason to just the 'sample preparation and particle generation' (P11 L14-16), speculate potential reasons (e.g., P12 L1-3; P13 L10-14) and put it off as future work (e.g., P14 L28-32; P15 L20-25). To date, there have been some recent publications attempting to identify potential reasons of the data diversity due to different experimental methods and sample preparation methods (e.g., Hiranuma et al., 2015, ACP, Emersic et al.,

2015, ACP). I suggest the authors to elaborate what can be further clarified on top of given previous findings. Reporting only the IN observations seems not novel enough to complement the previous ash IN study (i.e., Umo et al., 2015).

***Consider removing Sect. 4 -*** The atmospheric implications (P14 L8-17) sound too speculative and ambitious. The abundance data (concentration and size distribution) with some spatio-temporal distributions seem necessary to estimate the ambient ash-derived INPs. The back of envelop calculation presented here seems not novel enough for you to support your sub-conclusion, which appears in P14 L13-14 ("In conclusion,…") and P1 L13-14 ("ash from brown…a regional scale"). If the authors wish to keep the atmospheric implication section, the difference between airborne (fly) ash and surface (bottom) ash with respect to their mixing state, degree of agglomeration and atmospheric lifetime should be somehow discussed. Also, discuss the contribution of natural ashes to ambient INPs vs. that of anthropogenic ones. Otherwise, the authors may consider removing the entire section.

***Tighten up the writing by removing unnecessary words -*** Improving the language, structure, presentation seems necessary. The authors should avoid making a review question how careful the research team is.

For example, I suggest minimizing ambiguous (and unnecessary) adverbs and adjectives to make the manuscript more scientifically sound than the current form. Such words should be replaced with specific and explanatory descriptions/values. Otherwise, the authors should reinforce them by adding proper citations. My suggestions include, but not limited to:

P1 L6: more (specify how much in what *T* range?)
P1 L14: at least
P2 L12: major
P2 L15: rough
P2 L20: for a long time & large
P2 L22: strongly
P2 L24: very similar
P2 L32: lower (than what?)
P3 L4: slight
P3 L23: some
P3 L24: mostly (define which samples)
P3 L30: larger & more (than what?)
P4 L30: exactly
P5 L12: most (amongst what?)
P5 L13: slightly
P5 L14: significantly
P5 L17: most striking
P5 L19: small
P5 L23: perfectly (I do not think so) & most
P5 L24: significantly less (is it fair to say this by inspecting a single snapshot picture?)
P5 L27: similar
P5 L30: small
P5 L31: small
P6 L1: supposedly & obviously

P6 L5: possibly & weakly
P8 L10: almost entirely
Fig. 4 caption: at least
P10 L2: significant & low
P10 L4: most effective & least & rather similar
P10 L6: small & more (than what?)
P10 L7: more (than what?)
P11 L2: certain (specify)
P11 L15: considerably
P11 L16: by several tens of percent (just give a number)
P11 L20: probably exclusively & completely
P12 L1-2: most likely
P12 L24: numerous
P12 L25: large
P12 L26: higher (as compared to what?)
P12 L29: larger (how much?)
P13 L12: likely
P14 L15: significantly low
P14 L24: significantly (how much?)
P15 L3: barely
P15 L4: significant
P15 L6: likely
P15 L 11: eventually
P15 L20: most likely

***Improve the figure and table presentations*** - In general, all figures and tables should be self-explanatory. My suggestions include the followings:

Fig. 1: It seems that more than 50% of mass are composed of materials that are not listed in the figure. The authors need to clarify this point to the reader by adding descriptive text or adding another group of

bars showing the sum of the other components in the figure. The authors implies such contribution may in part come from 'amorphous' material (P11 L1). I suggest the authors to give an idea of what they are (perhaps carbonaceous materials?). I also suggest the authors to give a proper reason of why beech ash composition is not shown (P5 L9). Perhaps, presenting data (± uncertainty) with the table format may be more intuitive to the reader than using the figure format. In addition, the figure caption should read "bottom ashes from spruce, birch…".

Fig. 2: I see at least six particles that have >1 µm diameter in the panel b. As stated in P5 L24-26, the doubly charged particles of 300 nm cannot be >1 µm. The authors state that "…the number of particles with three or more negative charges was negligible…" (P3 L32-P4 L2), but it seems not negligible and contradicting. In terms of the number, those large particles may have only a small contribution. However, they may substantially contribute to the total surface. If that is the case, they should be accounted for the immersion freezing parameterization. Otherwise, your ice nucleation active surface site density would be overestimated because of overlooking the presence of large particles. Adequate surface estimation is one of the important keys for the $n_s$ parameterization. Ultimately, the authors may want to assess if such correction can explain the discrepancy between dry and wet (or not).

Just to start with, you may first estimate the particle size distributions by analyzing SEM images (i.e., estimate the area equivalent diameter for several hundreds of particles; see Hoffmann et al., 2013, AMT). This approach may be better than relying only on the USHAS measurements. In addition, such work will help clarify the vague statement in P5 L24-30.

Fig. 4: The authors need to explain the factor of 4.54 in text at the first appearance of Fig. 4. I also suggest clarifying what the 'clay mineral baseline' means. Better presentation may be made with multiple panels. For example, the authors may use individual panels for ash type comparison, dry vs. wet, +US vs. –US. The same may apply to Fig. 5.

Fig. 5: Specify if your $n_s$ metric is based on the BET specific surface area or the geometric surface. Though I agree with your statement in P13 L8-10 (i.e., the influence/difference is small anyway), it is important to compare the results with a same metric. If this figure contains both BET-based $n_s$ and geometric $n_s$ spectra, I suggest presenting all data and spectra using either one of two metrics. The authors may simply apply a factor of 4 (P13 L9) for the conversion.

***Add proper references*** – I suggest the authors to give a careful look at the followings:
P1 18: Add citation after "…and climate models".
P1 L21: Replace Murray et al., 2012 with Koop et al., 2000 (Nature); Murray et al., 2010 (Phys. Chem. Chem. Phys.); Rosenfeld and Woodley, 2000 (Nature).
P2 L5: Add citations after "…been conducted".
P3 L1-2: Whale et al. (2015, AMT) may be a good reference to add here.
P3 L20: Add reference for your dry dispersion method.
P4 L8: Add reference for "water contamination". The authors may also want to reduce the tone and call it as a background contribution or something similar.
P7-8: Provide the reference for R1 & R2.
P10 L17: Any reference for insoluble K to be highly ice active? You should not speculate for all insoluble materials to be IN active.
P12 L1-3: multiple citations seem missing.
P12 L8: Look into references given in Zolles et al. (2015). There have been a few other studies of active sites and their influence on IN activities.
P12 L16: e.g., Connolly et al., 2009 (ACP), Niemand et al., 2012 (JAS)

Other specific and technical suggestions sorted out for each section are listed below.

***Abstract***
P1 L2: …trigger ice nucleation when they interact with water vapor and/or supercooled droplets.
P1 L4: …worldwide, and…
P1 L6-7: I suggest separating this sentence into two sentences.
P1 L6: ice active in the immersion mode
P1 L8: …the effect of various particle generation methods on…
P1 L8: For this → For instance
P1 L14: heterogeneous ice nucleation → immersion freezing in the *T* range of XX to YY °C

***Sect. 1***
P1 L21-23: The authors may focus on immersion freezing by rephrasing L21-23 to, "At water saturation, this temperature limit of droplet freezing … referred to as immersion freezing.". Otherwise, describe heterogeneous IN and then immersion freezing. Note that the deposition nucleation can occur at temperatures below -38 °C.
P2 L4: However, up to now only a very few → To date, only a few
P2 L5: → material, which
P2 L12: contribute to
P2 L12: → Furthermore, the ash from natural source is…
P2 L13-14: Awkward sentence. Rephrase.
P2 L18: I wonder where the authors find the 7% number in DeMott et al. (2003).
P2 L19-20: I suggest clarifying that the result presented in DeMott et al. (2003) is from a limited time segment of measurements in cirrus clouds.
P2 L23-26: I do not understand what this sentence mean. Please clarify.
P2 L27-29: I suggest separating this part into two sentences.
P2 L33: What are water soluble components? Sugars? Any biological materials? How important are they for IN as compared to the insoluble components?
P2 L35-P3 L2: Provide more information regarding Umo et al. (2015). At least the investigated *T* ranges for individual ash materials and their IN efficiency/activity with some quantities.
P3 L2: Reword "however". I do not find smooth transition.
P3 L5: Reword "assumed". e.g., suggested/postulated
P3 L11-12: Move this part to the method section.

***Sect. 2.1.1***
P3 L21: as → along with
P3 L21-23: Subdivide the sentence ("Variations caused … transported downwards.") into two.
P3 L23-24: Clarify if glass beads have any influence on modification of particle surface (e.g., by scratching) and IN behavior.
P3 L24-26: I suggest rephrasing the complete sentence ("In addition … dryer unit."). Just say you used a custom-built atomizer. Is it perhaps the one used in Wex et al., 2015, ACP? If so, add it as a reference.
P3 L31: efficient → active due to the presence of large surface

***Sect. 2.1.2***
P4 L4-5: Awkward sentence. Rephrase. The LACIS reference is repetitive.
P4 L6: "LACIS offers … nucleation processes."; please provide the reference showing the 'surface' (of the droplet assay plate - I assume this is what you mean) interferes with IN measurements. If not, I suggest rephrasing the sentence.
P4 L14-15: I suggest briefly clarifying how the phase discrimination can be done in text.

*Sect. 2.2*

P4 L18-23: I suggest to summarize your ash samples using a table along with columns of ash type (bottom or fly), source (natural or anthropogenic), particle generation method (dry or wet), and information regarding your IN experiments (e.g., examined *T* range and IN observed *T* range).

P4 L25-26: "It has to be noted that…"; not much adding. I suggest removing this sentence.

P4 L27: If it is commercially available, specify it.

P4 L27: Explain how you "extracted" it. Provide some information regarding the electrostatic precipitator (or reference). What is the cutsize? What is the collection efficiency? etc.

P4 L28: → station, which

P5 L4: affected → modified

P5 L5: "one fly ash suspension sample"; specify which one and why the authors choose this one in text.

P5 L9: Provide brief description of atomic adsorption methods along with proper reference(s). The rest of the paragraph (up to L18) may better fit in the results section. Consider reorganizing the sections.

P5 L11: obtained → estimated

P5 L19-30: I suggest the authors to minimize over-interpretation from snap shot pictures. The authors may consider providing the reader with the proxy of particles' sphericity. For example, a particle aspect ratio of individual particles can be estimated by the 2D SEM images. Inspecting several hundreds of particles and providing statistically valid data in the result section will strengthen the paper.

P5 L23: Irregular shape may suggest the predominance of carbonaceous compounds (e.g., Hiranuma et al., 2008, Atmos. Environ.). Do the authors have any information regarding its organic fraction and content?

P5 L23-24: "…most show significantly less surface defects…"; it is hard to judge the presence of defects by looking at given SEM pictures with low magnification. Can the authors provide the image with high magnification as an example? Better option would be measuring BET surface of both bulk samples and comparing each other.

P5 L25-26: I wonder why UHSAS was not able to measure them.

P5 L27-28: "…similar particle losses should have occurred."; I suggest removing any opinion statements. The authors may provide the results instead if available.

P5 L33-35: "It is reasonable to assume … additional fly ash particles."; I do not understand your logic here. Please clarify.

P6 L3: → all components are dissolved in droplets

P6 L5 to the end of this section: this part seems not belonging to the method section. Should be moved to the discussion section?

P8 L9-10: Awkward sentences. I suggest rewording.

*Sect. 3*

P8 L13-16: Subdivide this sentence to two parts.

P8 L17-19: Why don't you look at only quasi-spherical ash particles on the SEM images? Analyzing the SEM images, you should be able to distinguish ash particles from the recrystallized components, which seem possessing high aspect ratios. This procedure perhaps enables you to estimate the contributions of multiple charge components. The same applies to your statement in P11 L12-13. The correction seems feasible and important.

P8 L21: → in Fig. 4. These calculations are based on…

*Sect. 3.1*

P10 L3: Specify the *T* range for the reader.

P10 L4-6: Discuss why all the wood burned ash particles have similar IN behavior. There seems difference in $SiO_2$ content for spruce (~10%) and birch (<5%). Based on your statement in P5 L12-13 (i.e., the higher $SiO_2$, the more IN efficient), could the $SiO_2$ content be responsible for the observed result of $n_{s\_spruce} > n_{s\_birch}$ in Fig. 5? If this holds true, the bulk beech ash should contain the highest $SiO_2$. Would you consider carrying out the atomic adsorption measurement of beech ash (currently missing without any intuitive explanation) to support your idea?

P10 L9 - P11 L3: The authors suggest that the fraction of $SiO_2$, Hg and insoluble K has some contributions to determine the immersion freezing behavior of ashes based on their bulk observation of atomic adsorption technique (P10 L9-22). In the following sentences, the authors introduce the hypothesis, which infers that the amorphous material content is also important as a determining factor of IN behavior. Discuss which one is more dominant. Further, what is the mixing state of your ash samples? What is the relative importance of the mixing state as compared to the "bulk" composition?

P10 L10-12: Rephrase. Not easy to follow.

P10 L14: → act as INP. This previous observation may be relevant to our study as…

P10 L14: Is the atomic adsorption technique used in this study sensitive to iodine? If so, please show I in Fig. 1 too.

P10 L16: Why soluble? Presumed? Or measured? Please clarify.

P10 L18-22: "Apart from the chemical composition, other properties such as ice active sites…"; Discuss the relative importance of what you found to the other potential factors. I suggest rephrasing and clarifying the last sentence. I do not understand what the authors mean here.

*Sect. 3.2*

P11 L12-13: I disagree. The correction is possible and feasible, and the authors should account for it. See my comment in Sect. 3.

P11 L13-14: This sentence seems not needed here. I suggest removing.

P11 L14-16: The authors need to **investigate and discuss** physical reasons of why the influence of the US application on IN behavior is **material dependent** rather than just saying 'sample preparation' methods resulted in the difference. The same goes to P13 L3-4.

P11 L18: meaning → suggesting

P11 L17-30: Multiple topics seem to be squashed within this short section. Reorganize and rephrase the sentences. Again, the scaling factor of 4.54 should be introduced prior to its first appearance in Fig. 4.

P12 L1-3: Sounds too speculative. Any experimental evidence of a destruction of former active sites? I do not understand the last sentence. How do water soluble components play a role in what?

P12 L8-13: What is the atmospheric implication of your findings here?

P12 L23-24: Repetitive. Delete.

P12 L24-30: Rephrase. This part may better fit in the introduction section as part of an example of previous work regarding ash IN characterization.

*Sect. 3.3*

P12 L15: ice nucleation active surface site density

P13 L5: nucleation spectrum → $n_s$ spectrum

P13 L10-14: Sounds too speculative. I suggest removing the opinion statements (that is, must be, expect, likely) and rephrasing the sentences.

*Sect. 4*

P14 L2-4: The authors may consider removing this paragraph, which seems not adding much.

P14 L5-8: These sentences better fit in the introduction section (e.g., P2 L15) rather than here.

*Sect. 5*

P14 L21: at the

P14 L23: differences (in what?)

P14 L28: "a decisive factor…"; I disagree. The authors need to provide much more comprehensive composition data (e.g., beech ash data, bulk vs. single particle, mixing state, organic speciation) than what are presented in this manuscript to have this statement as your conclusion.

P14 L28-32 & P15 L20-21: I agree with the authors that multiple particle properties inherently influence particle's immersion freezing behavior, and it is not easy to understand what the controlling(s) factor

is(are). With some extra experiments suggested above, the authors may be able to shed light on the questions raised here.

P15 L9: nucleation spectrum $\rightarrow$ $f_{ice}$ and $n_s$ spectra; the same applies to elsewhere, e.g., P15 L17

P15 L12-13: What is its atmospheric implication?

---

## Referee Comment (RC2) · Anonymous Referee #1 · 25 Apr 2016

**Review of "The immersion freezing behavior of ash particles from wood and brown coal burning" by Grawe et al.**

This manuscript presents an interesting data set regarding the ice nucleating abilities of ash particles which are currently poorly understood. Given the lack of information on this topic, and the potential that ash particles may have to influence mixed-phase cloud formation, the present results are a valuable contribution to the ice nucleation community. Although the present manuscript possesses many similarities with the Umo et al. (2015) study, I still see a small level of novelty on it. I got attracted by the title of the paper which indicates that the ice nucleating abilities of wood and brown coal burning ash particles were studied, but I am disappointed that a clear explanation of why brown coal ash particles are better INP than the wood ash particles is not provided. Additionally, the conclusions are not clearly supported given the lack of some key experiments. Therefore, I think that this paper could be accepted in ACP only after the following points are clearly addressed. Note that this review was prepared without reading the comments given by referee #3; therefore, I apologize for any overlap between the two reviews.

**Major comments:**

1. Multiple charge correction was applied to the dry samples only. Which percentages of the particles were multiple charged? Based on what data was this correction conducted as the authors indicate that the UHSAS did not clearly detect the multiple charged particles? How good is the agreement between the SEM and the UHSAS at detecting multiple charged particles? I encourage the authors to report the size distribution for each sample and the resulted size distributions after size selection (300 nm).

2. There is a poor consistency in the experiments conducted with the 5 ash samples as shown in the table below. I am wondering why there is too much data missing. This lack of information reduces the robustness of the drawn conclusions regarding the particle generation methods and the effect of treating the samples with ultrasound. I suggest conducting more experiments to fill out the table below.

3. Is it the needle formation exclusive to fly-ash brown coal particles? Why SEM images of the bottom brown coal (suspension) are not presented? Why the SEM analysis was not applied to the wood ash samples? The authors indicate that the needle formation in the fly-ash brown coal particles may be cause by $CaCO_3$ which was formed by the presence of CaO detected by the atomic adsorption (AA) analysis. However, Figure 1 shows that the levels of CaO in the wood samples are much higher than in the brown coal samples. Therefore, it would be nice to see the SEM images of e.g. Spruce which has the highest concentration of CaO.

4. I am not fully convinced that the particles produced through the wet system are less efficient. The authors conducted a direct comparison of the ice nucleating abilities of the wet and dry generated particles; however, it is necessary to demonstrated that the monodispersity of the 300 nm particles from both system is comparable.

| | Material | Fly/Bottom | Dry Generation | Wet Generation | Ultrasound | Without ultrasound | Filtered | SEM | AA |
|---|---|---|---|---|---|---|---|---|---|
| Wood | Spruce | B | X | X | X | | | | X |
| | Birch | B | X | | | | | | X |
| | Beech | B | X | | | | | | |
| Coal | Brown coal | B | X | X | X | | | X | X |
| | Brown coal | F | X | X | X | X | X | X | X |

**Specific comments:**

- Page 1, line 20: The Hallett-Mossop (1974) study introduces one of the multiple secondary ice formation mechanisms only. I suggest using a better reference such as Heymsfield and Willis (2014): Heymsfield, A. J., and Willis, P. (2014), Cloud conditions favoring secondary ice particle production in tropical maritime convection. *J. Atmos. Sci.,* 71, 4500–4526, doi:10.1175/JAS-D-14-0093.1.
- Page 1, line 21: I suggest replacing this reference with a more appropriate one.
- Page 2, line3: provide pages from P&K or the actual paper(s).
- Page 2, line 5: Cite the studies here.
- Page 2, line 12: "coal ashes contribute a major proportion of anthropogenic aerosol emissions" please provide the reference and number (estimate).
- Page 2, line 15: "yields global annual emissions of 30 Mt" is this comparable to mineral dust? Please provide the mineral dust annual emissions.
- Page 2, line 17: delete "however".
- Page 2, line 27: This needs to be better organized. The authors jump between old and recent studies back and forward.
- Page 2, line 32-34: "it could be shown that water soluble components were responsible for differences in the ice nucleation ability of fly ash samples from different power plants." Provide reference here.
- Page 3, line 5: "Umo et al. (2015) assumed that the different" Did they assume or did they provide evidence of?
- Pages 3-4, lines 32-1: "the number of particles with three or more negative charges was negligible". Please report the fractions or percentages of the multiple charged particles.
- Page 5, line 7: why 200 nm? What did not the authors try a smaller size to ensure that insoluble particles are not present? Please provide the resulting size distributions of the atomized filtered solution.
- Page 5, line 9: Why was not beech bottom ash particles analyzed by AA?
- Page 5: What are the uncertainties associated with the data presented in Figure 1?
- Page 5, lines 24-30: I am wondering how consistent and how reproducible is the data obtained from the dry generation system. Can the authors provide the size distributions obtained with the dry system separated by 15 or 30 minutes? Was it the multiple charged particles checked continuously with the UHSAS?
- Page 5, lines 33-34: Can the authors provide an activation scan with the filtered solution to confirm this?

- Page 8, lines 8-10: Which fraction of particles passed through? Was this confirmed with the size distributions from the UHSAS?
- Page 8, lines 15-16: Why the data from the water soluble material remaining in the filtered ash-water suspension is not shown?
- Page 10, lines 3-4: "There is a trend of beech bottom ash being the most effective". I am wondering why out of the three wood samples beech was the least studied (although it was the most effective at nucleating ice)?.
- Page 10, lines 15-18: "The fact that the wood ashes contain significantly more K than the brown coal ashes, which in this case is soluble, is a possible reason for the lower ice activity in comparison to the brown coal ashes. According to this, insoluble K could be the decisive element determining the freezing behavior of the brown coal Ashes" What about CaO?
- Page 10, lines 20-22: "the brown coal ash particles might be more efficient at nucleating ice because of surface defects such as lattice dislocations caused by impurities or crystallographic dislocations." This is not clearly supported by the presented data. I am wondering why the authors did not provide SEM images for the bottom wood ashes to compared the surface defects with those of the bottom coal ash particles.
- Page 11, Lines 1-3: "It has been shown that certain types of amorphous particles are able to nucleate ice (Murray et al., 2010b; Wilson et al., 2012), but it remains to be examined whether the amorphous components in fly ash are ice active as well". This only happens at very low temperatures relevant to cirrus clouds.
- Page 11, lines 4-6: "Tab. 1 and Fig. 4 additionally show the 5 parameters and fit curves to measurements with K-feldspar and mineral dust particles (K-feldspar, Arizona Test Dust, NXillite, Fluka kaolinite) coated with sulphuric acid (clay mineral baseline, Augustin-Bauditz et al., 2014). I may have missed but I could not find the Arizona Test Dust, NXillite, and fluka kaolinite data.
- Page 11, lines 26-30: "By counting ~ 900 particles on SEM images, it was determined that ~ 78 % of all particles are crystals. This value may be smaller in the flow tube as the fragile crystals could break upon impact on the filter leading to a multiplication. As only 22 % of the droplets contained a spherical fly ash particle during the experiments with the suspension sample (+US), the original data was corrected by a factor of 1/0.22 = 4.54 which is also shown in Fig. 4 for a direct comparability to the ice nucleation ability of dry particles from the same sample." How confident are the authors about this calculation? What is the uncertainty associate to it? 900±?? 78%±??, 22%±??, 4.54±??
- Page 11, lines32-35: "which is a clear lowering of the ice nucleation activity by a factor of 4 compared to dry particle generation, i.e., suspending the particles in water reduced their ice activity in the temperature range below -31 °C." I am not sure how valid is to directly compared the ice nucleating abilities of the ash particles generated from the wet and dry systems given that those obtained from the wet generation are not corrected for multiple charged particles.
- Page 12, lines 4-13: This is an interesting observation. I am wondering why the authors did not further expand this with other samples (e.g., bottom ash brown coal).
- Page 12, lines 4-5: "the fice values of the fly ash suspension which was not put in the ultrasonic bath are clearly lower than those 5 of the fly ash suspension with ultrasonic treatment." Here and along the results section, how many scans were conducted for each sample for each specific set of conditions? How reproducible are they?

- Page 14, lines 1-14: This part is too speculative with many unsupported assumptions. This should be deleted.
- Figures. Be consistent with the labels (a and b, or 1 and 2)
- Figure 1. I am not sure how useful is panel 2.
- Figure 3: It would be nice to have a similar image of the wet generation system of a bottom ash sample. Additionally, Fig. 1 indicates that Wood ashes contain much more CaO than the coal ashes. Therefore, the needle crystals should be more pronounced in the wood ashes if the reasoning presented here is correct.

---

## Referee Comment (RC3) · Anonymous Referee #4 · 26 Apr 2016

Comments have been uploaded in the attached pdf

Please also note the supplement to this comment:
http://www.atmos-chem-phys-discuss.net/acp-2016-208/acp-2016-208-RC3-supplement.pdf

[Figure]

Review of "The immersion freezing behavior of ash particles from wood and brown coal burning" by Grawe et al.

In this study the authors examine the immersion mode freezing efficiency of combustion ashes from different woods and brown coal burning using LACIS. Ashes from brown coal burning are found to exhibit higher nucleating abilities than those from the ashes generated from wood burning. The results presented here also seem to indicate that sample preparation can have impacts on ice nucleation efficiencies; an important point which will need to be considered in future studies.

My major comments below surround increasing the specificity and clarity of statements made. In particular, there are some vague statements made in attempting to account for observations in this work, and how it compares to others such as Umo (2015). Sentence and paragraph structure can also at times make it difficult to make out what is being said without multiple rereads of certain passages. While I recognize that investigating the nature of nucleating sites in a mixture as complex as ash is challenging, I suggest the authors could discuss the difficulties surrounding this endeavour and limit sweeping statements.

I am of the opinion that, following careful consideration of the points below and improvement of the manuscript in the areas listed, this could be accepted in ACP.

Main comments:

- The authors need to evaluate the use of 300 nm particles in this paper. It can be anticipated that physical and chemical composition varies with particle size, and in turn, perhaps the ice nucleating efficiencies. Without further experiments on larger particles it is difficult to see how the results of this study can be generalised to larger particles. I suggest that at the points in the manuscript where the authors are referring to size selected particles, they explicitly state this for clarity (e.g. section 3.1 L1-8).
- P2L12: "As a result, coal ashes contribute a major proportion.....". This strong statement needs a reference.
- P2L13-15: On the one hand, "the importance of ash particles as potential INPs must be put into perspective by comparing with concentrations" yet in the next sentence an emission rate is given, not a concentration. While both sentences on their own are fine, having these sentences one after the other could be misleading.
- P4L6: Why is this important? Reference to publications demonstrating that surfaces typically used for ice nucleation assays interfere with the nucleation process would seem to be appropriate here, bearing in mind that many studies have been performed where droplets are supported by substrates and this does not appear to be an issue.
- P4L7-8: This statement also needs referencing and elaboration: what size droplets are you speaking of? Homogeneous nucleation CAN be probed in picolitre sized droplets on cold stages.
- P4L24: only three woods are investigated in this study; is this representative of all "deciduous vs. coniferous" trees?

**Fig. 1.**

**Supplement:**

Review of "The immersion freezing behavior of ash particles from wood and brown coal burning" by Grawe et al.

In this study the authors examine the immersion mode freezing efficiency of combustion ashes from different woods and brown coal burning using LACIS. Ashes from brown coal burning are found to exhibit higher nucleating abilities than those from the ashes generated from wood burning. The results presented here also seem to indicate that sample preparation can have impacts on ice nucleation efficiencies; an important point which will need to be considered in future studies.

My major comments below surround increasing the specificity and clarity of statements made. In particular, there are some vague statements made in attempting to account for observations in this work, and how it compares to others such as Umo (2015). Sentence and paragraph structure can also at times make it difficult to make out what is being said without multiple rereads of certain passages. While I recognize that investigating the nature of nucleating sites in a mixture as complex as ash is challenging, I suggest the authors could discuss the difficulties surrounding this endeavour and limit sweeping statements.

I am of the opinion that, following careful consideration of the points below and improvement of the manuscript in the areas listed, this could be accepted in ACP.

Main comments:

- The authors need to evaluate the use of 300 nm particles in this paper. It can be anticipated that physical and chemical composition varies with particle size, and in turn, perhaps the ice nucleating efficiencies. Without further experiments on larger particles it is difficult to see how the results of this study can be generalised to larger particles. I suggest that at the points in the manuscript where the authors are referring to size selected particles, they explicitly state this for clarity (e.g. section 3.1 L1-8).
- P2L12: "As a result, coal ashes contribute a major proportion.....". This strong statement needs a reference.
- P2L13-15: On the one hand, "the importance of ash particles as potential INPs must be put into perspective by comparing with concentrations" yet in the next sentence an emission rate is given, not a concentration. While both sentences on their own are fine, having these sentences one after the other could be misleading.
- P4L6: Why is this important? Reference to publications demonstrating that surfaces typically used for ice nucleation assays interfere with the nucleation process would seem to be appropriate here, bearing in mind that many studies have been performed where droplets are supported by substrates and this does not appear to be an issue.
- P4L7-8: This statement also needs referencing and elaboration: what size droplets are you speaking of? Homogeneous nucleation CAN be probed in picolitre sized droplets on cold stages.
- P4L24: only three woods are investigated in this study; is this representative of all "deciduous vs. coniferous" trees?

- P5L1-2: What power does this sonicator deliver to the sample?
- P5 L6-7: is there any effects based on how long the samples are left to stir for? Also, what about aggregation in solution? Could you be removing particles smaller than 200 nm by filtration that have formed aggregates over the course of 24 hours of stirring?
- P5L6: Define "ice activity"
- P5L8: Can the authors say anything on the efficiency of these filters and pore size distribution? Can particles larger than 200 nm sometimes make it through? Also, are the authors defining "water soluble material" as any particle which will make it through a 200 nm filter? This should be stated if so.
- Figure 4: How is the "homogenous" region here determined? Citation of the appropriate paramaterization(s) should be added.
- Fig 4: State in the legend and text exactly what the "clay mineral baseline" is; i.e. what clays were used?
- Fig 4: Harrison et. al. ACPD (2016) show that K-Feldspar can have different activities: specify exactly which K-Feldspar was used.
- P10L7-8: I don't have an issue with this statement as such, but I think it's important to point out that onset freezing temperatures cannot be used for others to compare with. A simple phrase along the lines of "For within these experiments we can compare our onset freezing temperatures, which…."
- P10L9-13. This section is vague, and should be strongly caveated by noting that the authors don't know what the explanation for the differing nucleation activities is, or further experimental evidence showing that SiO2 is indeed responsible for this behaviour should be added. For both this point, and the point below, a statement explaining why it is difficult to conclusively identify the active site in such a complex sample at the beginning would definitely help here!
- Page 10L 14- Page 11L3: All of this section is very speculative and at points vague, and I suggest it be rewritten to make it more concise or removed. After providing several possible explanations for their observations, the authors note "otheir properties such as ice active sites on the particle surface might affect the ice nucleation ability of a substance." What does this mean? Taking the Vali (2015) definition of a site being a "Preferred location for ice nucleation on an INP", this statement is rather meaningless.
- Page 11 L5: again, which K-Feldspar was used?
- Page 11 L6: Is it reasonable that clay minerals treated with sulfuric acid are termed the "clay mineral baseline"? Surely this should be justified. Also, if justified, what it is should be stated explicitly in figure 4.
- Page 11 L17-page 12L3: There are a lot of concepts in this rather long paragraph: separating into at least two smaller paragraphs would help readability greatly
- P12L1-3: If it's not the physical or chemical particle properties changing and leading to a destruction of active sites, what else could it be? This non-specific nature of this statement makes it vacuous. The word "likely" doesn't help here either.

- Section 3.3: Discussion of possible differences in composition and hence ice nucleating abilities due to size selection of particles should be presented here.
- Page 13L12: Agreed, but explain why you would not expect it. Reference where appropriate
- P13L13-14: Again, vague. Is there nothing more which can be said here other than the samples were not "completely identical"
- P14 L26: All clay minerals? Which K-Feldspar?

Typos/other:

P3L1: Is the cold stage(s) used by Umo (2015) the same as that used in Murray 2010(a)?

P1L3: Largest rather than biggest

P8L26 and fig 4. SMB should be SBM no?

---

## Referee Comment (RC4) · Anonymous Referee #2 · 27 Apr 2016

**Referee comments for manuscript acp-2016-208 by Grawe et al. "The immersion freezing behavior of ash particles from wood and brown coal burning"**

**Summary:**

The manuscript presents results obtained at the Leipzig Aerosol Cloud Interaction Simulator, LACIS, on the ice nucleation behavior of ash particles from wood and coal combustion. The chemical composition of the different ash samples was obtained from atomic absorption methods. The main findings were that fly ash from coal burning was the most ice-active sample and was the sample with the highest fraction of $SiO_2$. Furthermore, the effect of differences in the particle generation method (dry vs. wet, treating the sample in an ultrasonic bath vs. not using it) was investigated. It was found that wet generation reduces the ice nucleation ability of the ash particles compared to the dry generation method. Particles treated in an ultrasonic bath had a higher ice nucleation ability than non-treated wet generated particles.

Ice nucleation induced by ash particles has received only little attention by the scientific community. The presented measurements are therefore of interest to the readers of *Atmospheric Chemistry and Physics*. Also the investigation of the influence of the generation method on the results is interesting. However, the manuscript and particularly the discussion of the results is often written in a vague and imprecise way and hardly adds to our understanding of the underlying ice nucleation process. I thus only recommend publication after addressing the comments below.

I apologize for any overlap with the other reviewers' comments.

**Major Comments:**

*Introduction:*

- The literature summary on p.2 l.27ff lacks depth. *Who* measured *what, at which conditions*? What type of bottom ash did Murray et al. (2010a) investigate? Regarding the first lines of page 3 it is not clear to me which results were found by Umo et al. (2015) and which by Murray et al (2010a).
- Since the study is very similar to that by Umo et al. (2015), the earlier study should be described in more detail and the novelty of the present study should be described. What sizes did Umo et al. (2015) investigate? Which surface area? Already describe the freezing method here. And convince the reader why there is a need for your study.
- One could also add motivation from field observations of biomass burning, e.g.: Prenni et al. 2012, GRL; McCluskey et al. 2014, JGR

*Methods:*

- The representativeness of 300 nm particles is not shown or even mentioned. Why were particles of this size selected? How do the size distributions of the investigated ashes look like? Is 300 nm the mode diameter or at least close to it? Can it be assumed that larger/smaller particles have a similar composition and is the ice nucleation ability expressed as ice-active surface site density thus potentially scalable or are changes to be expected with size?
- The description of the wood samples is very vague. What was burned? Just the wood or in addition bark, leaves, needles, beechnuts…? Which part of the wood? How old was the wood (i.e. freshly cut, stored for a long term). What type of spruce, birch, and beech? Have the wood samples been treated in any way before burning? Are these commercially available fuels?
- The disagreement between the measurements with the UHSAS and the observations of multiple charged particles on the SEM filters is disturbing. I am glad this is discussed in the manuscript but I encourage the authors to put more effort in finding the reasons for this discrepancy. That the filter analysis is not representative appears as an oversimplified argument. If this is the case then the analysis should be done in a way that it is representative, e.g. analyze a larger area of the filter. How representative are other conclusions drawn from the filters?

*Results and Discussion:*

- p.13 l.5 ff: It is not clear to me why you state that "the shape of the nucleation spectrum of fly ash suspension particles from our measurements correspond well to the findings by Umo et al. (2015),…". The fly ash data points in Figure 5 to me look either mostly linear (open purple circles), consist only of 2 data points (open purple circles with cross) or have a plateau which starts at a temperature 8 K lower than the fit from Umo et al. (2015) (full purple circles). Keeping in mind that for a given particle size and concentration, $n_s$ can only vary in a restricted range of 3-4 orders of magnitude in this case, the observed three orders of magnitude differences are massive.
- The correction factor $4.54 = 1/0.22 = 1/(1-0.78)$ is derived from the SEM images of the filters. It is already stated on p.11 l.27 that this value might be smaller in the flow tube due to less fractioning of the crystals. However, in the discussion of the results it is not considered as an upper limit but rather as a realistic estimate. Especially for the comparison of $n_s$ to Umo et al. (2015) it would be helpful to be aware that this value might be considerably smaller.

*Atmospheric Implications:*

I would recommend deleting this section completely as the calculation is not convincing nor adds value to the manuscript. To my understanding the dilution factor of 1000 over a distance of 80 km derived from Parungo et al. (1978a) only applies to very particulate atmospheric conditions and does not seem representative for general atmospheric situations. It seems to me that the background aerosol concentration has not been considered in the derivation of the dilution factor.

Furthermore, are the emissions of coal-fired power plants today the same as in the 70's? Without having discussed how representative 300 nm particles are for ash particles in general and how likely the composition and morphology and thus $n_s$ of 300 nm and 1000 nm particles are similar the given estimate is not convincing. A short comment on the atmospheric implication could be included in the summary part. However, I would suggest referring here to direct field observations of ash particles as ice nucleating particle or ice crystal residual.

*Language:*

It seems to me that there is an overuse of "could". It you are talking about your findings then the use of phrases like "was shown" or "was confirmed" instead of "it could be shown", "could be confirmed",… is more appropriate.

**Specific Remarks:**

p. 1 l.6: "lignite" is only mentioned in the abstract. If it is important to know it should be mentioned in the main paper body, if not it can be deleted.

p. 2 l. 3: Is "Pruppacher and Klett, 1997" truly the primary source? Give the original reference otherwise.

p. 2 l. 24: What is the chemical composition of mineral dust and ash that is very similar? Be more precise.

p.3 l.5ff: What other factors could determine the found differences between the two ashes? I can't think of any other factors, assuming same sized particles were used. Thus this sentence does not yield new information.

p.4 l.13 "second to last section" is this correct? Is last section referring to the second topmost or lowermost section? What is the residence time in the ice nucleation part if droplet activation only takes place in the second to last section?

p. 5 l. 12: delete "(quartz)" as the chemical composition analysis does not yield information on the lattice structure which defines the mineralogy.

p. 5 l. 12: How much is "more $SiO_2$"? Be more precise!

p. 5 l. 15: Why is K in biomass burning ash water soluble in but not in coal ash? Shortly state the chemical reason behind that.

p. 6, Figure 1: Do the mass fractions add up to 100 %? It appears to me that not. What is the composition of the remaining material? If this is not known, the exact amount of how much is unknown should be stated, given that the frozen fraction often is for most samples below 10 %.

p. 6: caption Figure 1: Replace "minerals" by "oxides". Again – what is shown is only chemistry, no lattice structure information which is needed for statements on the mineralogy.

p. 7 Figure 2/3: Why aren't images shown of non-US treated particles? To understand what the US treatment does to the particles it would be helpful. Also, is the chemical composition the same? This goes into the same direction as above – does the composition change with size? If it does than the US could lead to differences due to breaking up larger particles. Also the crystal ratio for non-US treated particles could be determined and a better comparison would be possible.

p. 8 l.24: The original publication is Vali (1971). It should at least be given in addition to Hartmann et al. (2013) if not replacing it.

p. 8 l.26: What does "saturation range" refer to in this case? The temperature range in which $f_{ice}$ saturates? Please be more precise!

p.9 Figure 4 and p.13 Figure 5: I appreciate that the color code is mostly consistent throughout the paper. However, it would be helpful to use in addition different marker symbols and not only circles. Also, "Fly ash from brown coal burning susp. +US *4.54" is in Figure 4 given as open circle with dot and in Figure as open circle without dot. Please change that.

p. 10 l.3: Which exact temperature range are you referring to "where effects of homogeneous nucleation can be rules out"?

p.10 l.7: How is "start nucleating ice" defined? In Figure 4 $f_{ice}$ data points are shown at temperatures warmer than -33°C and -29°C, respectively, indicating that freezing is taking place also at warmer temperatures. You seem to refer only to the plateau regime. Please write this explicitly, as this is different to other experiments.

p.10 l. 18: "ice active sites on the particle surface might affect the ice nucleation ability of a substance." is not very meaningful as ice active sites are defined by the ice nucleation ability of a particle. This sentence can be deleted.

p. 10 l.18 ff: The whole discussion on active sites is hardly based on observations but mostly speculative. It may well be that the statement given here is correct but since the mineralogy is not even known talking about crystallographic dislocations

p. 11 l.1: What type of amorphous material are you referring to? What could this material be based on the chemical composition information that you have? This sounds extremely vague.

p. 11 l.13: Is it possible to give a rough estimate by how much $f_{ice}$ would be reduced if multiple charge correction was applied?

p. 11 l.30: The sentence about $n_s$ does not belong here and should be moved further to the back.

p. 12 l.25: "increase" compared to what? I assume in comparison to LACIS. Then state that.

p.13 l. 4: How could the extended time of the ash particles in suspension lead to changes in $n_s$? Just a short comment what you ascribe it to? Reference that supports this suggestion?

**Technical Remarks:**

General: SBM has been several times spelled SMB. Please check.

p.1 l.8: replace "effect" with "influence"

p.2 l.31: insert "alone" after "insoluble fraction"

p.8 l.2: replace "could be" by "was"

Figure 4: Grey and black as colors for the two SBM fits from the literature are hard to differentiate.

p.11 l.22: typo in "homogeneous"

p.11 l.31: New paragraph before "In the plateau…"

p.14 l.21: insert space between "at" and "the Leipzig"

---

## Author Comment (AC1) · 12 Aug 2016

We thank the referee for the useful comments, all of which we considered carefully. In the following, we respond to them separately, where the original comments are colored, while the answers are given in black.

**Review of "The immersion freezing behavior of ash particles from wood and brown coal burning" by Grawe et al.**

The authors present the immersion freezing ability and efficiency of combustion ash particles from the laminar flow tube study. The authors provide the immersion freezing dataset in two metrics, f ice and n s , in the temperature (T) range of -38 °C < T < -24 °C. More specifically, this study suggests the followings: (1) the brown coal burning ash particles, which is the proxy of anthropogenic combustion ashes, are more ice nucleation (IN) active/efficient as compared to the wood burning products (the natural proxy), (2) the fly ash particles are more IN active/efficient as compared to the bottom ash particles, (3) two aerosolization processes, namely dry and wet dispersion, result in different f_ice (T) and n_s (T) spectra, (4) the ultrasonic bath application prior to particle generation increases the IN activity/efficiency of a ash sample, presumably due to the presence of fewer agglomerates in the sonicated sample than the non-treated one.

**General comments**

The topic itself is an important addition to ACP, and the authors' new IN results potentially complement the results from previous study (Umo et al., 2015, ACP), in which the droplet-freezing assay was used to investigate in the immersion freezing ability and efficiency of combustion ashes in the T range of -11 to -36 °C. In general, the authors conducted a careful study, with their dedicated effort to examine a variety of sample preparation methods, e.g. atomization vs. dry dispersion, ultrasonic bath application. Unfortunately, such care was not taken in the preparation of the manuscript, with the manuscript containing a number of ambiguous statements, non-intuitive figures and over-interpreted results (e.g., Sect. 4, P5 L19-30). I have numerous critical revisions as listed below. Major revisions/suggestions are listed first, followed by the section-based specific and technical revisions. I would urge the authors of the manuscript to thoroughly proof read their manuscript as this list is too long. While authors may be able to address these issues, I do believe that the revision of the manuscript could be time consuming and result in a significantly different paper. For these reasons, I encourage the authors to resubmit it with a completely different format.

**Specific comments**

I suggest the following major revisions.

**Discuss the representativeness of 300 nm diameter ash particles** - Justification of selecting 300 nm diameter (P3 L26-28) is missing. The authors state that the ash samples are a composite material (e.g., P15 L22), but do not discuss why 300 nm diameter is representative for their IN analyses. Different size of particles may possess different composition and/or IN behavior (e.g., Wheeler et al., 2014, J. Phys. Chem. A). I strongly suggest the authors to conduct additional IN measurements and surface characterization with different sizes. Having another set of measurements for e.g. 600 nm (or even with polydispersity aerosol population) would add clarity.

From size distribution measurements during coal combustion, we know that there is a bimodal distribution: one submicron peak below 0.1 µm and one broad supermicron peak in the range from 3 to 50 µm (Damle et al., 1981, Aerosol Science and Technology 1:1). As particle collection techniques are less efficient for submicron particles (Flagan and Seinfeld, 1988, Fundamentals of Air Pollution Engineering), we decided to select particles smaller than 1 µm. 300 nm particles were chosen because we could get sufficiently high and stable number concentrations from the variety of samples when selecting this size (the mode diameters vary from 130 to 200 nm for dry particle generation). While there are a lot of different particle types included in bulk fly ash (i.e., Ramsden and Shibaoka, 1982,

Atmos. Environ. 16, 9), it has also been found from single particle analysis that the chemical composition of fly ash particles with aerodynamic diameters between 0.2 µm and 4.8 µm is remarkably consistent (Kaufherr and Lichtman, 1984, Environ. Sci. Technol., 18). Hence, we would not assume to see a difference in n_s, would we select another particle size.

In case of bottom ash, exemplarily, measurements were performed with 500 nm brown coal bottom ash particles which showed no difference in n_s within the range of our measurement uncertainty compared to the measurements with 300 nm particles (see below, now also added as green triangles in Fig. 4). As LACIS measurements are very time consuming and we do not expect to see a difference in n_s in the size range we are able to select, we would like to avoid performing further experiments with other particle sizes.

In the manuscript, a paragraph explaining why 300 nm were chosen was added (P4 L17-28).

[Figure]

**Provide the results (no hype) to support your data interpretations and conclusions** - One of the major findings out of this study is that the immersion behavior of brown coal ash particles changes depending on the sample preparation methods. The authors need to **investigate and discuss** in-depth physical reasons of why the observed difference appears. It seems not meaningful to attribute the reason to just the 'sample preparation and particle generation' (P11 L14-16), speculate potential reasons (e.g., P12 L1-3; P13 L10-14) and put it off as future work (e.g., P14 L28-32; P15 L20-25). To date, there have been some recent publications attempting to identify potential reasons of the data diversity due to different experimental methods and sample preparation methods (e.g., Hiranuma et al., 2015, ACP, Emersic et al., 2015, ACP). I suggest the authors to elaborate what can be further clarified on top of given previous findings. Reporting only the IN observations seems not novel enough to complement the previous ash IN study (i.e., Umo et al., 2015).

You are correct that a difference in IN activity between dry particles and particles in suspensions has been observed before. The studies you cite above also did not reach final conclusions on this. And we are the first to examine the difference between using and not using ultrasonification. This is an important new contribution to the topic, while obtaining the reasons for our observations is beyond the scope of this study. However, in our manuscript we now also discuss the literature you suggest, adding their speculations to ours (P18 L5-P19 L3).

Additionally to the discussion about differences between LACIS and cold stage measurements, we included a paragraph about differences between dry dispersion and wet particle generation with the help of an atomizer (P16 L7-P12). Concerning this, there have been observations showing that composition of wet and dry generated mineral dusts differ from one another and causing differences in

hygroscopicity (Herich et al., 2009, Phys. Chem. Chem. Phys. 11; Sullivan et al., 2010, Aerosol Science and Technology, 44).

**Consider removing Sect. 4** - The atmospheric implications (P14 L8-17) sound too speculative and ambitious. The abundance data (concentration and size distribution) with some spatio-temporal distributions seem necessary to estimate the ambient ash-derived INPs. The back of envelop calculation presented here seems not novel enough for you to support your sub-conclusion, which appears in P14 L13-14 ("In conclusion,…") and P1 L13-14 ("ash from brown…a regional scale"). If the authors wish to keep the atmospheric implication section, the difference between airborne (fly) ash and surface (bottom) ash with respect to their mixing state, degree of agglomeration and atmospheric lifetime should be somehow discussed. Also, discuss the contribution of natural ashes to ambient INPs vs. that of anthropogenic ones. Otherwise, the authors may consider removing the entire section.

We consider this section an interesting supporting information to the manuscript. Differently from mineral dust particles, which have received much attention, not much is known about ash as INP in the atmosphere, and this section shows that it is worthwhile not omitting this aerosol type. We therefore do not want to remove this section. However, we have added some additional / updated information and more recent references.

**Tighten up the writing by removing unnecessary words** - Improving the language, structure, presentation seems necessary. The authors should avoid making a review question how careful the research team is.

For example, I suggest minimizing ambiguous (and unnecessary) adverbs and adjectives to make the manuscript more scientifically sound than the current form. Such words should be replaced with specific and explanatory descriptions/values. Otherwise, the authors should reinforce them by adding proper citations. My suggestions include, but not limited to:

P1 L6: more (specify how much in what T range?)
P1 L14: at least
P2 L12: major
P2 L15: rough
P2 L20: for a long time & large
P2 L22: strongly
P2 L24: very similar
P2 L32: lower (than what?)
P3 L4: slight
P3 L23: some
P3 L24: mostly (define which samples)
P3 L30: larger & more (than what?)
P4 L30: exactly
P5 L12: most (amongst what?)
P5 L13: slightly
P5 L14: significantly
P5 L17: most striking
P5 L19: small
P5 L23: perfectly (I do not think so) & most
P5 L24: significantly less (is it fair to say this by inspecting a single snapshot picture?)
P5 L27: similar
P5 L30: small
P5 L31: small
P6 L1: supposedly & obviously

P6 L5: possibly & weakly
P8 L10: almost entirely
Fig. 4 caption: at least
P10 L2: significant & low
P10 L4: most effective & least & rather similar
P10 L6: small & more (than what?)
P10 L7: more (than what?)
P11 L2: certain (specify)
P11 L15: considerably
P11 L16: by several tens of percent (just give a number)
P11 L20: probably exclusively & completely
P12 L1-2: most likely
P12 L24: numerous
P12 L25: large
P12 L26: higher (as compared to what?)
P12 L29: larger (how much?)
P13 L12: likely
P14 L15: significantly low
P14 L24: significantly (how much?)
P15 L3: barely
P15 L4: significant
P15 L6: likely
P15 L 11: eventually
P15 L20: most likely

The text is now more specific where useful. However, for those of the locations you listed below that refer to figures, adding numbers to the text only decreases the readability without adding information to the manuscript. Hence, where numbers can be seen from figures, we did not change anything. Furthermore, it is often not possible to give specific values, as those have not been reported in literature until now. Words like "possibly", "supposedly", "probably", "likely" need to be used in those parts of the manuscript where we discuss hypotheses.

**Improve the figure and table presentations** - In general, all figures and tables should be self-explanatory. My suggestions include the followings:

Fig. 1: It seems that more than 50% of mass are composed of materials that are not listed in the figure. The authors need to clarify this point to the reader by adding descriptive text or adding another group of 3 bars showing the sum of the other components in the figure. The authors implies such contribution may in part come from 'amorphous' material (P11 L1). I suggest the authors to give an idea of what they are (perhaps carbonaceous materials?). I also suggest the authors to give a proper reason of why beech ash composition is not shown (P5 L9). Perhaps, presenting data (± uncertainty) with the table format may be more intuitive to the reader than using the figure format. In addition, the figure caption should read "bottom ashes from spruce, birch…".

In addition to Fig. 1 we also included a table format (Tab. 3) with the corresponding values. There, the sum of the analyzed components as well as the associated uncertainties are given. We did not want to omit the diagram, as the illustration of the numbers might help to take in the large amount of information. You correctly observed that the given values do not add up to 100 %. This is due to the analysis procedure: It is standard to recalculate the concentrations of the ten major ions into the mass of their most common oxide forms, even though they may have had other counterions in the original sample. In some cases the compounds are indeed oxides, but we do not know the counterions in each case. K for example would be more likely to occur as KCl, K2CO3 etc., than as K2O. Any missing percentage is due to other than these ten elements and the fact that other counterions would have been involved. This has been added to the manuscript (P7 L2-P9 L2).

From previous studies we know that the amorphous material in fly ash is mainly aluminosilicate glass (Ramsden and Shibaoka, 1982, Atmos. Environ., 16, 9; Querol et al., 1996, Atmos. Environ., 30, 21). This was added on P14 L22-24. The low loss on ignition (LOI) value for fly ash (see Tab. 3) indicates that there is barely any carbon left in our sample. Hence it cannot be contained in the amorphous fraction.

Beech ash is not shown in the analysis as it was provided later, after the other samples had already been analyzed in Sweden (added on P6 L24-P7 L1). We decided to include the data, even without the chemical analysis, because we wanted to show that beech ash is comparable to the other wood bottom ashes.

The caption was corrected.

Fig. 2: I see at least six particles that have >1 µm diameter in the panel b. As stated in P5 L24-26, the doubly charged particles of 300 nm cannot be >1 µm. The authors state that "…the number of particles with three or more negative charges was negligible…" (P3 L32-P4 L2), but it seems not negligible and contradicting. In terms of the number, those large particles may have only a small contribution. However, they may substantially contribute to the total surface. If that is the case, they should be accounted for the immersion freezing parameterization. Otherwise, your ice nucleation active surface site density would be overestimated because of overlooking the presence of large particles. Adequate surface estimation is one of the important keys for the n_s parameterization. Ultimately, the authors may want to assess if such correction can explain the discrepancy between dry and wet (or not). Just to start with, you may first estimate the particle size distributions by analyzing SEM images (i.e., estimate the area equivalent diameter for several hundreds of particles; see Hoffmann et al., 2013, AMT). This approach may be better than relying only on the USHAS measurements. In addition, such work will help clarify the vague statement in P5 L24-30.

The multiple charge correction was performed based on size distribution measurements with the UHSAS. For this, 300 nm particles were generated in the same way as for LACIS measurements and sampled for several minutes. For UHSAS and LACIS measurements, particle production only operated for short amounts of time (<30 min), before cleaning the cyclone prior to a next set of measurements. The filter samples, however, were collected over several hours to assure a sufficiently high concentration on the filter surface. New UHSAS measurements (see below) show that the cyclone, which we used to minimize multiply charged particles, is filled after around 30 minutes, so that more larger particles are present than for the beginning of the UHSAS measurement. Additionally, it was also seen in APS measurements that there was an increasing amount of supermicron particles passing the DMA and cyclone as the generation system ran for more than 30 min. As LACIS measurements typically last around 20 minutes only (due to wall glaciation effects), and the cyclone was cleaned after each LACIS measurement, we are certain that the UHSAS determined multiple charge fractions (see below) correspond to the actual fractions in the flow tube. The large particles which can be seen in Fig. 2 b) of the original manuscript have accumulated during the long collection time in which the cyclone was not cleaned often enough. We decided to collect fly ash particles from dry generation on filters once again, now cleaning the cyclone every 30 minutes. The new SEM pictures (see examples below, one was added to the manuscript as Fig 1b) show a majority of particles in the size range of 300 nm. 68 of 84 counted particles were in the size range of around 300 nm, while 16 particles classified as particles larger than 500 nm. This might not be significant from a statistical point of view, but it is similar to the findings of the UHSAS measurements, where 19 % of all particles were identified as doubly charged.

We omitted Fig. 2b) from the original manuscript to avoid confusing the reader with the occurrence of large particles which are only present on these particular filters but not in the LACIS measurements. In light of the new observations, we changed the paragraph on P5 L19-30 of the original manuscript accordingly (P9 L14-24).

[Figure]

SEM images of fly ash particles from dry generation:

[Figure]

Fig. 4: The authors need to explain the factor of 4.54 in text at the first appearance of Fig. 4. I also suggest clarifying what the 'clay mineral baseline' means. Better presentation may be made with multiple panels. For example, the authors may use individual panels for ash type comparison, dry vs. wet, +US vs. –US. The same may apply to Fig. 5.
Fig. 4 (now Fig. 3) was changed as suggested and the correction factor is now explained in the caption. We added further explanation concerning the clay mineral baseline on P14 L30-P15 L1 as well as in the caption of Tab. 4. The original Fig. 5 (now Fig. 4) was not altered in terms of its layout.

Fig. 5: Specify if your $n_s$ metric is based on the BET specific surface area or the geometric surface. Though I agree with your statement in P13 L8-10 (i.e., the influence/difference is small anyway), it is important to compare the results with a same metric. If this figure contains both BET-based $n_s$ and geometric $n_s$ spectra, I suggest presenting all data and spectra using either one of two metrics. The authors may simply apply a factor of 4 (P13 L9) for the conversion.
This is a fair point. However, instead of showing the BET based $n_s$ values calculated from the specific surface area given by Umo et al. (2015), we decided on introducing a second y-axis to the plot. The left y-axis is for the $n_s$ values from our measurements assuming spherical particles (circles) and the right y-axis is for the BET based $n_s$ values (lines) reported by Umo et al. (2015). We added arrows to indicate which axis belongs to which data set. We decided to present the data in this way, because we cannot provide BET measurements for our samples within this short time frame and we cannot be sure that our samples compare to the samples from the Umo et al. (2015) paper concerning their specific surface area. However, we also recalculated our $n_s$ into a BET based $n_s$ using the specific surface area values given by Umo et al. (2015) and saw a change of a factor 3.5 at most. This and a detailed explanation of Fig. 4 (former Fig. 5) have been added to the manuscript (P17 L10-12 and P17 L23-33).

**Add proper references** – I suggest the authors to give a careful look at the followings:

P1 18: Add citation after "…and climate models".
We added Koop and Zobrist (2009, Phys. Chem. Chem. Phys. 11) as a reference.
P1 L21: Replace Murray et al., 2012 with Koop et al., 2000 (Nature); Murray et al., 2010 (Phys. Chem. Chem. Phys.); Rosenfeld and Woodley, 2000 (Nature).
Done.
P2 L5: Add citations after "…been conducted".
Done.
P3 L1-2: Whale et al. (2015, AMT) may be a good reference to add here.
Added.
P3 L20: Add reference for your dry dispersion method.

There is no publication mentioning the generator. We now cite a dissertation from our group which is unfortunately only available in German (M. Rösch, 2015).

 Add reference for "water contamination". The authors may also want to reduce the tone and call it as a background contribution or something similar.

We changed water contamination to impurities. As references we added Budke and Koop, (2015, AMT 8) and Whale et al. (2015, AMT, 8) (P5 L9-11).

P7-8: Provide the reference for R1 & R2.

We chose not to include the chemical reactions any more, as they were already described in the text. Steenari et al. (1999, Fuel, 78) was added as a reference (P11 L1).

P10 L17: Any reference for insoluble K to be highly ice active? You should not speculate for all insoluble materials to be IN active.

K is insoluble in K-feldspars which are known to be ice active. Sure, we do not know about the mineralogy of our samples, but we have made it clear that this is an assumption rather than a fact.

P12 L1-3: multiple citations seem missing.

This part was revised. We added two studies as a reference where it was shown that particle generation affects particle surface properties and composition (Herich et al., 2009, Phys. Chem. Chem. Phys. 11; Sullivan et al., 2010, Aerosol Science and Technology, 44).

P12 L8: Look into references given in Zolles et al. (2015). There have been a few other studies of active sites and their influence on IN activities.

We now also cite Hiranuma et al. (2014, ACP) and state that the fragmentation in the ultrasonic bath could have a comparable effect on the ice fraction as the milling procedure.

P12 L16: e.g., Connolly et al., 2009 (ACP), Niemand et al., 2012 (JAS)

Added.

Other specific and technical suggestions sorted out for each section are listed below.

**Abstract**

P1 L2: …trigger ice nucleation when they interact with water vapor and/or supercooled droplets.
P1 L4: …worldwide, and…
P1 L6-7: I suggest separating this sentence into two sentences.
P1 L6: ice active in the immersion mode
P1 L8: …the effect of various particle generation methods on…
P1 L8: For this → For instance
All done.
P1 L14: heterogeneous ice nucleation → immersion freezing in the T range of XX to YY °C

We changed "heterogeneous ice nucleation" to "immersion freezing". However, including the temperature range does not help the readability of the sentence.

**Sect. 1**

P1 L21-23: The authors may focus on immersion freezing by rephrasing L21-23 to, "At water saturation, this temperature limit of droplet freezing … referred to as immersion freezing.". Otherwise, describe heterogeneous IN and then immersion freezing. Note that the deposition nucleation can occur at temperatures below -38 °C.
Done.
P2 L4: However, up to now only a very few → To date, only a few
Done.
P2 L5: → material, which
Done.
P2 L12: contribute to
Done.
P2 L12: → Furthermore, the ash from natural source is…

Biomass burning is often anthropogenically induced. We'd rather not imply that all biomass burning ash is emitted due to natural causes.

P2 L13-14: Awkward sentence. Rephrase.

We changed the sentence to: "The impact of ash particles as potential INPs must be put into perspective by comparing ash emission rates to those of other INP containing aerosols…". Hopefully, this brings more clarity.

P2 L18: I wonder where the authors find the 7% number in DeMott et al. (2003).

It is not stated directly in the text. However one can calculate that 20% of 33% is equal to 7% of the total ice crystal residue number.

P2 L19-20: I suggest clarifying that the result presented in DeMott et al. (2003) is from a limited time segment of measurements in cirrus clouds.

Done.

P2 L23-26: I do not understand what this sentence mean. Please clarify.

We are not completely sure what is unclear. This part was divided into two sentences to increase readability.

P2 L27-29: I suggest separating this part into two sentences.

Done.

P2 L33: What are water soluble components? Sugars? Any biological materials? How important are they for IN as compared to the insoluble components?

The sentence before the one you refer to here explains your last question ("…particles freed from water soluble components initiated freezing at lower temperatures."). Maybe you were thinking of ice active bacterial macromolecules, which is not what we are talking about here. Hence, the addition (P3 L7-8) that it is dominantly CaSO4 hopefully clarifies your questions well enough.

P2 L35-P3 L2: Provide more information regarding Umo et al. (2015). At least the investigated T ranges for individual ash materials and their IN efficiency/activity with some quantities.

Done.

P3 L2: Reword "however". I do not find smooth transition.

Done.

P3 L5: Reword "assumed". e.g., suggested/postulated

Done.

P3 L11-12: Move this part to the method section.

Another referee requested a more detailed motivation of our study in the introduction. We would hence like to keep the sentence in this position because it underlines the difference in methodology in comparison to the previous study.

**Sect. 2.1.1**

P3 L21: as → along with

Done.

P3 L21-23: Subdivide the sentence ("Variations caused … transported downwards.") into two.

Done.

P3 L23-24: Clarify if glass beads have any influence on modification of particle surface (e.g., by scratching) and IN behavior.

We do not expect the beads to have any influence on the IN behavior of the particles. Firstly, we used only 20 beads, hence, the probability of a particle which would later be investigated colliding with one of the beads is rather low. Secondly, if collisions with these beads would lead to a particle modification, then collisions with the flask wall would also. Wall collisions are common with other dry dispersion techniques (e.g., fluidized bed generator) and are not expected to modify the particle surface, so why should our beads? Secondly, the beads did not appear milky after several hours of particle generation, indicating that no significant scratching took place. An explanation was added to the manuscript (P4 L11-12).

P3 L24-26: I suggest rephrasing the complete sentence ("In addition … dryer unit."). Just say you used a custom-built atomizer. Is it perhaps the one used in Wex et al., 2015, ACP? If so, add it as a reference.

We prefer not to cite anything here, as this is a standard procedure used frequently by many groups in aerosol research.

P3 L31: efficient → active due to the presence of large surface

Done.

**Sect. 2.1.2**

P4 L4-5: Awkward sentence. Rephrase. The LACIS reference is repetitive.

We changed the sentence to "The immersion freezing behavior of the previously generated and size selected ash particles was investigated with LACIS."  The reference was deleted.

P4 L6: "LACIS offers … nucleation processes."; please provide the reference showing the 'surface' (of the droplet assay plate - I assume this is what you mean) interferes with IN measurements.  If not, I suggest rephrasing the sentence.

We omitted the half sentence in question.

P4 L14-15: I suggest briefly clarifying how the phase discrimination can be done in text.

We added that "the approach to determining the phase state of the hydrometers is based on the fact that the former polarization of light is maintained for scattering at spherical hydrometers (supercooled water droplets) while non-spherical hydrometeors (ice particles) cause depolarization" (P5 L18-20).

**Sect. 2.2**

P4 L18-23: I suggest to summarize your ash samples using a table along with columns of ash type (bottom or fly), source (natural or anthropogenic), particle generation method (dry or wet),  and information regarding your IN experiments (e.g., examined T range and IN observed T range).

We included the suggested table in the manuscript (Tab. 2).

P4 L25-26: "It has to be noted that…"; not much adding. I suggest removing this sentence.

The composition of the coal depends on the deposit it was taken from. Consequently, ash from different coal will have different compositions. We think the reader should know that both coal types were taken from different deposits.

P4 L27: If it is commercially available, specify it.

We could only find out the manufacturer of one of the domestic ovens. For two of the ovens it was simply not traceable. It turned out that one was actually home-built. In the light of this, we omitted "commercially available" in the cited sentence.

P4 L27: Explain how you "extracted" it. Provide some information regarding the electrostatic precipitator (or reference). What is the cutsize? What is the collection efficiency? etc.

The ash was simply taken out of the collection tank of the electrostatic precipitator. Our contact at the power plant provided the overall efficiency which is 99.98 %. This information was added in Sec. 4. Furthermore, the flue gas behind the filter must not contain more than 50 mg per m³. There is no information concerning the cut-size or the size-dependence of the efficiency.

P4 L28: → station, which

Done.

P5 L4: affected → modified

Done.

P5 L5: "one fly ash suspension sample"; specify which one and why the authors choose this one in text.

This simply implies that several identical samples were prepared, one of which was not put into the ultrasonic bath. Nothing was changed.

P5 L9: Provide brief description of atomic adsorption methods along with proper reference(s). The rest of the paragraph (up to L18) may better fit in the results section. Consider reorganizing the sections.

Our partners in Sweden further specified the method to be Inductively Coupled Plasma Sector-Field Mass Spectrometry (ICP-SFMS). This is now stated in the text. We added a reference describing the method (Zheng and Yamada, 2006, Talanta, 69).

We did not move the chemical composition analysis to the results section as none of the other referees raised objections. We think the sample characterization better fits in the methods and materials section.

P5 L11: obtained → estimated

Done.

P5 L19-30:  I suggest the authors to minimize over-interpretation from snap shot pictures. The authors may consider providing the reader with the proxy of particles' sphericity. For example, a particle aspect ratio of individual particles can be estimated by the 2D SEM images. Inspecting several hundreds of particles and providing statistically valid data in the result section will strengthen the paper.

We weakened the statement and avoided the use of the word "spherical".

P5 L23: Irregular shape may suggest the predominance of carbonaceous compounds (e.g., Hiranuma et al., 2008, Atmos. Environ.). Do the authors have any information regarding its organic fraction and content?

Yes, there is information concerning the organic fraction which is now included in Tab. 3. Here the loss on ignition (LOI) value corresponds to the organic content, i.e., it tells us about amount of unburnt fuel in the sample:

Spruce: 22.9 %, Coal bottom: 10.5 %, Coal fly: -0.8 %, Birch: 26.7 % (uncertainty +- 5%)

We do not know the exact composition of this organic fraction, but there will be carbonaceous matter included since these particles are produced during incomplete combustion (Kucbel et al., 2016, Perspectives in Science, Vol. 7). We added a paragraph about the relation between carbon content and particle shape including the citation you suggested (P9 L25-32).

P5 L23-24: "…most show significantly less surface defects…"; it is hard to judge the presence of defects by looking at given SEM pictures with low magnification. Can the authors provide the image with high magnification as an example? Better option would be measuring BET surface of both bulk samples and comparing each other.

We weakened the statement and now say that the lowest BET values were found for coal fly ash in the previous study by Umo et al. (P9 L31-32).

P5 L25-26: I wonder why UHSAS was not able to measure them.

Our new SEM images show that the UHSAS was never the problem. The UHSAS has an upper size limit of 1 µm. As already elaborated on in more detail above, we saw the large particles on the filters because particles were collected for several hours during which the cyclone filled up. The UHSAS and LACIS measurements, however, did not last longer than 20 min and the cyclone was cleaned after each measurement. We are now sure that the UHSAS determined double charge fractions correspond to the fractions in the flow tube (see estimation given above).

P5 L27-28: "…similar particle losses should have occurred."; I suggest  removing any opinion statements. The authors may provide the results instead if available.

The entire paragraph (P5 L24-30 of the original manuscript) where we try to explain possible reasons for the discrepancy between SEM and UHSAS was removed. As we now know that the filled cyclone was the reason, the discussion about particle losses seems unnecessary at this point. However, we added a short sentence concerning this on P5 L4-5.

P5 L33-35: "It is reasonable to assume … additional fly ash particles."; I do not understand your logic here. Please clarify.

The needle shaped particles are not present in the dry sample and hence are special to the suspension. Insoluble material will not change when put in water, so these needle shaped particles can be assumed to be formed from dissolved material, and hence our statement. As it is not clear to us what was unclear to you. Here, nothing was changed.

P6 L3: → all components are dissolved in droplets

Surely not all, just the water soluble components. Nothing was changed.

P6 L5 to the end of this section: this part seems not belonging to the method section. Should be moved to the discussion section?

The section is called "Methods and materials". As we kept the discussion of the chemical composition of the dry samples in its place, it would not be consistent to move the part about the soluble components to the results section.

P8 L9-10: Awkward sentences. I suggest rewording.

Done.

**Sect. 3**

P8 L13-16: Subdivide this sentence to two parts.

Done.

P8 L17-19: Why don't you look at only quasi-spherical ash particles on the SEM images? Analyzing the SEM images, you should be able to distinguish ash particles from the recrystallized components, which seem possessing high aspect ratios. This procedure perhaps enables you to estimate the contributions of multiple charge components. The same applies to your statement in P11 L12-13. The correction seems feasible and important.

The correction you suggest here was done. This was described on P11 L26-30 of the original manuscript. As the number of ash particles on the filters, compared to others, was rather small, a number of multiply charged ash particles cannot be derived from these SEM images. With the UHSAS it is not possible because the two populations cause overlapping signals (P8 L17-19 of the original manuscript). However, to estimate the multiple charge fractions in the suspension measurements, we weighted the bipolar charge distribution (Wiedensohler, 1988, Journal of Aerosol Science), i.e., the probability of the particles to receive one, two or three negative charges in the neutralizer, with the measured size distributions of the ash-water suspensions. There is a caveat to this estimate, as we do not know how the size distributions would look like for the insoluble particles only. It turned out, that the highest multiple charge fractions would probably occur for the fly ash suspension (+US), where we calculated 80.5 % singly, 16.8 % doubly, and 2.7 % triply charged particles. The multiple charge fractions were even lower for the other suspension samples. Would we perform the multiple charge correction using these fractions, our measured data would be reduced by a maximum factor of 2 only. A short version of this was added to the manuscript (P15 L10-16).

P8 L21: → in Fig. 4. These calculations are based on…

Done.

**Sect. 3.1**

P10 L3: Specify the T range for the reader.

Done.

P10 L4-6: Discuss why all the wood burned ash particles have similar IN behavior. There seems difference in $SiO_2$ content for spruce (~10%) and birch (<5%). Based on your statement in P5 L12-13 (i.e., the higher $SiO_2$, the more IN efficient), could the $SiO_2$ content be responsible for the observed result of $n\_s$ spruce $> n\_s$ birch in Fig. 5? If this holds true, the bulk beech ash should contain the highest $SiO_2$. Would you consider carrying out the atomic adsorption measurement of beech ash (currently missing without any intuitive explanation) to support your idea?

Yes, $SiO_2$ could play a role for the wood ashes as well. However, the wood ashes are really close together (logarithmic scale) so that we would like to avoid the over-interpretation of these slight tendencies.

P10 L9 - P11 L3: The authors suggest that the fraction of $SiO_2$, Hg and insoluble K has some contributions to determine the immersion freezing behavior of ashes based on their bulk observation of atomic adsorption technique (P10 L9-22). In the following sentences, the authors

introduce the hypothesis, which infers that the amorphous material content is also important as a determining factor of IN behavior.

Discuss which one is more dominant. Further, what is the mixing state of your ash samples? What is the relative importance of the mixing state as compared to the "bulk" composition?

Our measurement methods are not able to distinguish between the importance of these different possible contributions, and hence we discuss them as suggestions. The same applies for the mixing state: as long as it is not known what it is that makes the difference (or, in other words, what it is that makes the IN activity - which is something the whole IN community is looking for), it can also not be stated how the mixing state of this unknown component is. Still, we had hoped that the bulk composition would reveal something more meaningful. This was, unfortunately, not the case.

P10 L10-12: Rephrase. Not easy to follow.

Done.

P10 L14: → act as INP. This previous observation may be relevant to our study as…

Done.

P10 L14: Is the atomic adsorption technique used in this study sensitive to iodine? If so, please show I in Fig. 1 too.

Iodine was not investigated because, initially, we expected it to be a minor component in the ash samples. Unfortunately, there is no time to rerun the analysis and investigate for the I content. We cannot rule out HgI2 as a factor, so nothing was changed.

P10 L16: Why soluble? Presumed? Or measured? Please clarify.

We added the following explanation on P14 L15-16: "K in wood ashes is present in the form of soluble salts and oxides, whereas coal ashes contain K in clay minerals with low solubility", citing Steenari et al., 1999, Fuel, 78. In the course of this, we changed the previous citation of Andreae et al., 2004, Science, 303 (original manuscript P5L16) to Steenari et al., 1999, because the reasoning is presented more clearly in the latter.

P10 L18-22: "Apart from the chemical composition, other properties such as ice active sites…"; Discuss the relative importance of what you found to the other potential factors. I suggest rephrasing and clarifying the last sentence. I do not understand what the authors mean here.

This sentence was deleted. We cannot make any statements on how likely these speculations are.

**Sect. 3.2**

P11 L12-13: I disagree. The correction is possible and feasible, and the authors should account for it. See my comment in Sect. 3.

As stated earlier, we are talking at cross purposes here. The multiple charge correction is not synonymous with the correction for the occurrence of needle shaped crystals (P11 of this document).

P11 L13-14: This sentence seems not needed here. I suggest removing.

The sentence is needed to remind the reader that n_s values might be lower for the wet generation measurements.

P11 L14-16: The authors need to investigate and discuss physical reasons of why the influence of the US application on IN behavior is material dependent rather than just saying 'sample preparation' methods resulted in the difference. The same goes to P13 L3-4.

We now discuss possible reasons for the difference in IN efficiency when changing from dry to wet particle generation in more detail (P16 L6-25 and P17 L35-P19 L9). That we do not see a decrease in f_ice for spruce suspension is hence related to the fact that the discussed possible reasons are apparently not relevant in case of particles from the spruce ash suspension.

P11 L18: meaning → suggesting

Done.

P11 L17-30: Multiple topics seem to be squashed within this short section. Reorganize and rephrase the sentences. Again, the scaling factor of 4.54 should be introduced prior to its first appearance in Fig. 4.

The scaling factor is now mentioned in the caption of Fig. 3 (former Fig. 4). We think the detailed explanation better fits where we discuss the results for wet particle generation. However, we reorganized the paragraph and now start with the discussion of the filtered fly ash suspension and later go on with the non-filtered suspension. We hope that the need for the scaling factor of 4.54 and the paragraph itself are more understandable now.

P12 L1-3: Sounds too speculative. Any experimental evidence of a destruction of former active sites? I do not understand the last sentence. How do water soluble components play a role in what?

This was rephrased. We do not mention the destruction of sites anymore. Instead there is a discussion about how particle properties could be changed during wet particle generation (P16 L6-12).

P12 L8-13: What is the atmospheric implication of your findings here?

The relevance of these findings is, that possibly samples that are treated with an ultrasonic bath might yield different concentrations of ice nucleation active particles than when this treatment is omitted, and that the use of the respective reported concentrations should be done with caution. This follows implicitly from the text, and nothing was changed.

P12 L23-24: Repetitive. Delete.

Done.

P12 L24-30: Rephrase. This part may better fit in the introduction section as part of an example of previous work regarding ash IN characterization.

We think the information is better suited to be stated here, as it belongs to the explanation of Fig. 4 (former Fig. 5).

**Sect. 3.3**

P12 L15: ice nucleation active surface site density

Done.

P13 L5: nucleation spectrum → ns spectrum

Done.

P13 L10-14: Sounds too speculative. I suggest removing the opinion statements (that is, must be, expect, likely) and rephrasing the sentences.

We added a references for the time-dependence statement (Ervens and Feingold, 2012; Welti et al., 2012; Wex et al., 2014) and a discussion about how the methodology (wet generation vs. cold stage suspension) could affect n_s values citing the suggested papers by Hiranuma et al. (2015, ACP) and Emersic et al. (2015, ACP).

**Sect. 4**

P14 L2-4: The authors may consider removing this paragraph, which seems not adding much.

P14 L5-8: These sentences better fit in the introduction section (e.g., P2 L15) rather than here.

The section has been revised (see P3 of this document).

**Sect. 5**

P14 L21: at the

Done.

P14 L23: differences (in what?)

Differences in the immersion freezing behavior. This was added.

P14 L28: "a decisive factor…"; I disagree. The authors need to provide much more comprehensive composition data (e.g., beech ash data, bulk vs. single particle, mixing state, organic speciation) than what are presented in this manuscript to have this statement as your conclusion.

As we cannot provide the requested data for now, we weakened the statement and omitted "decisive" so that the sentence now reads: "… however, a factor could be the presence of insoluble K in the coal ashes.".

P14 L28-32 & P15 L20-21: I agree with the authors that multiple particle properties inherently influence particle's immersion freezing behavior, and it is not easy to understand what the controlling(s) factor is(are). With some extra experiments suggested above, the authors may be able to shed light on the questions raised here.

Additional experiments to round up our results were done with respect to multiply charged particles (see our answers to your remark on P5 of this document), as well as regarding particle hygroscopicity (as requested by another reviewer, now added in P11 L17-21). However, further additional examinations are beyond the scope of the here presented study, so that mentioning these issues here, at the end of the summary and conclusion sections, can be seen rather as a hint towards readers on directions in which additional rewarding research can be made.

P15 L9: nucleation spectrum → $f_{ice}$ and $n_s$ spectra; the same applies to elsewhere, e.g., P15 L17
Done.
P15 L12-13: What is its atmospheric implication?
We added: "…which could imply that previous results might have overestimated the ice nucleation ability of certain substances." (P21 L18-19).

References cited in our answers:

Andreae et al., Science (2004)
Budke and Koop, AMT (2015) and
Damle et al., Aerosol Science and Technology (1981)
Emersic et al., ACP (2015)
Ervens and Feingold, ACP (2012)
Flagan and Seinfeld, Fundamentals of Air Pollution Engineering (1988)
Herich et al., 2009, Phys. Chem. Chem. Phys.
Hiranuma et al., ACP (2014)
Hiranuma et al., ACP (2015)
Kaufherr and Lichtman, Environ. Sci. Technol. (1984)
Koop and Zobrist, Phys. Chem. Chem. Phys. (2009)
Kucbel et al., Perspectives in Science (2016)
Querol et al., Atmospheric Environment (1996)
Ramsden and Shibaoka, Atmospheric Environment (1984)
Rösch, PhD thesis University of Leipzig (2015)
Steenari et al., Fuel (1999)
Sullivan et al., ACP (2009)
Umo et al., ACP (2015)
Welti et al., ACP (2012)
Wex et al., ACP (2014)
Whale et al. AMT (2015)
Wiedensohler, Journal of Aerosol Science (1988)
Zheng and Yamada, Talanta (2006)

---

## Author Comment (AC2) · 12 Aug 2016

We thank the referee for the useful comments, all of which we considered carefully. In the following, we respond to them separately, where the original comments are colored, while the answers are given in black.

**Review of "The immersion freezing behavior of ash particles from wood and brown coal burning" by Grawe et al.**

This manuscript presents an interesting data set regarding the ice nucleating abilities of ash particles which are currently poorly understood. Given the lack of information on this topic, and the potential that ash particles may have to influence mixed-phase cloud formation, the present results are a valuable contribution to the ice nucleation community. Although the present manuscript possesses many similarities with the Umo et al. (2015) study, I still see a small level of novelty on it. I got attracted by the title of the paper which indicates that the ice nucleating abilities of wood and brown coal burning ash particles were studied, but I am disappointed that a clear explanation of why brown coal ash particles are better INP than the wood ash particles is not provided. Additionally, the conclusions are not clearly supported given the lack of some key experiments. Therefore, I think that this paper could be accepted in ACP only after the following points are clearly addressed. Note that this review was prepared without reading the comments given by referee #3; therefore, I apologize for any overlap between the two reviews.

**Major comments:**

1. Multiple charge correction was applied to the dry samples only. Which percentages of the particles were multiple charged? Based on what data was this correction conducted as the authors indicate that the UHSAS did not clearly detect the multiple charged particles? How good is the agreement between the SEM and the UHSAS at detecting multiple charged particles? I encourage the authors to report the size distribution for each sample and the resulted size distributions after size selection (300 nm).

The multiple charge correction was performed based on size distribution measurements with the UHSAS. New UHSAS and SEM measurements show that we saw a discrepancy between the two because the cyclone, which we used to minimize multiply charged particles, is usually filled after 30 min. Filter samples were taken for several hours, meaning that multiply charged particles accumulated as soon as the cyclone was filled and not immediately cleaned. The new measurements show a good agreement between UHSAS and SEM, as long as the cyclone is cleaned sufficiently often. As LACIS measurements typically last around 20 minutes only (due to wall glaciation effects), and the cyclone was cleaned after each LACIS measurement, we are certain that the UHSAS determined doubly charged fractions (which have been added to the manuscript in Tab. 1) correspond to the actual fractions in the flow tube.
We omitted Fig. 2b) from the original manuscript to avoid confusing the reader with the occurrence of large particles which are only present on these particular filters but not in the LACIS measurements. In light of the new observations, we changed the paragraph on P5 L19-30 of the original manuscript accordingly (P9 L14-24).

2. There is a poor consistency in the experiments conducted with the 5 ash samples as shown in the table below. I am wondering why there is too much data missing. This lack of information reduces the robustness of the drawn conclusions regarding the particle generation methods and the effect of treating the samples with ultrasound. I suggest conducting more experiments to fill out the table below.

| | Material | Fly/Bottom | Dry Generation | Wet Generation | Ultrasound | Without ultrasound | Filtered | SEM | AA |
|---|---|---|---|---|---|---|---|---|---|
| Wood | Spruce | B | X | X | X | | | | X |
| | Birch | B | X | | | | | | X |
| | Beech | B | X | | | | | | |
| Coal | Brown coal | B | X | X | X | | | X | X |
| | Brown coal | F | X | X | X | X | X | X | X |

You are right in suggesting that further experiments would shed more light on some of the drawn conclusions. However, LACIS measurements are very time consuming so that we do not see a possibility to provide the missing data on a time scale of weeks.

We investigated the most efficient of our samples (coal fly ash) most intensely, also including filtering of the suspension and suspensions without ultrasonic treatment as we expected the biggest difference for this sample. Measurements of the bottom ash suspensions without ultrasonic treatment would not be adding much, as they were already close to the detection limit with ultrasonic treatment.

The beech ash was not investigated by means of atomic adsorption spectroscopy as it was provided later, after the other samples had already been analyzed in Sweden. We decided to include the data, even without the chemical analysis, because we wanted to show that beech ash is comparable to the other wood bottom ashes.

3. Is it the needle formation exclusive to fly-ash brown coal particles? Why SEM images of the bottom brown coal (suspension) are not presented? Why the SEM analysis was not applied to the wood ash samples? The authors indicate that the needle formation in the fly-ash brown coal particles may be cause by CaCO 3 which was formed by the presence of CaO detected by the atomic adsorption (AA) analysis. However, Figure 1 shows that the levels of CaO in the wood samples are much higher than in the brown coal samples. Therefore, it would be nice to see the SEM images of e.g. Spruce which has the highest concentration of CaO.

The only time when SEM images were needed for our evaluation was when we determined the fraction of fly ash particles vs. the fraction of solution particles for experiments with the fly ash suspension. Hence, with the selection of SEM images shown, we did not intend to give an overview for all samples. Now, we included an additional SEM picture of spruce ash particles from wet particle generation in the manuscript (Fig 2c) which does not show needle formation. However, we must mention that recent analyses of fly ash from wet generation did not show needle formation either. Here, we could observe crystals in the form of hexagonal plates (see below). Both needles as well as hexagons are among the shapes that can be formed by CaCO3 (Kim et al., Journal of Materials Chemistry, 2009). We assume that the shape which the soluble components take upon drying, is influenced by slight changes in the relative humidity in our dryer unit. This might also be the case for spruce ash from wet generation. Potential hexagonal plates are difficult to distinguish from the irregularly shaped insoluble particles on the pictures because of the limited spatial resolution of the SEM (an example is shown below). Hence, we cannot rule out that CaO is responsible for the needle formation in case of the fly ash suspension just because we do not see any needles from the spruce suspension. This has been added to the manuscript (P11 L1-6).

Filtered fly ash suspension:                    Unfiltered spruce suspension:

[Figure]

[Figure]

4. I am not fully convinced that the particles produced through the wet system are less efficient. The authors conducted a direct comparison of the ice nucleating abilities of the wet and dry generated particles; however, it is necessary to demonstrated that the monodispersity of the 300 nm particles from both system is comparable.

Even though we did not perform the multiple charge correction in the case of the suspension samples, our results show clearly that the ash particles from wet generation must be less IN efficient than the particles from dry generation. Accounting for multiply charged particles would only lead to a further decrease in $f_{ice}$ and $n_s$, respectively (this has been stated in the original manuscript, P11L13-14). However, we understand your concerns and attempted to estimate the effect a large number of multiple charges could have on $f_{ice}$. By selecting 300 nm particles for the experiments, which is far to the right of the maximum of the size distribution, we already assured that a majority of multiply charged particles is not possible. To estimate the multiple charge fractions in the suspension measurements, we weighted the bipolar charge distribution (Wiedensohler, 1988, Journal of Aerosol Science), i.e., the probability of the particles to receive one, two or three negative charges in the neutralizer, with the measured size distributions of the ash-water suspensions. However, there is a caveat to this estimate, as crystals will probably have occurred in the size distribution measurements as well. It turned out, that the highest multiple charge fractions would probably occur for the fly ash suspension (+US), where we calculated 80.5 % singly, 16.8 % doubly, and 2.7 % triply charged particles. The multiple charge fractions were even lower for the other suspension samples. Would we perform the multiple charge correction using these fractions, our measured data would be reduced by a maximum factor of 2 only. A short version of this was added to the manuscript (P15 L10-16). Having a look at the SEM pictures of the suspension particles shows us, that we collected a rather monodisperse aerosol (apart from the needles in the case of fly ash). Large particles in the size range of doubly or triply charged particles are nearly absent (see figures below), telling us that the multiple charge fraction is likely even lower than the one from our example calculation.

Spruce suspension:                              Fly ash suspension:

[Figure]

**Specific comments:**

- Page 1, line 20: The Hallett-Mossop (1974) study introduces one of the multiple secondary ice formation mechanisms only. I suggest using a better reference such as Heymsfield and Willis (2014): Heymsfield, A. J., and Willis, P. (2014), Cloud conditions favoring secondary ice particle production in tropical maritime convection. J. Atmos. Sci., 71, 4500–4526, doi:10.1175/JAS-D-14-0093.1.

Done.

- Page 1, line 21: I suggest replacing this reference with a more appropriate one.

We now cited Rosenfeld and Woodley, Nature (2000), Koop et al., Nature (2000), Murray et al., Phys. Chem. Chem. Phys. (2010)

- Page 2, line3: provide pages from P&K or the actual paper(s).

We now cited Szyrmer and Zawadzki, BAMS (1997)

- Page 2, line 5: Cite the studies here.

Done.

- Page 2, line 12: "coal ashes contribute a major proportion of anthropogenic aerosol emissions" please provide the reference and number (estimate).

Since we cannot provide an estimate, we now only state that ash is the primary coal combustion product and cite a report of the US Geological Survey (Kalyoncu and Olson, 2001).

- Page 2, line 15: "yields global annual emissions of 30 Mt" is this comparable to mineral dust? Please provide the mineral dust annual emissions.

We added that global annual dust emissions are estimated to vary between 700 and 3000 Mt/a. This is taken from Textor et al., ACP (2006).

- Page 2, line 17: delete "however".

Done.

- Page 2, line 27: This needs to be better organized. The authors jump between old and recent studies back and forward.

We omitted the first reference to Umo et al., 2015. Now the studies are in chronological order.

- Page 2, line 32-34: "it could be shown that water soluble components were responsible for differences in the ice nucleation ability of fly ash samples from different power plants." Provide reference here.

Done.

- Page 3, line 5: "Umo et al. (2015) assumed that the different" Did they assume or did they provide evidence of?

There is no evidence provided. We replaced "assume" with "suggest", otherwise the sentence was not altered.

- Pages 3-4, lines 32-1: "the number of particles with three or more negative charges was negligible". Please report the fractions or percentages of the multiple charged particles.

A table with the doubly charged fractions was added in the manuscript (Tab. 1). We replaced "negligible" with "smaller than 1 %" to be more precise.

- Page 5, line 7: why 200 nm? What did not the authors try a smaller size to ensure that insoluble particles are not present? Please provide the resulting size distributions of the atomized filtered solution.

Filtering down to sizes as small as 100 nm would be possible but would require more effort. We cannot rule out that insoluble particles smaller than 200 nm passed through in the process of filtering. The SEM images of the filtered ash suspension indicate that the fraction of insoluble particles is low. Below, we added a comparison of the size distributions of the filtered and not filtered fly ash suspensions. It can be seen that a significant fraction of particles larger than 200 nm has been removed. Those that are still there are formed from dissolved material.

Also, LACIS measurements of the filtered fly ash suspension yielded f_ice values close to what was found for ammonium sulfate particles. This shows us that the particles from the filtered suspension only have very little, if any, IN activity, and this is the point we wanted to make. The LACIS measurements of the filtered fly ash suspension, as well as the ammonium sulfate measurements, have been added in the manuscript (Fig. 3d).

[Figure]

- Page 5, line 9: Why was not beech bottom ash particles analyzed by AA?

The beech ash sample was provided late in the course of our experiments. At this point, the other samples had already been analyzed in Sweden. However, we did not want to omit the quite interesting data. We added an explanation in the manuscript (P6 L24-P7L1).

- Page 5: What are the uncertainties associated with the data presented in Figure 1?

We decided to show the results of the atomic adsorption spectroscopy measurements, which are presented in Fig. 1, in the form of a table as well (Tab. 3). Here, the uncertainties are reported for each sample and each investigated compound.

- Page 5, lines 24-30: I am wondering how consistent and how reproducible is the data obtained from the dry generation system. Can the authors provide the size distributions obtained with the dry system separated by 15 or 30 minutes? Was it the multiple charged particles checked continuously with the UHSAS?

UHSAS measurements were not performed continuously during the course of the experiments as the instrument was not available over the entire period. However, the multiple charge fractions were checked on several occasions for each of the five samples. Therefore, particles were generated in the same way as for LACIS measurements and sampled for several minutes. We included a time series of a 35 min UHSAS measurement below. During this time, the fraction of multiply charged particles does indeed increase. As our measurement time with LACIS is in the range of 20 minutes, we are certain that a comparable amount of multiply charged particles was led into the flow tube as was determined in the UHSAS measurements.

We do not see a need to include this graph in the manuscript as it does not contribute to a better understanding of the presented results.

[Figure]

- Page 5, lines 33-34: Can the authors provide an activation scan with the filtered solution to confirm this?

CCNc measurements with the filtered fly ash sample have been performed. We added the following to the manuscript (P11 L17-21): "CCNc measurements with particles from the filtered fly ash suspension indicated a rather low hygroscopicity (kappa = 0.06 +/- 0.01). However, this does not necessarily mean that the components in the generated particles are not soluble. Sullivan et al. (2009) give a value of kappa = 0.011 for CaCO3, which is weakly soluble (Plummer and Busenberg, 1982). The generated particles could hence be composed of a mixture of CaCO3 together with other compounds.".
We also added a sentence concerning the CCNc in the methodology section (P5 L3-4).

- Page 8, lines 8-10: Which fraction of particles passed through? Was this confirmed with the size distributions from the UHSAS?

From the UHSAS measurements of the filtered fly ash suspension we cannot derive the fraction of insoluble particles left after filtering. This is due to the overlapping signal of the two particle populations. On the SEM images of the filtered suspension, however, we do not see any other particles than the crystals. We could detect an ice fraction of 25 % at -35 °C for the unfiltered ash suspension (this is the value we obtained when accounting for the occurrence of the crystals, i.e., only for the non-crystalline particles). For the filtered suspension, we only detected 1.3 % ice fraction which is close to our detection limit. This tells us a) that there cannot be a lot of the insoluble particles in the filtered suspension (because these were found to be IN active), and b) that the crystals are not very good INPs.

- Page 8, lines 15-16: Why the data from the water soluble material remaining in the filtered ash-water suspension is not shown?

We changed Fig. 4 from the original manuscript in so far as that it is a multiple panel figure now (Fig. 3). This gives us the opportunity to show the requested data, as well as the ammonium sulfate data points (Fig. 3d) without overcrowding the plot.

- Page 10, lines 3-4: "There is a trend of beech bottom ash being the most effective". I am wondering why out of the three wood samples beech was the least studied (although it was the most effective at nucleating ice)?

See our answers above (P2 and P5 of this document).

- Page 10, lines 15-18: "The fact that the wood ashes contain significantly more K than the brown coal ashes, which in this case is soluble, is a possible reason for the lower ice activity in comparison to the brown coal ashes. According to this, insoluble K could be the decisive element determining the freezing behavior of the brown coal ashes" What about CaO?

According to our hypothesis, the CaO in both wood and coal ashes would react with water to form Ca(OH)2. This would happen both in the suspension and as well as for dry particle generation (after activation to cloud droplets in the flow tube). However, if the larger CaO content in the wood ashes really is the reason for them being less IN efficient, then why would the coal fly ash be more efficient than the bottom ash, even though it contains more CaO? This is not logical from our point of view and hence we did not mention CaO as a possible factor in the original manuscript.

- Page 10, lines 20-22: "the brown coal ash particles might be more efficient at nucleating ice because of surface defects such as lattice dislocations caused by impurities or crystallographic dislocations." This is not clearly supported by the presented data. I am wondering why the authors did not provide SEM images for the bottom wood ashes to compared the surface defects with those of the bottom coal ash particles.

An SEM image of spruce particles from wet generation was included (Fig. 2c). Unfortunately, we cannot make clear statements about the surface defects because of the limited spatial resolution. This is why we formulated our hypothesis in the form of a speculation. We do not know if this really is the case, but we wanted to share our thoughts with the reader.

- Page 11, Lines 1-3: "It has been shown that certain types of amorphous particles are able to nucleate ice (Murray et al., 2010b; Wilson et al., 2012), but it remains to be examined whether the amorphous components in fly ash are ice active as well". This only happens at very low temperatures relevant to cirrus clouds.

Amorphous fly ash particles are likely composed of aluminosilicate glass (Ramsden and Shibaoka, Atmospheric Environment, 1984; Querol et al., Atmospheric Environment, 1996). This has been added to the manuscript (P14 L17-19). Unlike the glassy SOA particles which were investigated in the references you cited, glass is stable at temperatures higher than -38 °C (even at temperatures higher than 0 °C). Hence, in case the amorphous particles in the fly ash cause IN, we assume that this would happen at temperatures higher than the homogeneous freezing limit.

- Page 11, lines 4-6: "Tab. 1 and Fig. 4 additionally show the 5 parameters and fit curves to measurements with K-feldspar and mineral dust particles (K-feldspar, Arizona Test Dust, NXillite, Fluka kaolinite) coated with sulphuric acid (clay mineral baseline, Augustin-Bauditz et al., 2014). I may have missed but I could not find the Arizona Test Dust, NXillite, and fluka kaolinite data.

For the data presented in Augustin-Bauditz et al. (2014), K-feldspar, ATD, illite and kaolinite particles were coated with sulfuric acid. Their immersion freezing behavior was then investigated with LACIS. It turned out that all coated particles featured the same freezing behavior, even though they differed from another before coating. Because the data was on the same line, and because weathering feldspars turn into clay minerals, it was proposed to call this line the "clay mineral base line". This was added to the manuscript (P14 L30-P15 L1). Only this line is shown in Fig. 3, not the results of the individual dusts. We hope that this brings more clarity.

- Page 11, lines 26-30: "By counting ~ 900 particles on SEM images, it was determined that ~ 78 % of all particles are crystals. This value may be smaller in the flow tube as the fragile crystals could break upon impact on the filter leading to a multiplication. As only 22 % of the droplets contained a spherical fly ash particle during the experiments with the suspension sample (+US), the original data was corrected by a factor of 1/0.22 = 4.54 which is also shown in Fig. 4 for a direct comparability to the ice nucleation ability of dry particles from the same sample." How confident are the authors about this calculation? What is the uncertainty associate to it? 900±?? 78%±??, 22%±??, 4.54±??

These numbers were determined by counting the number of crystals and spherical particles on an SEM image. In doing so, particles that have been counted were marked so that there is no possibility for double counts. In case you are alluding to the statistical significance of this estimate, we now included the confidence interval for the 95 % confidence level, which is +/- 3% of the given values. This means that the actual fraction of needle shaped crystals is between 75% and 81% with a probability of 95%. This is now included in the manuscript (P15 L27-28).

-  "which is a clear lowering of the ice nucleation activity by a factor of 4 compared to dry particle generation, i.e., suspending the particles in water reduced their ice activity in the temperature range below -31 °C." I am not sure how valid is to directly compared the ice nucleating abilities of the ash particles generated from the wet and dry systems given that those obtained from the wet generation are not corrected for multiple charged particles.

Here we would like to refer to the estimate given above (P3 of this document). Accounting for the multiple charges in the experiments with wet particle generation would only lead to a further lowering of f_ice and n_s. After the cited sentence we added "Note that this lowering might be larger depending on the multiple charge fractions in case of wet particle generation." (P16 L4).

- Page 12, lines 4-13: This is an interesting observation. I am wondering why the authors did not further expand this with other samples (e.g., bottom ash brown coal).

In case of the other ash suspensions (brown coal bottom ash and spruce bottom ash), the result are already close to the LACIS detection limit. As omitting the ultrasonic treatment only makes the samples less IN efficient, we would not expect to see a significant difference for these samples. Also, this behavior has been observed for soil dust suspension particles (cited in the original manuscript P12L10).

- Page 12, lines 4-5: "the fice values of the fly ash suspension which was not put in the ultrasonic bath are clearly lower than those of the fly ash suspension with ultrasonic treatment." Here and along the results section, how many scans were conducted for each sample for each specific set of conditions? How reproducible are they?

As stated in the caption of Fig. 4 on P9 of the original manuscript, at least three measurement were performed in case there is an error bar on the data point. In case there is no error bar shown, we performed one or two measurements (added on P11 L30-32). For each measurement at least 2000 particles were detected by our optics. LACIS measurements are very reproducible due to this large number of counted particles and the small temperature uncertainty (added on P5 L21-22). But they are also very time consuming which is why we need to consider costs and benefits of repeating measurements.

- Page 14, lines 1-14: This part is too speculative with many unsupported assumptions. This should be deleted.

We revised this section in terms of citing more field observations of fly ash particles and calculating the in-stack concentration especially for the power-plant Lippendorf. Even though the section contains a lot of speculation, we would like to keep it in the manuscript. Firstly, we clearly state when we make a certain assumption and secondly we think that this estimate supports the relevance of the topic.

Figures:
- Be consistent with the labels (a and b, or 1 and 2)

Done.

- Figure 1. I am not sure how useful is panel 2.

We would like to include the second part of the analysis, as this actually shows measured fractions of the investigated elements, whereas the first part was estimated by recalculating the measured concentrations of major ions into their most common oxide forms. The second part hence provides valuable information on top of the calculated composition.

- Figure 3: It would be nice to have a similar image of the wet generation system of a bottom ash sample. Additionally, Fig. 1 indicates that Wood ashes contain much more CaO than the coal ashes. Therefore, the needle crystals should be more pronounced in the wood ashes if the reasoning presented here is correct.

An SEM image of spruce ash suspension particles has been added to the manuscript (Fig. 2c). As stated earlier the non-existence of needles on this image does not necessarily disprove our hypothesis. At one occasion, we also observed hexagonal plates on the SEM images of fly ash particles from wet generation. This could also be the case here and we cannot distinguish the plates from the insoluble spruce ash particles due to the limited spatial resolution of the SEM.

References cited in our answers:

Augustin-Bauditz et al., GRL (2014)
Kalyoncu and Olson, US Geological Survey (2001)
Kim et al., Journal of Materials Chemistry (2009)
Koop et al., Nature (2000)
Murray et al., Phys. Chem. Chem. Phys. (2010)
Plummer and Busenberg, Geochimica Et Cosmochimica Acta (1982)
Querol et al., Atmospheric Environment (1996)
Ramsden and Shibaoka, Atmospheric Environment (1984)
Rosenfeld and Woodley, Nature (2000)
Sullivan et al., ACP (2009)
Szyrmer and Zawadzki, BAMS (1997)
Textor et al., ACP (2006).
Umo et al., ACP (2015)
Wiedensohler, Journal of Aerosol Science (1988)

---

## Author Comment (AC3) · 12 Aug 2016

We thank the referee for the useful comments, all of which we considered carefully. In the following, we respond to them separately, where the original comments are colored, while the answers are given in black.

Review of "The immersion freezing behavior of ash particles from wood and brown coal burning" by Grawe et al.

In this study the authors examine the immersion mode freezing efficiency of combustion ashes from different woods and brown coal burning using LACIS. Ashes from brown coal burning are found to exhibit higher nucleating abilities than those from the ashes generated from wood burning. The results presented here also seem to indicate that sample preparation can have impacts on ice nucleation efficiencies; an important point which will need to be considered in future studies.

My major comments below surround increasing the specificity and clarity of statements made. In particular, there are some vague statements made in attempting to account for observations in this work, and how it compares to others such as Umo (2015). Sentence and paragraph structure can also at times make it difficult to make out what is being said without multiple rereads of certain passages. While I recognize that investigating the nature of nucleating sites in a mixture as complex as ash is challenging, I suggest the authors could discuss the difficulties surrounding this endeavor and limit sweeping statements.

I am of the opinion that, following careful consideration of the points below and improvement of the manuscript in the areas listed, this could be accepted in ACP.

Main comments:

- The authors need to evaluate the use of 300 nm particles in this paper. It can be anticipated that physical and chemical composition varies with particle size, and in turn, perhaps the ice nucleating efficiencies. Without further experiments on larger particles it is difficult to see how the results of this study can be generalized to larger particles. I suggest that at the points in the manuscript where the authors are referring to size selected particles, they explicitly state this for clarity (e.g. section 3.1 L1-8).

From size distribution measurements during coal combustion, we know that there is a bimodal distribution: one submicron peak below 0.1 µm and one broad supermicron peak in the range from 3 to 50 µm (Damle et al., 1981, Aerosol Science and Technology 1:1). As particle collection techniques are less efficient for submicron particles (Flagan and Seinfeld, 1988, Fundamentals of Air Pollution Engineering), we decided to select particles smaller than 1 µm. 300 nm particles were chosen because we could get sufficiently high and stable number concentrations from the variety of samples when selecting this size (the mode diameters vary from 130 to 200 nm for dry particle generation). While there are a lot of different particle types included in bulk fly ash (i.e., Ramsden and Shibaoka, 1982, Atmos. Environ. 16, 9), it has also been found from single particle analysis that the chemical composition of fly ash particles with aerodynamic diameters between 0.2 µm and 4.8 µm is remarkably consistent (Kaufherr and Lichtman, 1984, Environ. Sci. Technol., 18). Hence, we would not assume to see a difference in n_s, would we select another particle size.

In case of bottom ash, exemplarily, measurements were performed with 500 nm brown coal bottom ash particles which showed no difference in n_s within the range of our measurement uncertainty compared to the measurements with 300 nm particles (see below, now also added in Fig. 4 as green triangles). As LACIS measurements are very time consuming and we do not expect

to see a difference in n_s in the size range we are able to select, we would like to avoid performing further experiments with other particle sizes.

In the manuscript, a paragraph explaining why 300 nm were chosen was added (P4 L17-28).

[Figure]

- P2L12: "As a result, coal ashes contribute a major proportion…..". This strong statement needs a reference.

Since another referee requested an estimate, which we cannot provide, we now only state that ash is the primary coal combustion product and cite a report of the US Geological Survey (Kalyoncu and Olson, 2001) on P2 L16-17.

- P2L13-15: On the one hand, "the importance of ash particles as potential INPs must be put into perspective by comparing with concentrations" yet in the next sentence an emission rate is given, not a concentration. While both sentences on their own are fine, having these sentences one after the other could be misleading.

This was reworded to: "The impact of ash particles as potential INPs must be put into perspective by comparing ash emission rates to those of other INP containing aerosols, e.g., mineral dust…" (P2 L18-19)

- P4L6: Why is this important? Reference to publications demonstrating that surfaces typically used for ice nucleation assays interfere with the nucleation process would seem to be appropriate here, bearing in mind that many studies have been performed where droplets are supported by substrates and this does not appear to be an issue.

The half sentence in question was omitted.

- P4L7-8: This statement also needs referencing and elaboration: what size droplets are you speaking of? Homogeneous nucleation CAN be probed in picolitre sized droplets on cold stages.

We changed the sentence to: "Furthermore, as water is brought into the system via the gas phase, impurities which are known to cause the freezing of pure water droplets above the homogeneous freezing limit on cold stages (Budke and Koop, 2015, AMT; Whale et al., 2015, AMT), can be ruled out for our experiments." (P5 L9-11). In citing these two studies, it is made clear that we are not relating to picolitre sized droplets.

- P4L24: only three woods are investigated in this study; is this representative of all "deciduous vs. coniferous" trees?

"Deciduous vs. coniferous" was omitted.

- P5L1-2: What power does this sonicator deliver to the sample?

The ultrasonic bath has a peak output of 320 W and is operated at 35 kHz. We did not include this information in the manuscript as it is not needed for a better understanding and none of the other referees asked for it.

- P5 L6-7: is there any effects based on how long the samples are left to stir for? Also, what about aggregation in solution? Could you be removing particles smaller than 200 nm by filtration that have formed aggregates over the course of 24 hours of stirring?

The aim of the filtration process was to remove insoluble material and leave a solution. So if particles smaller than 200 nm aggregated and were removed, this would actually be a positive effect (in terms of removing the insolubles). We did not study the effect of stirring duration as we wanted to prepare the samples in the same way as described by Umo et al. (2015, ACP). From our point of view, this does not necessarily have to be mentioned in the manuscript.

- P5L6: Define "ice activity"

We chose to avoid using this term and replaced it with "ice nucleation efficiency" which is more precise and, from our point of view, does not need further explanation.

- P5L8: Can the authors say anything on the efficiency of these filters and pore size distribution? Can particles larger than 200 nm sometimes make it through? Also, are the authors defining "water soluble material" as any particle which will make it through a 200 nm filter? This should be stated if so.

We do not have any information about the pore size distribution. The manufacturer only states that the filters have a "sharply defined pore size". We define anything coming through the filter as water soluble because we collected particles from this solution and only found crystals on the filters. However, you are right in saying that some insoluble material smaller than 200 nm might make it through. Those particles could also be inside the crystals so that we cannot identify them on the SEM images. We added the following sentence: "We are referring to the remaining compounds as "water soluble" even though we are aware that insoluble material smaller than the filter pore size of 200 nm might still be present. This is justified, because the selected 300 nm particles predominantly include soluble substances." (P11 L15-17).

- Figure 4: How is the "homogenous" region here determined? Citation of the appropriate paramaterization(s) should be added.

We define homogeneous nucleation according to the freezing behavior of highly diluted ammonium sulfate droplets which has been investigated with LACIS but was not shown in the original manuscript as we did not want to overcrowd the plots. However, we now changed Fig. 4 (now Fig. 3) to a 4 panel figure. This gives us the opportunity to show the ammonium sulfate data (Fig. 3d). From these measurements we see homogeneous nucleation becoming the dominant freezing mechanism below -38 °C. We added an explanatory sentence in the figure caption.

- Fig 4: State in the legend and text exactly what the "clay mineral baseline" is; i.e. what clays were used?

The used dusts were not all clay minerals. We included the following sentences in the manuscript for further explanation (P14 L27-P15 L1): "…measurements with an untreated feldspar sample (76 % microcline, K-feldspar, and 24 % albite, Na-feldspar, Augustin-Bauditz et al., 2014), […]. Also shown is the curve for different kinds of mineral dust particles (same feldspar sample, Arizona Test Dust, NX-illite, Fluka kaolinite) coated with sulfuric acid (Augustin-Bauditz et al., 2014). The coating caused the dusts to show a similar immersion freezing behavior even though differences were observed without coating, presumably due to different amounts of K-feldspar contained in the samples. Weathering feldspars turns them into clay minerals and it was argued in Augustin-Bauditz et al. (2014) that the coating had a comparable effect, i.e., consuming all feldspars in the different samples and leaving clay minerals only. Hence the line on which the data from all the different coated mineral dusts fell was termed the "clay mineral baseline"."

A short version of this is now also included in the caption of Fig. 3 (former Fig. 4). We did not add an explanation in the legend as this would overcrowd the plot.

- Fig 4: Harrison et al. ACPD (2016) show that K-Feldspar can have different activities: specify exactly which K-Feldspar was used.

The used sample is not pure K-feldspar but a mixture of 76 % microcline (K-feldspar) and 24 % albite (Na-feldspar). This was determined by X-ray diffraction measurements (Augustin-Bauditz et al., 2014, GRL). This was added to the manuscript (P14 L28-29) and the caption of Fig. 3.

- P10L7-8: I don't have an issue with this statement as such, but I think it's important to point out that onset freezing temperatures cannot be used for others to compare with. A simple phrase along the lines of "For within these experiments we can compare our onset freezing temperatures, which…."

Done.

- P10L9-13. This section is vague, and should be strongly caveated by noting that the authors don't know what the explanation for the differing nucleation activities is, or further experimental evidence showing that SiO2 is indeed responsible for this behaviour should be added. For both this point, and the point below, a statement explaining why it is difficult to conclusively identify the active site in such a complex sample at the beginning would definitely help here!

Such a statement has been added at the beginning of the paragraph (P14 L1-3): "To date, there is no experimental evidence on the ice-nucleation-determining compound in ashes, presumably because it has rarely been studied and because it is a very complex mixture (Ramsden and Shibaoka, 1982; Umbria et al., 2004; Zhang et al, 2011).". As we cannot provide evidence to our SiO2 hypothesis on a time scale of weeks, we adopt your first proposal by saying that: "…even though this holds true for the ashes presented here, a larger number of samples would have to be investigated to make a conclusive statement." (P14 L10-12).

- Page 10L 14- Page 11L3: All of this section is very speculative and at points vague, and I suggest it be rewritten to make it more concise or removed. After providing several possible explanations for their observations, the authors note "other properties such as ice active sites on the particle surface might affect the ice nucleation ability of a substance." What does this mean? Taking the Vali (2015) definition of a site being a "Preferred location for ice nucleation on an INP", this statement is rather meaningless.

We agree that the cited sentence does not add to the paragraph. It was hence removed. We adapted the point about the potential influence of K on the immersion freezing behavior to be more precise

(P14 L14-17). Also, the point about the potential effect of amorphous components in the fly ash is now discussed in more detail (P14 L21-26).

- Page 11 L5: again, which K-Feldspar was used?

This was specified.

- Page 11 L6: Is it reasonable that clay minerals treated with sulfuric acid are termed the "clay mineral baseline"? Surely this should be justified. Also, if justified, what it is should be stated explicitly in figure 4.

We added further explanation. See P4 of this document.

- Page 11 L17-page 12L3: There are a lot of concepts in this rather long paragraph: separating into at least two smaller paragraphs would help readability greatly

Done.

- P12L1-3: If it's not the physical or chemical particle properties changing and leading to a destruction of active sites, what else could it be? This non-specific nature of this statement makes it vacuous. The word "likely" doesn't help here either.

This sentences has been reformulated. We also added some additional information (P16 L6-12): "These differences are, as for brown coal bottom ash but apparently not for spruce bottom ash, related to a change of physical and/or chemical particle properties due to the change in particle generation. A change in particle composition has been observed before for mineral dust particles which featured different hygroscopicities when being generated firstly via dry dispersion and secondly via atomization (Herich et al., 2009; Sullivan et al., 2010). It was assumed that soluble material present in a fraction of the dry particles was redistributed across all particles contained in the droplets as a coating (Herich et al., 2009). Sullivan et al. (2010) state that changes in surface structure and chemistry from dry to wet particle generation might not only affect the hygroscopycity but also the ice nucleation behavior of the particles."

- Section 3.3: Discussion of possible differences in composition and hence ice nucleating abilities due to size selection of particles should be presented here.

This is fair point. In Sec. 2, we stated that the composition of fly ash particles is consistent in a size range from 0.2 to 4.8 μm (Kaufherr and Lichtman, 1984, Environ. Sci. Technol.) and that we do not see a difference in n_s between the selection of 300 nm particles and 500 nm particles for brown coal bottom ash. However, much larger particles (up to 200 μm, mode diameters at 10 μm) are included in case of the experiments by Umo et al. (2015, ACP). We added the following (P19 L3-9): "Differences in ice nucleation efficiency between our samples and those investigated by Umo et al. (2015) could be related to differences in composition due to size selection. In our case, the immersion freezing behavior of 300 nm particles was investigated, whereas the suspensions examined by Umo et al. (2015) contained much larger particles (average volume-equivalent diameters of 10 μm and 8 μm for coal fly ash and bottom ashes, respectively). This might be relevant as there are studies indicating that the trace elemental composition in fly ash is inversely proportional to the particle size in the supermicron range and not strongly size dependent for submicron particles (Davison et al, 1974; Smith, 1979)."
The last sentence of this paragraph from the original manuscript ("In addition to the stated possible reasons, the fact that our samples were not completely identical to the samples investigated by Umo et al. (2015) concerning their chemical composition and morphology might contribute to the observed differences in n_s.") was deleted.

- Page 13L12:  Agreed, but explain why you would not expect it. Reference where appropriate

We changed the sentence to: "Although a time-dependence of the nucleation process has been observed before (Ervens and Feingold, 2012; Welti et al., 2012; Wex et al., 2014), this effect is too small to describe the here found discrepancies." (P17 L34-35).

- P13L13-14: Again, vague. Is there nothing more which can be said here other than the samples were not "completely identical"

We added the following (P18 L5-P19 L2): "There are recent studies investigating the effect of different experimental methods on the ice nucleation behavior of mineral dust particles where most cold stage methods yielded lower $n\_s$ values than dry dispersion methods (Hiranuma et al., 2015, ACP, Emersic, 2015, ACP). These findings could be relevant for ash samples as well. Hiranuma et al. (2015) argue that a high degree of agglomeration in the dry-dispersed particle measurements leads to a larger surface area being exposed to liquid water and consequently larger $n\_s$ values in comparison to the rather de-agglomerated suspensions. On the other hand, Emersic (2015) present the hypothesis that particles may coalesce in suspension leading to a reduction of the surface area available for ice nucleation."

- P14 L26: All clay minerals? Which K-Feldspar?

Specified.

- Typos/other:

- P3L1: Is the cold stage(s) used by Umo (2015) the same as that used in Murray 2010(a)?

The cold stage used for the experiments by Umo et al. (2015) is described in Murray et al. (2010a). We added Whale et al. (2015, AMT) where additional information concerning the setup is provided (P3 L10).

- P1L3: Largest rather than biggest

Done.

- P8L26 and fig 4. SMB should be SBM no?

Yes. This was changed.

References cited in our answers:

Augustin-Bauditz et al., GRL (2014)
Budke and Koop, AMT (2015)
Damle et al., Aerosol Science and Technology (1981)
Davison et al., Environ. Sci. Technol. (1974)
Emersic et al., ACP (2015)
Ervens and Feingold, ACP (2012)
Flagan and Seinfeld, Fundamentals of Air Pollution Engineering (1988)
Herich et al., Phys. Chem. Chem. Phys. (2009)
Hiranuma et al., ACP (2015)

Kalyoncu and Olson, US Geological Survey (2001)
Kaufherr and Lichtman, Environ. Sci. Technol. (1984)
Murray et al., Phys. Chem. Chem. Phys. (2010)
Ramsden and Shibaoka, Atmospheric Environment (1984)
Smith et al., Environ. Sci. Technol. (1979)
Sullivan et al., Aerosol Science and Technology (2010)
Umbría et al., Atmósfera (2004)
Umo et al., ACP (2015)
Welti et al., ACP (2012)
Wex et al., ACP (2014)
Whale et al., AMT (2015)
Zhang et al., Chinese Science Bulletin (2011)

---

## Author Comment (AC4) · 12 Aug 2016

We thank the referee for the useful comments, all of which we considered carefully. In the following, we respond to them separately, where the original comments are colored, while the answers are given in black.

**Referee comments for manuscript acp-2016-208 by Grawe et al. "The immersion freezing behavior of ash particles from wood and brown coal burning"**

**Summary:**

The manuscript presents results obtained at the Leipzig Aerosol Cloud Interaction Simulator, LACIS, on the ice nucleation behavior of ash particles from wood and coal combustion. The chemical composition of the different ash samples was obtained from atomic absorption methods. The main findings were that fly ash from coal burning was the most ice-active sample and was the sample with the highest fraction of $SiO_2$. Furthermore, the effect of differences in the particle generation method (dry vs. wet, treating the sample in an ultrasonic bath vs. not using it) was investigated. It was found that wet generation reduces the ice nucleation ability of the ash particles compared to the dry generation method. Particles treated in an ultrasonic bath had a higher ice nucleation ability than non-treated wet generated particles.

Ice nucleation induced by ash particles has received only little attention by the scientific community. The presented measurements are therefore of interest to the readers of *Atmospheric Chemistry and Physics*. Also the investigation of the influence of the generation method on the results is interesting. However, the manuscript and particularly the discussion of the results is often written in a vague and imprecise way and hardly adds to our understanding of the underlying ice nucleation process. I thus only recommend publication after addressing the comments below.

I apologize for any overlap with the other reviewers' comments.

**Major Comments:**

*Introduction:*

– The literature summary on p.2 l.27ff lacks depth. Who measured what, at which conditions? What type of bottom ash did Murray et al. (2010a) investigate? Regarding the first lines of page 3 it is not clear to me which results were found by Umo et al. (2015) and which by Murray et al (2010a).

The cold stage used for the experiments by Umo et al. (2015) is described in Murray et al. (2010a), this is why the reference was added after "cold stage setup". We added Whale et al. (2015, AMT) where additional information concerning the setup is provided. The structure of the sentence was changed.

– Since the study is very similar to that by Umo et al. (2015), the earlier study should be described in more detail and the novelty of the present study should be described. What sizes did Umo et al. (2015) investigate? Which surface area? Already describe the freezing method here. And convince the reader why there is a need for your study.

We added that µL and nL droplets of 0.1 wt% ash in water were investigated by Umo et al. (2015), either by nebulizing droplets onto a glass slide or by pipetting (P3 L10-12). Total surface areas per droplet are not given explicitly in the Umo et al. (2015) paper which is why they are not mentioned here. We also added that there is a need for further investigations as there are only few measurements concerning the IN efficiency of ash particles which is why it is important to better understand the effect of sample origin and composition (P3 L20-21). Furthermore we were able to use different kinds of particle generation and sample preparation which has not been investigated by Umo et al. (2015).

– One could also add motivation from field observations of biomass burning, e.g.: Prenni et al. 2012, GRL; McCluskey et al. 2014, JGR

We added the suggested references on P2 L6, following the citation of Petters et al., JGR (2009).

*Methods:*

– The representativeness of 300 nm particles is not shown or even mentioned. Why were particles of this size selected? How do the size distributions of the investigated ashes look like? Is 300 nm the mode diameter or at least close to it? Can it be assumed that larger/smaller particles have a similar composition and is the ice nucleation ability expressed as ice-active surface site density thus potentially scalable or are changes to be expected with size?

From size distribution measurements during coal combustion, we know that there is a bimodal distribution: one submicron peak below 0.1 µm and one broad supermicron peak in the range from 3 to 50 µm (Damle et al., 1981, Aerosol Science and Technology 1:1). As particle collection techniques are less efficient for submicron particles (Flagan and Seinfeld, 1988, Fundamentals of Air Pollution Engineering), these will be more abundant in the atmosphere, and we decided to select particles smaller than 1 µm. 300 nm particles were chosen because we could get sufficiently high and stable number concentrations from the variety of samples when selecting this size (the mode diameters vary from 130 to 200 nm for dry particle generation). While there are a lot of different particle types included in bulk fly ash (i.e., Ramsden and Shibaoka, 1982, Atmos. Environ. 16, 9), it has also been found from single particle analysis that the chemical composition of fly ash particles with aerodynamic diameters between 0.2 µm and 4.8 µm is remarkably consistent (Kaufherr and Lichtman, 1984, Envirn. Sci. Technol., 18). Hence, we would not assume to see a difference in $n\_s$, would we select another particle size.

In case of bottom ash, exemplarily, measurements were performed with 500 nm brown coal bottom ash particles which showed no difference in $n\_s$ within the range of our measurement uncertainty compared to the measurements with 300 nm particles (see below, now also added in Fig. 4 as green triangles). As LACIS measurements are very time consuming and we do not expect to see a difference in $n\_s$ in the size range we are able to select, we would like to avoid performing further experiments with other particle sizes.

In the manuscript, a paragraph explaining why 300 nm were chosen was added on P4 L18-28.

[Figure]

– The description of the wood samples is very vague. What was burned? Just the wood or in addition bark, leaves, needles, beechnuts…? Which part of the wood? How old was the wood (i.e. freshly cut, stored for a long term). What type of spruce, birch, and beech? Have the wood samples been treated in any way before burning? Are these commercially available fuels?

All wood ashes were burned in private fireplaces, and the detailed information you ask for is mostly not available. As the conditions were different, it is astounding that the wood ashes are all very similar in terms of their IN ability. This supports that wood bottom ash in general might be similar in its IN activity, and we do not see a reason to go that far into detail. In the manuscript, we added that no leaves or small branches were included and that the wood was stored for drying prior to the combustion process (P6 L6-8).

– The disagreement between the measurements with the UHSAS and the observations of multiple charged particles on the SEM filters is disturbing. I am glad this is discussed in the manuscript but I encourage the authors to put more effort in finding the reasons for this discrepancy. That the filter analysis is not representative appears as an oversimplified argument. If this is the case then the analysis should be done in a way that it is representative, e.g. analyze a larger area of the filter. How representative are other conclusions drawn from the filters?

The multiple charge correction was performed based on size distribution measurements with the UHSAS (optical diameter). For this, 300 nm particles were generated in the same way as for LACIS measurements and sampled for several minutes. The filter samples, however, were collected over several hours to assure a sufficiently high concentration on the filter surface. New UHSAS measurements (see below) show that the cyclone, which we used to minimize multiply charged particles, is filled after around 30 minutes, so that more larger particles are detected than in the beginning of the UHSAS measurement. As LACIS measurements typically last around 20 minutes only (due to wall glaciation effects), and the cyclone was cleaned after each LACIS measurement, we are certain that the UHSAS determined multiple charge fractions (now in the manuscript as Tab. 1) correspond to the actual fractions in the flow tube. The large particles which could be seen in the original manuscript as Fig. 2 a) and b) have accumulated during the long collection time for the filter, during which the cyclone was not cleaned often enough. Following your suggestion, we did additional tests, where we added an optical particle measurement for the detection of super-micron particles (APS, aerodynamic particle sizer, TSI), and indeed saw that while there were no super-micron particles visible during the beginning of the particle generation (~30 min), these particles became more and more abundant afterwards. Based on that, we decided to collect fly ash particles from dry generation on filters once again, now cleaning the cyclone every 30 minutes. The new SEM pictures (see examples below, another one was added to the manuscript as Fig 2 b) show a majority of particles in the size range of 300 nm. 68 of 84 counted particles were in the size range of around 300 nm, while 16 particles classified as particles larger than 500 nm. This might not be significant from a statistical point of view, but it is similar to the findings of the UHSAS measurements, where on average 19 % of all particles were identified as doubly charged. This was derived from measurements as shown in the figure below for ~30 min of sampling, during which already some increase in doubly charged particles can be seen.

We omitted Fig. 2 b from the original manuscript to avoid confusing the reader with the occurrence of large particles which are only present on these particular filters but not in the LACIS measurements. Fig. 2 a was left in the manuscript, as we did not manage to collect sample the coal bottom ash again in the limited time. In light of the new observations, we changed the paragraph on P5L19-30 of the original manuscript accordingly (P9 L14-24).

[Figure]

Size distribution UHSAS
(20 s samples taken during a 35 min measurement)

SEM images of fly ash particles from dry generation:

[Figure]

*Results and Discussion:*

- p.13 l.5 ff: It is not clear to me why you state that "the shape of the nucleation spectrum of fly ash suspension particles from our measurements correspond well to the findings by Umo et al. (2015),…". The fly ash data points in Figure 5 to me look either mostly linear (open purple circles), consist only of 2 data points (open purple circles with cross) or have a plateau which starts at a temperature 8 K lower than the fit from Umo et al. (2015) (full purple circles). Keeping in mind that for a given particle size and concentration, n_s can only vary in a restricted range of 3-4 orders of magnitude in this case, the observed three orders of magnitude differences are massive.

We agree that 3 orders of magnitude in n_s are a massive difference. What we wanted to say with this sentence is that we observe a plateau in the temperature range between -24 °C and -32 °C which can also be seen in the measurements by Umo et al., 2015. We reformulated the sentence to avoid any misunderstanding (P17 L18-20).

- The correction factor 4.54 = 1/0.22 = 1/(1-0.78) is derived from the SEM images of the filters. It is already stated on p.11 l.27 that this value might be smaller in the flow tube due to less

fractioning of the crystals. However, in the discussion of the results it is not considered as an upper limit but rather as a realistic estimate. Especially for the comparison of n_s to Umo et al. (2015) it would be helpful to be aware that this value might be considerably smaller.

As you suggested, we now mention that the number is an upper estimate more prominently in the manuscript. Would the number of crystals be lower than in our estimate, f_ice would be lower as well. The extreme case would be no crystals at all, meaning that our measurements would not need correcting. Then the gray circles in Fig 3 d) would correspond to f_ice of the fly ash suspension particles. Translated to n_s, this would correspond to a reduction of less than an order of magnitude, i.e. still two orders of magnitude above the n_s values reported by Umo et al. (2015). A short version of this was added on P15 L32-33 and P17 L20-22.

*Atmospheric Implications:*

I would recommend deleting this section completely as the calculation is not convincing nor adds value to the manuscript. To my understanding the dilution factor of 1000 over a distance of 80 km derived from Parungo et al. (1978a) only applies to very particulate atmospheric conditions and does not seem representative for general atmospheric situations. It seems to me that the background aerosol concentration has not been considered in the derivation of the dilution factor. Furthermore, are the emissions of coal-fired power plants today the same as in the 70's? Without having discussed how representative 300 nm particles are for ash particles in general and how likely the composition and morphology and thus n s of 300 nm and 1000 nm particles are similar the given estimate is not convincing. A short comment on the atmospheric implication could be included in the summary part. However, I would suggest referring here to direct field observations of ash particles as ice nucleating particle or ice crystal residual.

This section was partly rewritten and now includes more references to field observations of fly ash particles. In the new version, we estimated the in-stack particle concentration especially for the power-plant Lippendorf instead of relying on measurements at other sites. We now consider the background aerosol concentration and clearly state that the dilution factor is valid for this specific set of meteorological conditions. As we selected 300 nm particles in our measurements, we now use the same size for our estimate.
Even though the section includes speculation, we would like to keep it in the manuscript. In our opinion, it contains valuable insights into the potential effect of fly ash emission on immersion freezing in clouds close to the source or in the power-plant plume itself.

*Language:*

It seems to me that there is an overuse of "could". It you are talking about your findings then the use of phrases like "was shown" or "was confirmed" instead of "it could be shown", "could be confirmed",… is more appropriate.
Done.

**Specific Remarks:**

p. 1 l.6: "lignite" is only mentioned in the abstract. If it is important to know it should be mentioned in the main paper body, if not it can be deleted.
We added that the coal ashes stem from lignite burning but will be referred to as brown coal ashes according to the generic term. This can be found on P6 L8-9.
p. 2 l. 3: Is "Pruppacher and Klett, 1997" truly the primary source? Give the original reference otherwise.
We now cite Szyrmer and Zawadzki, BAMS 78, No. 2, 1997 on P2 L5.

p. 2 l. 24: What is the chemical composition of mineral dust and ash that is very similar? Be more precise.

We added that "both include several common mineral components such as Si, Na, Ca, Fe and oxides" (P2 L29-30).

p.3 l.5ff: What other factors could determine the found differences between the two ashes? I can't think of any other factors, assuming same sized particles were used. Thus this sentence does not yield new information.

We included this sentence to point out to the reader that no clear reason for the differences in the IN ability of different ashes had been found in the earlier study. Nothing was changed.

p.4 l.13 "second to last section" is this correct? Is last section referring to the second topmost or lowermost section? What is the residence time in the ice nucleation part if droplet activation only takes place in the second to last section?

Here we do mean the second to last section counting from top to bottom, i.e., section 6 of 7. The nucleation time is 1.6 s. This was added in the manuscript (P5 L15-16).

p. 5 l. 12: delete "(quartz)" as the chemical composition analysis does not yield information on the lattice structure which defines the mineralogy.

Done.

p. 5 l. 12: How much is "more SiO2"? Be more precise!

Done.

p. 5 l. 15: Why is K in biomass burning ash water soluble in but not in coal ash? Shortly state the chemical reason behind that.

We added the following explanation on P14 L15-17: "K in wood ashes is present in the form of soluble salts and oxides, whereas coal ashes contain K in clay minerals with low solubility", citing Steenari et al., 1999, Fuel, 78. In the course of this, we changed the previous citation of Andreae et al., 2004, Science, 303 (original manuscript P5 L16) to Steenari et al., 1999, because the reasoning is presented more clearly in the latter.

p. 6, Figure 1: Do the mass fractions add up to 100 %? It appears to me that not. What is the composition of the remaining material? If this is not known, the exact amount of how much is unknown should be stated, given that the frozen fraction often is for most samples below 10 %.

In addition to Fig. 1 the manuscript now includes a table with the corresponding values (Tab. 3), where the sum of the analyzed components is shown. You correctly observed that the given values do not add up to 100 %. This is due to the analysis procedure: It is standard to recalculate the concentrations of the ten major ions into the mass of their most common oxide forms, even though they may have had other counterions in the original sample. In some cases the compounds are indeed oxides, but we do not know the counterions in each case. K for example would be more likely to occur as KCl, K2CO3 etc., than as K2O. Any missing percentage is due to other than these ten elements and the fact that other counterions would have been involved. This has been added to the manuscript (P7 L2-P9 L2).

p. 6: caption Figure 1: Replace "minerals" by "oxides". Again – what is shown is only chemistry, no lattice structure information which is needed for statements on the mineralogy.

Done.

p. 7 Figure 2/3: Why aren't images shown of non-US treated particles? To understand what the US treatment does to the particles it would be helpful. Also, is the chemical composition the same? This goes into the same direction as above – does the composition change with size? If it does than the US could lead to differences due to breaking up larger particles. Also the crystal ratio for non-US treated particles could be determined and a better comparison would be possible.

We do not expect to see a visual difference due to the treatment with the ultrasonic bath, while surface charges might have changed, which will be difficult to detect with chemical analysis. We added to the manuscript "that the chemical composition of fly ash particles with aerodynamic diameters between 0.2 µm and 4.8 µm is remarkably consistent (Kaufherr and Lichtman, 1984)" (P4 L27-28). Therefore, a change in chemical composition for the fly ash is not to be expected, which would also imply that a breakup of larger particles would not lead to a change in chemical composition for this sample.

p. 8 l.24: The original publication is Vali (1971). It should at least be given in addition to Hartmann et al. (2013) if not replacing it.
Done.
p. 8 l.26: What does "saturation range" refer to in this case? The temperature range in which f ice saturates? Please be more precise!
We changed the formulation to "… with f_ice* being the ice fraction in the temperature range in which f_ice saturates." (P12 L8).
p.9 Figure 4 and p.13 Figure 5: I appreciate that the color code is mostly consistent throughout the paper. However, it would be helpful to use in addition different marker symbols and not only circles. Also, "Fly ash from brown coal burning susp. +US *4.54" is in Figure 4 given as open circle with dot and in Figure 5 as open circle without dot. Please change that.
With respect to the comments of the other referees, Fig. 4 (now Fig. 3) was changed to a 4 panel figure for better clarity. We think that the use of further marker symbols is not necessarily needed now. The suggested change in the case of "Fly ash from brown coal burning susp. +US *4.54" has been made.
p. 10 l.3: Which exact temperature range are you referring to "where effects of homogeneous nucleation can be ruled out"?
We detect effects of homogeneous nucleation at temperatures as cold as -38 °C. This limit was determined measuring the IN behavior of highly diluted ammonium sulfate droplets which should freeze due to homogeneous nucleation only (data is now included in Fig. 3d). We changed the cited sentence to: "In case of the wood ash particle, f_ice does not exceed 10 % between -35 °C and -37 °C." (P12 L13-14).
p.10 l.7: How is "start nucleating ice" defined? In Figure 4 f ice data points are shown at temperatures warmer than -33°C and -29°C, respectively, indicating that freezing is taking place also at warmer temperatures. You seem to refer only to the plateau regime. Please write this explicitly, as this is different to other experiments.
With "start nucleating ice", we mean that we detected an ice fraction that is above our detection limit. This was added in the manuscript (P14 L2). The given values are simply wrong and were corrected to -27 °C and -32 °C (P14 L2-3).
p.10 l. 18: "ice active sites on the particle surface might affect the ice nucleation ability of a substance." is not very meaningful as ice active sites are defined by the ice nucleation ability of a particle. This sentence can be deleted.
Done.
p. 10 l.18 ff: The whole discussion on active sites is hardly based on observations but mostly speculative. It may well be that the statement given here is correct but since the mineralogy is not even known talking about crystallographic dislocations
You are right, there is a lot of speculation in this sentence. However, we wanted to mention a possible reason for the differences in IN behavior of the ashes, even though this was observed for mineral dust and even though we do not know anything about the mineralogy. We clearly state that the given explanation is mere speculation in saying that this "might" be a factor.
p. 11 l.1: What type of amorphous material are you referring to? What could this material be based on the chemical composition information that you have? This sounds extremely vague.
From previous studies we know that the amorphous material in fly ash is mainly aluminosilicate glass (Ramsden and Shibaoka, 1982, Atmos. Environ., 16, 9; Querol et al., 1996, Atmos. Environ., 30, 21). This was added in the manuscript and related to the high content of SiO2 and Al2O3 in our fly ash sample (P14 L22-24).
p. 11 l.13: Is it possible to give a rough estimate by how much f ice would be reduced if multiple charge correction was applied?
As we selected 300 nm particles which is far to the right of the maximum of the size distribution, we made sure that a majority of multiply charged particles is not possible. To estimate the multiple charge fractions in the suspension measurements, we weighted the bipolar charge distribution (Wiedensohler, 1988, Journal of Aerosol Science), i.e., the probability of the particles to receive one, two or three negative charges in the neutralizer, with the measured size distributions of the ash-water suspensions.

We must mention that crystals will probably have occurred during the size distribution measurements as well and we are not sure how the size distributions look like for the insoluble only. It turned out, that the highest multiple charge fractions would probably occur for the fly ash suspension (+US), where we calculated 80.5 % singly, 16.8 % doubly, and 2.7 % triply charged particles. The multiple charge fractions were even lower for the other suspension samples. Would we perform the multiple charge correction using these fractions, our measured data would be reduced by a maximum factor of 2 only. A short version of this was added to the manuscript (P15 L10-16).

p. 11 l.30: The sentence about n_s does not belong here and should be moved further to the back.
Done.

p. 12 l.25: "increase" compared to what? I assume in comparison to LACIS. Then state that.
Done.

p.13 l. 4: How could the extended time of the ash particles in suspension lead to changes in n_s? Just a short comment what you ascribe it to? Reference that supports this suggestion?
We can only speculate what leads to the observed change in n_s. The extended time the samples spent in water could possibly lead to the dissolution of more material. Peckhaus et al., ACPD (2016, see Suppl. Fig. 6) observed dissolution of different elements from feldspar samples continuing for weeks. This has been added to the manuscript (P18 L1-5).

**Technical Remarks:**

General: SBM has been several times spelled SMB. Please check.
Corrected.
p.1 l.8: replace "effect" with "influence"
Done.
p.2 l.31: insert "alone" after "insoluble fraction"
Done.
p.8 l.2: replace "could be" by "was"
Done.
Figure 4: Grey and black as colors for the two SBM fits from the literature are hard to differentiate.
As we did not want to introduce yet another color, we used a lighter grey for the SBM fit of the clay mineral base line.
p.11 l.22: typo in "homogeneous"
Corrected.
p.11 l.31: New paragraph before "In the plateau…"
Done.
p.14 l.21: insert space between "at" and "the Leipzig"
Done.

References cited in our answers:

Andreae et al., Science (2004)
Damle et al., Aerosol Science and Technology (1981)
Flagan and Seinfeld, Fundamentals of Air Pollution Engineering (1988)
Kaufherr and Lichtman, Environ. Sci. Technol. (1984)
Murray et al., Phys. Chem. Chem. Phys. (2010)
Peckhaus et al., ACPD (2016)
Petters et al., GRL (2009)
Querol et al., Atmos. Environ. (1996)
Ramsden and Shibaoka, Atmos. Environ. (1982)
Steenari et al., Fuel (1999)
Szyrmer and Zawadzki, BAMS (1997)
Umo et al., ACP (2015)
Whale et al., AMT (2015)
Wiedensohler, Journal of Aerosol Science (1988)

---

## Author Response (AR2)

**Author's response**

**Referee #1**

We would like to thank the referee for reviewing the manuscript once again and for pointing out still existing ambiguities. The referee comments are colored, our answers are given in black.

I would like to thank the authors for improving the quality of their manuscript and for clarifying many important points listed by all 4 reviewers. Although most on the points highlighted in my original review were addressed (mainly the specific comments), there are key points that need to be clarified before the manuscript can be accepted. The main points are listed below:

1. The addition of Table 1 with the fraction of doubly charged particles is very valuable. However, I am surprised there are not triply charged particles in the studied ash samples. Note that you provided the triply charged particle fraction in the answer to point #4 but they are not given in the Table 1. The authors did not provide the particle size distribution (PSD) of the 300 nm size selected particles as requested. I think this needs to be part of the main text. Note that the PSD should be in log scale to better identify the doubly and possibly triply charged particles.

We are not sure in what way the results of the UHSAS measurements could contribute to the overall comprehensibility of the manuscript. Such a graph would only contain information that has already been included in the form of Table 1 (note: Table 1 gives the double charge fractions of the dry generated particles, only!). In our answers to the referee comments of the first discussion round, we did include a graph showing the size distribution of dry fly ash particles from UHSAS measurements. Because the referee asked for it a second time, we included the same example, now in logarithmic scale and showing explicitly how we determined the multiple charge fractions (see the three modal fit in the figure below). From the analysis of this measurement, we found a single charge fraction of 80 %, a double charge fraction of 18.9 % and a triple charge fraction of 0.9 %, i.e., the number of triply charged particles was negligibly low. The results for the other dry generated samples are presented in Tab. 1. As stated in the manuscript, the triple charge fractions were determined to be smaller than 1% in all cases. As none of the remaining referees saw a need for including the UHSAS measurements in the manuscript, we would like to avoid adding another graph which is not substantial to the topic itself.

With the triple charge fraction given in the answer to point 4 from the first discussion round, we were referring to an estimate for the suspensions (i.e., wet particle generation), which we calculated by weighing the bipolar charge distribution with the measured size distributions, i.e., this number is rather modeled than measured. The wet particle generation data was not corrected concerning these calculated values, because we produced particles from purely soluble material probably in all size ranges and cannot be sure how the size distributions look like for the particles from insoluble material only. This makes a correction impossible. This was discussed in our answers to point 4 in the last review, and parts of it were included in the manuscript (P15 L10-16).

[Figure]

2. It is really sad that the authors were not able to conduct the missing experiments. I think an effort in this direction could have been made by requesting an extension to submit the revised manuscript. "We decided to include the data, even without the chemical analysis, because we wanted to show that beech ash is comparable to the other wood bottom ashes." Although it is true that it is nice to show that the ice nucleating abilities of beech ash and the wood bottom ashes are comparable, the reviewer does not see the point to add this data if the reasons of why beech ash particles behave similarly to the wood bottom ashes are not provided. I mean, is this observation simply a coincidence, or is it because the chemical composition, soluble material, and/or particle's surface properties of both samples are really comparable?

We agree that additional LACIS measurements likely would have given some interesting insights. We would like to clarify that we usually obtain a few data points a day after hous of preparation and measurements, meaning that it would take us several months to conduct all the experiments that the referee proposed in the first discussion round. Furthermore, the instrument has been shut down for the last months because of construction work in our laboratory.

Admittedly, it is a bit unfortunate that we do not have the chemical composition analysis for the beech ash, because then we could have checked if its K content is similar to the one of spruce and birch. From our point of view, including the beech data is an interesting addition because its ice nucleation efficiency is so similar to the other wood ashes. We do not think that this is a coincidence but rather related to the amount of (in this case soluble) K. As this has been discussed in the manuscript already, nothing was changed.

3. The reviewer is not really satisfied with the answer given to this point: "Possibly, CaCO3 is the dominant phase of the water soluble fraction. During the combustion process, CaO is produced (present in the initial sample, see Fig. 1 and Table 3) which may react with H2O to form Ca(OH)2. CaCO3 may form from Ca(OH)2 upon reaction with CO2 from the air (Steenari et al., 1999). That we do not see any needles on the SEM image of spruce ash suspension particles (Fig. 2 (c)), even though this sample contains 11 % more CaO than the fly ash sample, is possibly due to the variety of different crystal shapes which CaCO3 is known to occur in and which include needles, hexagonal plates, and others (Kim et al., 2009). Particularly hexagonal plates might not be as easily

distinguishable from the insoluble particles as the prominent needles and might be seen, at least to some extent, in Fig. 2 (c), where the resolution of the pictures unfortunately does not allow a better analysis." The presence or absence of this large needle particles or small hexagonal plates after passing through the dryer needs to be better quantified given that they significantly change the size distribution of the particles entering the LACIS and hence the concentration and potentially the size of the cloud droplets. I think this points deserves a solid and clear explanation.

The fraction of solution particles should only depend on the concentration of ash in the suspension (which we kept constant throughout the experiments). As they are only a side effect of our particle generation, we focused on determining their abundance, which is important for the correction of the $f\_ice$ values. The size distribution of the droplets in LACIS is always comparably similar and independent of the shape of the aerosol we feed in, as high supersaturations (>10%) ensure that every single particle is activated to a droplet. Nothing was changed.

4. A good way to support the conclusion that the particles produced via the wet system are less efficient that those produced via the dry system will be adding the size distributions of the 300 nm (wet system) together with the size distribution of the dry system. In this figure readers will see how comparable the PSD are and what is the effect of multiple charges in the particles. The authors mentioned that the fraction of multiple charge particles is around 20% for the dry system and very low for the wet system. 20% is a large fraction and this can be the reason of why the dry particles showed a better ice nucleating efficiency. Is this 20% from the PSD obtained at the beginning of the experiment or at the end? The authors showed that the fraction of multiple charged particles increases with time.

As stated earlier, we likely produced a fraction of particles consisting of soluble material when we measured size distributions from atomizing the suspensions. Consequently, it would not make sense to directly compare the size distributions from wet and dry particle generation and expect to see a difference in the double charge fractions. We included this comparison here (see below) as it was requested by the referee, but we do not see any reason to include it in the manuscript. It might rather give the (wrong) impression that we know the size distribution of the insoluble particles generated from suspensions.

We indeed stated that the double charge fraction for particles from wet generation is possibly lower than what we determined for particles from dry generation, because we saw only few particles larger than 300 nm on the SEM images. However, this difference in the double charge fraction cannot be the reason for the higher ice nucleation efficiency of the particles from dry generation, because for the latter we performed a double charge correction. Basically, this means that we subtracted the ice fraction caused by the large doubly charged particles from our measured ice fractions (for details, see Hartmann et al., 2016, JAS). This correction was not done for the particles from wet generation, and the correction would rather decrease the $f\_ice$ values further. Regarding this, we said the following concerning the wet generated particles in the manuscript (P15 L11 ff): "We found that the highest amount of multiply charged particles was probably present in the experiments with the fly ash suspension with ultrasonic treatment (80.5 % singly, 16.8% doubly, and 2.7 % triply charged particles). Would we perform the multiple charge correction using these fractions, our measured data would be reduced by a maximum factor of 2 only." Therefore, an uncertainty related to the omitted multiple charge correction has been given in the text, which we feel is all that is needed.

Yes, the fraction of doubly charged particles increases with time. However, this increase is only small over the first 30 minutes of particle generation. As said in the manuscript, and our answers in the first discussion round, all LACIS measurements are done within these first 30 minutes. After thawing

and re-icing, our particle generation setup was prepared for a new experiment, which includes cleaning of the cyclone and checking of the flows. The double charge fractions presented in Table 1 correspond to the mean values of doubly charged particles in LACIS during these first 30 minutes of dry particle generation. Unfortunately, we could not check the multiple charge fractions continuously with the UHSAS over the entire measurement period, as the instrument was not available at all times. However, they were checked on several occasions.

[Figure]

5. Given that lack of data I think the following statement should be softened or the lack of data acknowledged: "There is a trend of beech bottom ash being the most effective"

We are only talking about a trend here, meaning that a tendency can be seen but there is no profound evidence. We are not sure how to soften this statement further, so nothing was changed.

6. "We revised this section in terms of citing more field observations of fly ash particles and calculating the in-stack concentration especially for the power-plant Lippendorf. Even though the section contains a lot of speculation, we would like to keep it in the manuscript. Firstly, we clearly state when we make a certain assumption and secondly we think that this estimate supports the relevance of the topic". The reviewer considers that there is no need to keep this in the manuscript. It does not add anything valuable to the paper and it will confuse readers which are not very familiar with this topic/calculation.

We are still of the opinion that Sec. 4 should remain in the manuscript because it puts our results into perspective. Furthermore, we have often been asked to include an estimate of the atmospheric relevance in other papers from our group in the past. As none of the remaining referees wants any more changes with respect to this section, we would not want to omit it.

**Referee #2**

We would like to thank the referee for reviewing the manuscript once again. The original comments are colored, our answers are given in black.

General comments:
I have two more comments regarding the BET calculations:
- First, the new left y-axis in Fig. 4 is dispensable from my point of view as it is the same as the right y-axis. The concern to compare the same metrics to each other is not addressed with this second y-axis. I agree that it might be too much to get the BET surface for your samples, however, since you did already an estimate of the conversion factor between your ns and the ns, BET provided by Umo et al. (2015), why don't you use it in the plot? You could provide an uncertainty area around the curve. This would be more valuable than duplicating the y-axis.

It was our intention to convey the difference between the two n_s metrics by adding the second axis label. Converting our data into BET based n_s values and using only one axis might make the wrong impression, i.e., that we know the specific surface area which we did not measure. In the text, we discuss the possible uncertainty that may arise from comparing geometric and BET based n_s values. As a result, we would prefer to leave Fig. 4 unchanged.

- Second, for the calculation on P17,L24 the volume of the particles needs to be calculated, therefore the volume equivalent diameter needs to be used instead of the mobility diameter (Kulkarni 2011). You assumed a spherical shape and a mobility diameter of 300nm for your particles. The validy of using the mobility diameter depends on the shape factor of the ash particles. This is not considered in the calculations. From your manuscript I understand that for fly ash particles the assumption of sphericity seems to be valid. However, the SEM images do not support this assumption for the bottom ash particles. Please re-consider your calculations. This should be included in an uncertainty band around the curve in Fig.4

You are of course right in saying that the volume equivalent diameter needs to be used for the calculation. However, when assuming spherical particles, volume equivalent and mobility diameters are identical. The assumption of spherical particles has been made because we cannot give a profound estimate of the shape factor without analyzing a statistically relevant amount of particles on SEM images with higher resolution than the ones currently available. Furthermore, we do not expect the deviation from the spherical shape to cause a difference of 3 orders of magnitude in n_s, i.e., a significant improvement in the agreement between our data and those from Umo et al. (2015) is not expected. Hence, we would like to avoid the time consuming derivation of the volume-equivalent diameter of the non-spherical ash particles.

Technical comments:
- Delete "(a)" on p.13, first line of Figure 3 caption. The full circles are shown in all other subplots as well, not only in (a).

Done.

**Referee #3**

Referee #3 agreed with the current version of the manuscript and supported publication.

**Marked differences**

Nothing was changed in the manuscript except the deletion of (a) on the first line of the caption of Fig. 3.

---

## Author Response (AR3)

**Author's response**

We now added a number of the points that were mentioned by referees #1 and #2 in the second discussion round to the manuscript. Some of our answers have therefore changed slightly and now include citations (marked in blue) of the added sentences. However, we feel we cannot address all the issues in the manuscript because they would lead to impressions we cannot support. The original referee comments are given in red, our answers are given in black.

**Referee #1**

I would like to thank the authors for improving the quality of their manuscript and for clarifying many important points listed by all 4 reviewers. Although most on the points highlighted in my original review were addressed (mainly the specific comments), there are key points that need to be clarified before the manuscript can be accepted. The main points are listed below:

1. The addition of Table 1 with the fraction of doubly charged particles is very valuable. However, I am surprised there are not triply charged particles in the studied ash samples. Note that you provided the triply charged particle fraction in the answer to point #4 but they are not given in the Table 1. The authors did not provide the particle size distribution (PSD) of the 300 nm size selected particles as requested. I think this needs to be part of the main text. Note that the PSD should be in log scale to better identify the doubly and possibly triply charged particles.

With the triple charge fraction given in the answer to point 4 from the first discussion round, we were referring to an estimate for the suspensions (i.e., wet particle generation), which we calculated by weighing the bipolar charge distribution with the measured size distributions, i.e., this number is rather modeled than measured. The wet particle generation data was not corrected concerning these calculated values, because we produced particles from purely soluble material probably in all size ranges and cannot be sure how the size distributions look like for the particles from insoluble material only. This makes a correction impossible. This was discussed in our answers to point 4 in the last review, and parts of it had been previously included in the manuscript (P16 L10-16): "[…], we estimated the multiple charge fractions in the suspension measurements by weighting the bipolar charge distribution (Wiedensohler, 1988) with measured size distributions. We found that the highest amount of multiply charged particles was probably present in the experiments with the fly ash suspension with ultrasonic treatment (80.5 % singly, 16.8 % doubly, and 2.7 % triply charged particles). Would we perform the multiple charge correction using these fractions, our measured data would be reduced by a maximum factor of 2 only. There is a caveat to this estimate, as crystals will probably have been present during the size distribution measurements as well and we cannot be sure how the size distributions would look like for the insoluble particles only.".

Concerning the size distributions after dry particle generation and size selection, which were measured with a UHSAS, we now included an exemplary measurement for dry generated fly ash particles in the manuscript (Fig. 1, see below). This shows explicitly how we determined the multiple charge fractions (see the four modal fit). From the analysis of this measurement, we found a singly charged fraction of 79.8 %, a doubly charged fraction of 18.9 %, a triply charged fraction of 0.9 %, and a quadruply charged fraction of 0.4 %, i.e., the number of particles with three of more negative charges was negligibly low. These particles contribute only 6 % to the total surface. For a proper correction for the triply and quadruply charged particles, ice nucleation measurements at these sizes would preferentially be needed, but cannot be done, as the particle number concentrations for the triply charged particles are already too low and the quadruply charged particles cannot be selected with a DMA. Therefore we only corrected for doubly charged particles. In the manuscript we added

the following (P4 L33 - P5 L9): "An example of the UHSAS measurements with dry generated brown coal fly ash particles after size selection can be seen in Fig. 1. Note that the UHSAS detects the optical particle diameter which is smaller than the electrical mobility diameter in the shown example. To determine the multiple charge fractions, a four-modal normal distribution was fit to the measured data and the respective integrals were determined. In this case, we found 79.8 % singly, 18.9 % doubly, 0.9 % triply, and 0.4 % quadruply charged particles, i.e., the triply and quadruply charged particles contribute only 6 % to the total surface area. Because of the low fraction of particles with three or more negative charges, which was in the range of 1 % for all of the dry dispersed ash samples, only the doubly charged particles (see Table 1) were accounted for the correction of the immersion freezing experimental results. A detailed explanation of the multiple charge correction procedure is given by Hartmann et al. (2016)."

[Figure]

2. It is really sad that the authors were not able to conduct the missing experiments. I think an effort in this direction could have been made by requesting an extension to submit the revised manuscript. "We decided to include the data, even without the chemical analysis, because we wanted to show that beech ash is comparable to the other wood bottom ashes." Although it is true that it is nice to show that the ice nucleating abilities of beech ash and the wood bottom ashes are comparable, the reviewer does not see the point to add this data if the reasons of why beech ash particles behave similarly to the wood bottom ashes are not provided. I mean, is this observation simply a coincidence, or is it because the chemical composition, soluble material, and/or particle's surface properties of both samples are really comparable?

We changed the first two paragraphs of Sec. 3.1 (P13 L 4 – P15 L17) in such a way, that it becomes clearer that we relate the difference in ice nucleation efficiency between wood and coal ashes to the amount and solubility of potassium in the samples. We now say the following right in the beginning of Sec. 3.1 (P13 L6 - P15 L3): "It is striking that the ice nucleation efficiency of all three examined wood ashes is very similar, which leads us to the conclusion that the influence of the burned wood type on the immersion freezing behavior of the bottom ash particles is small for the investigated samples. In this context, the amount of K in the wood ashes could play a role. Although no chemical composition analysis was performed in case of beech ash, it is known that wood ash in general

contains K (Steenari et al., 1999a), which has also been found in the here examined spruce and birch ash samples (see Fig. 2 and Table 3). Additionally, it has been shown that different wood ash samples contain similar amounts of K (Steenari et al., 1999a), justifying the assumption that the beech ash sample also contains K in amounts similar to the other two wood ashes. On the one hand, the fact that K in wood ashes is largely soluble, because it is present in the form of soluble salts and oxides (Steenari et al., 1999b), might be the reason for the low ice nucleation efficiency of the wood ashes. On the other hand, the supposedly similar amount of soluble K could be the reason for the comparable ice nucleation efficiency of the three investigated wood ashes. […] Again, the ice nucleation efficiency of the coal ashes could be related to the K content, because, in contrast to the soluble K in wood ashes, coal ashes contain K in clay minerals with low solubility (Steenari et al., 1999b)."

3. The reviewer is not really satisfied with the answer given to this point: "Possibly, $CaCO_3$ is the dominant phase of the water soluble fraction. During the combustion process, $CaO$ is produced (present in the initial sample, see Fig. 1 and Table 3) which may react with $H_2O$ to form $Ca(OH)_2$. $CaCO_3$ may form from $Ca(OH)_2$ upon reaction with $CO_2$ from the air (Steenari et al., 1999). That we do not see any needles on the SEM image of spruce ash suspension particles (Fig. 2 (c)), even though this sample contains 11 % more $CaO$ than the fly ash sample, is possibly due to the variety of different crystal shapes which $CaCO_3$ is known to occur in and which include needles, hexagonal plates, and others (Kim et al., 2009). Particularly hexagonal plates might not be as easily distinguishable from the insoluble particles as the prominent needles and might be seen, at least to some extent, in Fig. 2 (c), where the resolution of the pictures unfortunately does not allow a better analysis." The presence or absence of this large needle particles or small hexagonal plates after passing through the dryer needs to be better quantified given that they significantly change the size distribution of the particles entering the LACIS and hence the concentration and potentially the size of the cloud droplets. I think this points deserves a solid and clear explanation.

The fraction of solution particles should only depend on the concentration of ash in the suspension (which we kept constant throughout the experiments). As they are only a side effect of our particle generation, we focused on determining their abundance, which is important for the correction of the f_ice values. The size distribution of the droplets in LACIS is always comparably similar and independent of the shape of the aerosol we feed in, as high supersaturations (>10%) ensure that every single particle is activated to a droplet. We included the following in the manuscript (P16 L26-29): "As mentioned in Sec. 2, the crystalline soluble particles were also observed in the form of hexagonal plates. However, this difference in shape does not influence the size distribution of the cloud droplets in LACIS as high supersaturations (> 10 %) ensure that every particle is activated. Hence, only the fraction of crystalline particles in comparison to the insoluble particles must be determined. For this,…".

4. A good way to support the conclusion that the particles produced via the wet system are less efficient that those produced via the dry system will be adding the size distributions of the 300 nm (wet system) together with the size distribution of the dry system. In this figure readers will see how comparable the PSD are and what is the effect of multiple charges in the particles. The authors mentioned that the fraction of multiple charge particles is around 20% for the dry system and very low for the wet system. 20% is a large fraction and this can be the reason of why the dry particles showed a better ice nucleating efficiency. Is this 20% from the PSD obtained at the beginning of the experiment or at the end? The authors showed that the fraction of multiple charged particles increases with time.

As stated earlier, we likely produced a fraction of particles consisting of soluble material when we measured size distributions from atomizing the suspensions. Consequently, it would not make sense to directly compare the size distributions from wet and dry particle generation and expect to see a difference in the double charge fractions. We included this comparison here (see below) as it was requested by the referee, but would like to avoid showing it in the manuscript. It might rather give the (wrong) impression that we know the size distribution of the insoluble particles generated from suspensions.

We indeed stated that the double charge fraction for particles from wet generation is possibly lower than what we determined for particles from dry generation, because we saw only few particles larger than 300 nm on the SEM images. However, this difference in the double charge fraction cannot be the reason for the higher ice nucleation efficiency of the particles from dry generation, because for the latter we performed a double charge correction. Basically, this means that we subtracted the ice fraction caused by the large doubly charged particles from our measured ice fractions (for details, see Hartmann et al., 2016, JAS). This correction was not done for the particles from wet generation, and the correction would rather decrease the f_ice values further. Regarding this, we already had said the following concerning the wet generated particles in the manuscript (P16 L10-14): "However, we estimated the multiple charge fractions in the suspension measurements by weighting the bipolar charge distribution (Wiedensohler, 1988) with measured size distributions. We found that the highest amount of multiply charged particles was probably present in the experiments with the fly ash suspension with ultrasonic treatment (80.5 % singly, 16.8% doubly, and 2.7 % triply charged particles). Would we perform the multiple charge correction using these fractions, our measured data would be reduced by a maximum factor of 2 only." Therefore, an uncertainty related to the omitted multiple charge correction has been given in the text, which we feel is all that is needed.

Yes, the fraction of doubly charged particles increases with time. However, this increase is only small over the first 30 minutes of particle generation. As said in the manuscript (P10 L6-7), and in our answers in the first discussion round, all LACIS measurements are done within these first 30 minutes. After thawing and re-icing, our particle generation setup was prepared for a new experiment, which includes cleaning of the cyclone and checking of the flows. The double charge fractions presented in Table 1 correspond to the mean values of doubly charged particles in LACIS during these first 30 minutes of dry particle generation. Unfortunately, we could not check the multiple charge fractions continuously with the UHSAS over the entire measurement period, as the instrument was not available at all times. However, they were checked on several occasions.

[Figure]

5. Given that lack of data I think the following statement should be softened or the lack of data acknowledged: "There is a trend of beech bottom ash being the most effective"

The sentence was changed in the following way (P13 L6-7): "It is striking that the ice nucleation efficiency of all three examined wood ashes is very similar…"

6. "We revised this section in terms of citing more field observations of fly ash particles and calculating the in-stack concentration especially for the power-plant Lippendorf. Even though the section contains a lot of speculation, we would like to keep it in the manuscript. Firstly, we clearly state when we make a certain assumption and secondly we think that this estimate supports the relevance of the topic". The reviewer considers that there is no need to keep this in the manuscript. It does not add anything valuable to the paper and it will confuse readers which are not very familiar with this topic/calculation.

We are still of the opinion that Sec. 4 should remain in the manuscript because it puts our results into perspective. Furthermore, we have often been asked to include an estimate of the atmospheric relevance in other papers from our group in the past. As none of the remaining referees wants any more changes with respect to this section, we feel that the deletion of the entire section would have to be discussed with them.

**Referee #2**

General comments:
I have two more comments regarding the BET calculations:
- First, the new left y-axis in Fig. 4 is dispensable from my point of view as it is the same as the right y-axis. The concern to compare the same metrics to each other is not addressed with this second y-axis. I agree that it might be too much to get the BET surface for your samples, however, since you did already an estimate of the conversion factor between your ns and the ns, BET provided by Umo et al. (2015), why don't you use it in the plot? You could provide an uncertainty area around the curve. This would be more valuable than duplicating the y-axis.

It was our intention to convey the difference between the two $n_s$ metrics by adding the second axis label. Converting our data into BET based $n_s$ values and using only one axis might make the wrong impression, i.e., that we know the specific surface area which we did not measure for our samples but took from typical literature values instead. In the text, we discuss the possible uncertainty that may arise from comparing geometric and BET based $n_s$ values (P18 L26 - P 19 L2): "To make sure that this disagreement is not caused by the difference in particle surface area determination, we also calculated BET-based n s values for our measurements. For this, the particle surface area was determined according to $A_p = \rho \cdot V_p \cdot SSA$, with $\rho$ the ash density taken from the literature (wood bottom ash: 827 kg m$^{-3}$, Naik et al., 2001; brown coal bottom ash: 1415 kg m$^{-3}$, U.S. Department of Transportation, 1998; brown coal fly ash: 2456 kg m$^{-3}$, Shoumkova et al., 2005), $V_p$ the particle volume assuming a spherical shape and a diameter of 300 nm, and SSA the specific surface area reported by Umo et al. (2015) from BET measurements (wood bottom ash: 6.98 m$^2$ g$^{-1}$; coal bottom ash: 8.86 m$^2$ g$^{-1}$; coal fly ash: 2.54 m$^2$ g$^{-1}$). It must be noted that the BET specific surface area is a sample specific quantity. However, as similar materials were combusted at similar conditions, we may assume that the BET values reported by Umo et al. (2015) are comparable to those of our samples. With this, $n_{s\_BET}$ values derived from our data would even increase, in maximum by a factor of 3.5 in comparison to the $n_s$ values shown as circles in Fig. 5, meaning that effects other than the difference in surface area determination must be responsible for the discrepancy between our data set and that by Umo et al. (2015)."
To avoid the wrong impression that we can derive $n_{s\_BET}$ for our LACIS measurements, we do not want to change the figure.

- Second, for the calculation on P17,L24 the volume of the particles needs to be calculated, therefore the volume equivalent diameter needs to be used instead of the mobility diameter (Kulkarni 2011). You assumed a spherical shape and a mobility diameter of 300nm for your particles. The validy of using the mobility diameter depends on the shape factor of the ash particles. This is not considered in the calculations. From your manuscript I understand that for fly ash particles the assumption of sphericity seems to be valid. However, the SEM images do not support this assumption for the bottom ash particles. Please re-consider your calculations. This should be included in an uncertainty band around the curve in Fig.4

To account for the non-sphericity we now assume a dynamic shape factor of 1.25 which is the largest value that has been reported by Kaaden et al. (Tellus 61B, 2009) for mineral dust particles. Calculating the volume-equivalent diameter and from that the surface area leads to a change in $n_s$ of a factor of 0.6 in comparison to the assumption of spherical particles. Adding uncertainty bands in Fig. 5 does not make sense as these would be narrower than the actual symbol size. We included the following in the manuscript (P19 L3-8): "To estimate a possible influence of particle non-sphericity on $n_s$, one could assume a dynamic shape factor of chi = 1.25, which was observed for atmospheric dust particles (Kaaden et al., 2009) but which is likely much larger than chi for the here examined ash

particles. This would only account for an increase in n_s by a factor of 1.4. Even a much larger chi of 1.57 as given by Hinds (1999) for sand would only increase n_s by a factor of 1.9. So overall, omitting the possible effects of non-sphericity, as was done in this work, only accounts for comparably small changes in n_s and cannot explain the difference between our fly ash sample and the one examined by Umo et al. (2015)."

Technical comments:
- Delete "(a)" on p.13, first line of Figure 3 caption. The full circles are shown in all other subplots as well, not only in (a).

Done.

**Marked differences**

See next pages.

[revised manuscript text omitted]